# Ductile deformation during carbonation of serpentinized peridotite

Manuel D. Menzel [1,2 ✉], Janos L. Urai[1], Estibalitz Ukar[3], Greg Hirth[4], Alexander Schwedt[5], András Kovács [6], Lidia Kibkalo[6] & Peter B. Kelemen [7]

Carbonated serpentinites (listvenites) in the Samail Ophiolite, Oman, record mineralization of 1–2 Gt of $CO_2$, but the mechanisms providing permeability for continued reactive fluid flow are unclear. Based on samples of the Oman Drilling Project, here we show that listvenites with a penetrative foliation have abundant microstructures indicating that the carbonation reaction occurred during deformation. Folded magnesite veins mark the onset of carbonation, followed by deformation during carbonate growth. Undeformed magnesite and quartz overgrowths indicate that deformation stopped before the reaction was completed. We propose deformation by dilatant granular flow and dissolution-precipitation assisted the reaction, while deformation in turn was localized in the weak reacting mass. Lithostatic pore pressures promoted this process, creating dilatant porosity for $CO_2$ transport and solid volume increase. This feedback mechanism may be common in serpentinite-bearing fault zones and the mantle wedge overlying subduction zones, allowing massive carbonation of mantle rocks.

[1] Tectonics and Geodynamics, RWTH Aachen University, Lochnerstrasse 4-20, D-52056 Aachen, Germany. [2] now at: Instituto Andaluz de Ciencias de la Tierra (IACT) (CSIC-Universidad de Granada), Avenida de las Palmeras 4, 18100 Armilla, Granada, Spain. [3] The University of Texas at Austin, Bureau of Economic Geology, Austin, TX, USA. [4] Brown University, Department of Earth, Environmental and Planetary Sciences, Providence, RI, USA. [5] RWTH Aachen University, Central Facility for Electron Microscopy, Aachen, Germany. [6] Ernst Ruska-Centre for Microscopy and Spectroscopy with Electrons, Forschungszentrum Jülich, Jülich, Germany. [7] Lamont–Doherty Earth Observatory, Columbia University, New York, NY, USA. ✉email: manuel.menzel@csic.es

Listvenites–fully carbonated peridotites mostly composed of Mg-rich carbonate minerals and quartz[1]–can be a natural analogue for carbon storage applications[2,3], and may offer an opportunity to study mass transfer and deformation processes acting at the leading edge of the mantle wedge[4]. Listvenites commonly occur along major shear zones or faults that can act as fluid conduits (e.g.[5–9]), but the extent to which deformation plays a role in carbonation reactions is poorly constrained.

Deformation can enhance fluid-rock interactions, and altered rocks are far more abundant in tectonically active zones. Thermo-hydro-mechanical-chemical processes of coupled fluid flow, metasomatism, and deformation are common in extensional detachments, oceanic transform faults, and the plate interface of subduction zones[10–15]. The presence of aqueous fluids can enhance deformation by lowering effective stress, allowing dilatancy and activation of dislocation creep as well as pressure solution at high water activities[16–19]. At multiple scales, this involves nonlinear coupling of deformation, fluid flow and chemical reactions, over time scales that are difficult to investigate in the laboratory.

The formation of listvenite requires prolonged fluid flow that adds about 30 wt% $CO_2$ to the rock (e.g.[6,20],). Even for unusually high $CO_2$ contents in aqueous fluids (on the order of 1 wt%), this requires time-integrated fluid rock ratios > 30. Serpentinization and carbonation converting peridotite to listvenite involve a combined increase in solid volume by up to 68%[21–24], potentially clogging pore space and decreasing permeability with reaction progress. This effect, in combination with the formation of reaction rims that can inhibit diffusion and reaction, is the main reason why carbonation is inferred to be self-limiting in many experiments[25–27]. In natural settings such as the Samail ophiolite, where massive carbonation of peridotite to listvenite went to completion in volumes of 1–2 km[38] (equivalent to 1–2 Gt $CO_2$), tectonic stress and related deformation may play a key role in maintaining sufficiently high permeability for complete carbonation. Positive feedback mechanisms of deformation on permeability and reactivity include: (i) grain size reduction and related increase in reactive surface area[18], (ii) creep cavitation during viscous grain boundary sliding and pressure solution[28], (iii) dilatancy during granular flow[29], and (iv) fluid transport along fractures[30,31]. Listvenites contain a rich variety of (micro)structures, such as preserved reaction fronts, a multitude of veins, and growth zoning in magnesite due to variable redox conditions during reaction progress[6,20,32]. Their textural evolution over time can shed light on the relationship between carbonation and deformation.

The Samail ophiolite exposes large-scale $CO_2$-fluid-rock interactions in mantle rocks[8,9]. The Oman Drilling Project (OmanDP) was an international endeavor to systematically sample key sections of the oceanic lithosphere from crust to the basal thrust. OmanDP Hole BT1B[32] provided a unique sample of fully carbonated mantle rocks (listvenites) unaffected by surface weathering for targeted analysis.

In this study we analyze the temporal evolution of microstructures in listvenite of Hole BT1B to understand the interaction between changing rheology, effective stress, and fluid flow during progressive carbonation of peridotite. We use multiscale optical and electron microscopic imaging and analysis to study the temporal relationships between phase changes and rock fabrics. Results indicate syn-carbonation, brittle and ductile deformation, and creation of inter- and intra-granular porosity, which, together with the abundant veins present in these rocks, contributed to the permeability network.

## Results

**Geological background**. Listvenites crop out as thick bands in serpentinite at or close to the base of the Samail ophiolite (Fig. 1).

Hole BT1B recovered 196 m of listvenite and serpentinite, and, separated by a fault, 104 m of underlying metamorphic sole (Fig. 1d)[32]. The presence of quartz-serpentine intergrowths, together with recrystallized quartz and chalcedony after opal suggest temperatures of 80–150 °C during listvenite formation[8]. Low temperatures are supported by a nearly complete lack of talc, and by intergrown hematite and graphite or amorphous carbon, which require <200 °C if formed in equilibrium[33]. Clumped isotope thermometry points to temperatures from 45 ± 5 to 247 ± 52 °C for carbonate precipitation in listvenite and serpentinite[8,34], consistent with the inferences from mineral parageneses. The pressure of listvenite formation is poorly constrained, with a possible range from ~0.3 GPa (considering that they occur within the basal peridotite unit[33] and assuming 8–10 km ophiolite thickness, consistent with data from autochthonous carbonates below the ophiolite[35]) to the pressure recorded by the metamorphic sole as an upper bound (0.7–1.0 GPa[36,37]). An internal Rb-Sr isochron with an imprecise age of 97 ± 29 Ma (2σ)[8] (2σ uncertainty confirmed by personal communication with L. Falk) suggests that listvenite formation may have been concurrent with intra-oceanic subduction (~ 96–85 Ma; e.g.[38]), continental subduction of the Arabian margin and ophiolite obduction (85 – 74 Ma; e.g.[39],), and/or an early phase of post-obduction extension[35,40]. Sr and C stable isotope geochemistry suggests that the likely source of $CO_2$-bearing fluids was meta-sedimentary rocks similar to those of the underlying Hawasina Formation (Fig. 1)[8,41].

Listvenite outcrops north and northeast of Site BT1 define a broad antiform[8,33]. In this structure, they are overlain by banded peridotites of the base of the Samail ophiolite, and underlain by the metamorphic sole and multiply deformed sediments of the Hawasina Formation. These structural characteristics indicate that the listvenites formed within or just above a shallowly dipping fault zone along the interface between the ophiolite, the metamorphic sole, and (meta-)sediments beneath the ophiolite.

Structural core logging and microstructural investigation of cross-cutting relationships reveal multi-stage deformation and fluid-rock interaction in listvenites and serpentinites[32]. The listvenites in BT1B have been deformed and overprinted by cataclasis and faults[42], which obscured pre- and syn-carbonation textures. Here we focus on segments of the core that are not overprinted.

**Serpentinite - the starting material**. Serpentinites commonly show mesh/bastite textures and serpentine veins, typical of hydrated mantle peridotite[43]. Relict olivine or pyroxene are absent in core and in ~1 to 10 m serpentinite reaction zones surrounding listvenite in outcrops, but relicts of primary Cr-spinel are common. Locally, mesh cells (delineated by Fe-oxides) are flattened, defining a foliation (Fig. 2a). Localized deformation of serpentinite in shear zones produced a strong shape preferred orientation (SPO) and crystallographic preferred orientation (CPO) of serpentine (Fig. 2b), with mesh structures obliterated by medium- to fine-grained serpentine. In these shear zones, electron backscatter diffraction (EBSD) analysis indicates that crystalline antigorite is absent or rare. Moderately coarse lizardite shows undulose extinction, deformation lamellae, and kinking, whereas fine-grained serpentine develops along grain boundaries of coarser crystals and within aligned bands (Fig. 2c). Such textures indicate strong deformation[44,45]. Locally, serpentine veins and tabular serpentine aggregates are folded (Supplementary Fig. S5). Cr-spinel in carbonate-bearing serpentinite (ophicarbonate) is commonly fragmented and partially replaced by Fe-chromite or carbonate. Anastomosing carbonate veins resemble an S-C fabric with the primary vein orientation parallel to serpentine cleavage planes (Fig. 1g).

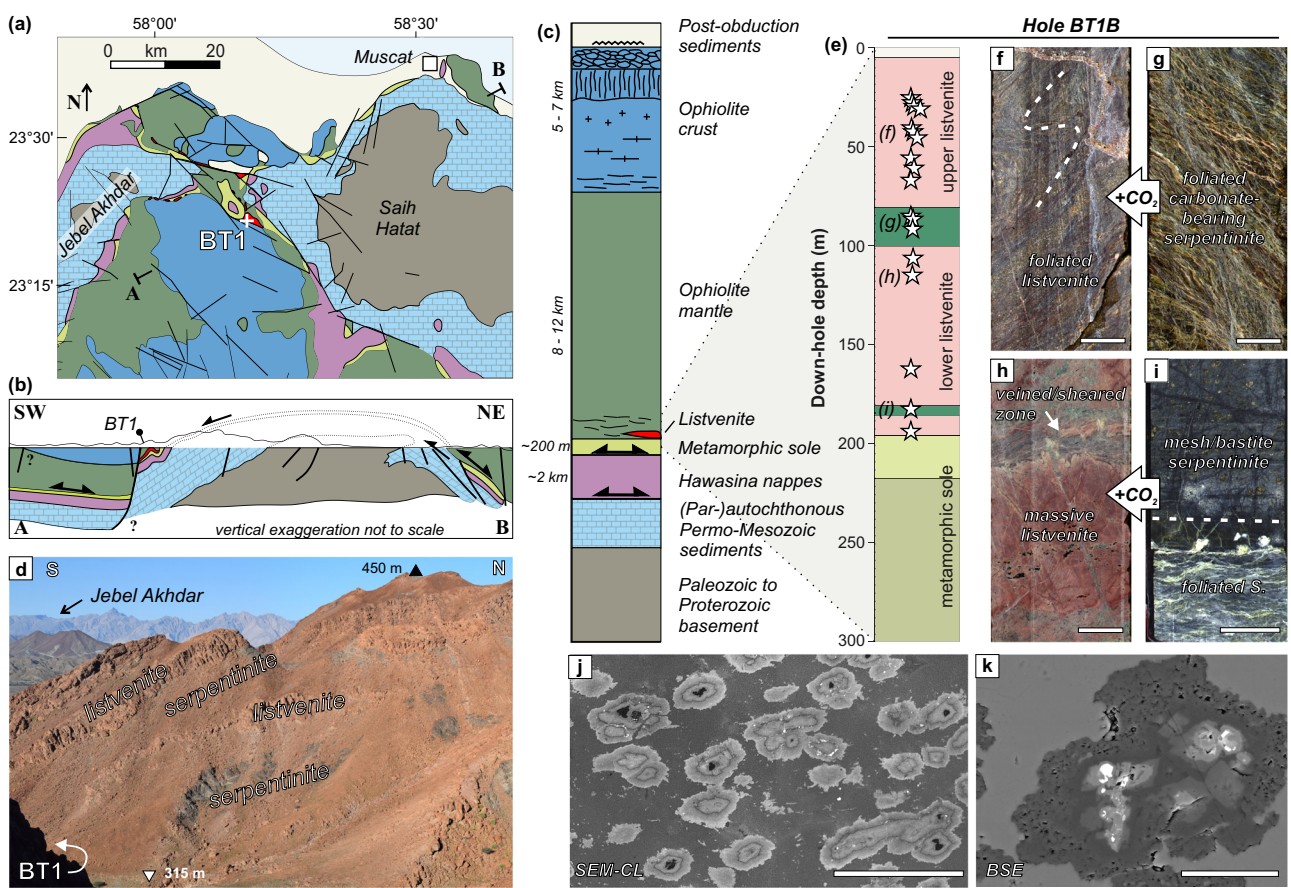

**Fig. 1 Geological context and key features of Oman Drilling site BT1. a** Simplified geological overview of the northern Samail massif in Northern Oman (adapted after[73]), with site BT1 indicated, **b** schematic cross-section, and **c** simplified stratigraphy of the Samail ophiolite and underlying units (from[74]). Present-day thickness of units in the area is commonly reduced due to tectonic thinning. **d** Field view of listvenites and serpentinites close to site BT1. Serpentinite is commonly covered by listvenite scree. **e** Overview of Hole BT1B (modified from[32]) with the location of studied samples. Split core images of **f** foliated and folded listvenite (sample BT1B_21-3_35-40), **g** foliated carbonate-bearing serpentinite (sample BT1B_39-4_14-18), **h** veined shear zone in massive, non-foliated listvenite (sample BT1B_51-1_20-25), and **i** zone of foliated serpentinite cutting non-foliated mesh-bastite textured serpentinite (sample BT1B_74-1_59-62). **j** Scanning-electron microscopy cathodoluminescence (SEM-CL) image showing zoning of magnesite ellipsoids with shape preferred orientation (BT1B_14-3_77-80). **k** Backscatter electron (BSE) image of a magnesite ellipsoid with Fe-oxide inclusions (brightest) and Fe-magnesite cores (BT1B_15-1_32-34). Scale bar in **f–i** is 2 cm, in **j** 100 μm and in **k** 25 μm.

**Listvenite**. Listvenite in BT1B consists of magnesite, quartz, and minor Cr-spinel, Fe-oxides and, locally, chromian muscovite[32]. Dolomite is the dominant carbonate mineral in a few core intervals, and in listvenite bands north of Site BT1[8,32]. In polished wadi outcrops and in core, listvenites can be massive or foliated (Fig. 1, Supplementary Fig. S1). Microscopically, BT1B listvenites are commonly characterized by a high density of early, subparallel carbonate veins, which often define the foliation. In the listvenite matrix between these carbonate veins two microstructures are common: (i) spheroidal / ellipsoidal to euhedral magnesite in a finer-grained quartz or quartz-chalcedony groundmass (Fig. 1j, k), and (ii) magnesite-quartz intergrowths that resemble the protolith serpentinite mesh microstructure[32]. Locally, this matrix has a penetrative foliation oblique to the preferred orientation of early veins.

**Foliation in listvenite matrix**. Penetrative-foliated listvenites containing folded and transposed veins are most evident in 25 m–67 m and 188 m–197 m depth intervals of Hole BT1B (Fig. 1), although often not visible macroscopically. These foliated listvenites constitute ~10% (13/115) of the studied listvenite thin sections, and are present in mm-scale shear zones in 3 additional samples (Supplementary Table S1). The penetrative foliation is defined by clustered, elongated magnesite ellipsoids (Fig. 1j), aligned magnesite dendrites, hematite grain aggregates (Supplementary Fig. S13), and/or carbonate vein fragments. Scanning electron microscopy (SEM) compositional mapping (EDS) and SEM-cathodoluminescence (CL; Methods) show that in many cases, ellipsoidal magnesites have a Fe-rich core and concentric compositional zoning of Fe, Mg, Mn and Ca and variable abundance of $SiO_2$ inclusions (Fig. 1j, k), similar to magnesite in non-foliated listvenite[34]. In 3D, X-ray microtomography (micro-CT) shows that magnesite ellipsoids are oblate in the foliation plane (Supplementary Fig. S6). EBSD analysis shows that in some samples with an SPO of magnesite, quartz also has a weak SPO (Supplementary Fig. S11).

In foliated listvenites, the foliation wraps around relict Cr-spinel porphyroclasts, which occasionally form boudins or have sigmoidal strain shadows marked by hematite grain aggregates. Figure 3 shows a boudinaged Cr-spinel with magnesite single crystals in the boudin necks and Fe-magnesite in the interstices between partially rotated spinel fragments, with magnesite orientation distinct from the CPO of magnesite ellipsoids in the matrix. Magnesite has abundant low-angle grain boundaries (Fig. 3g); fibrous aggregates that are typical of strain shadows[46] are not evident. Magnesite in the boudin neck has a patchy

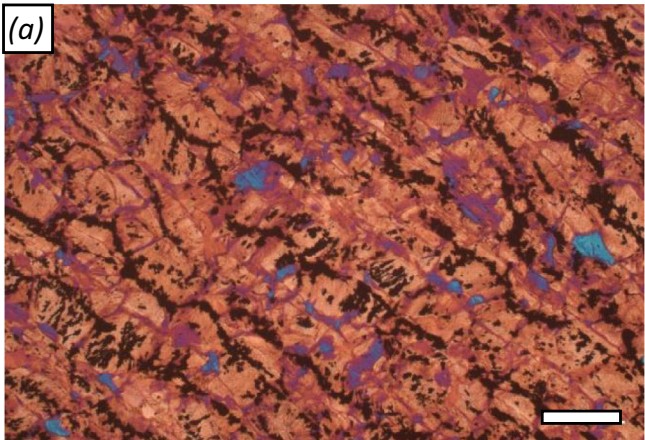

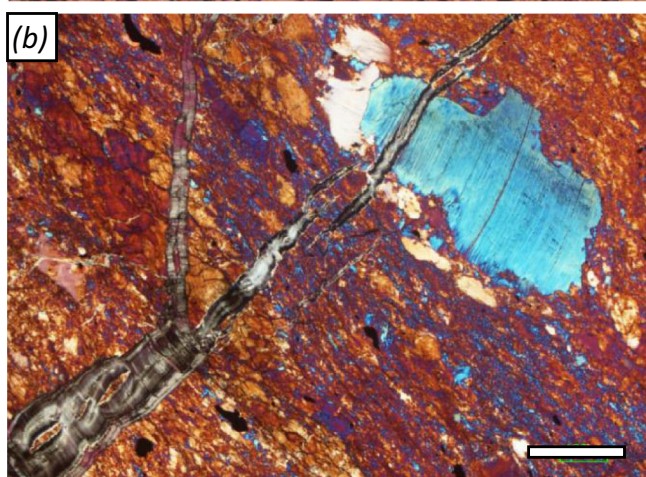

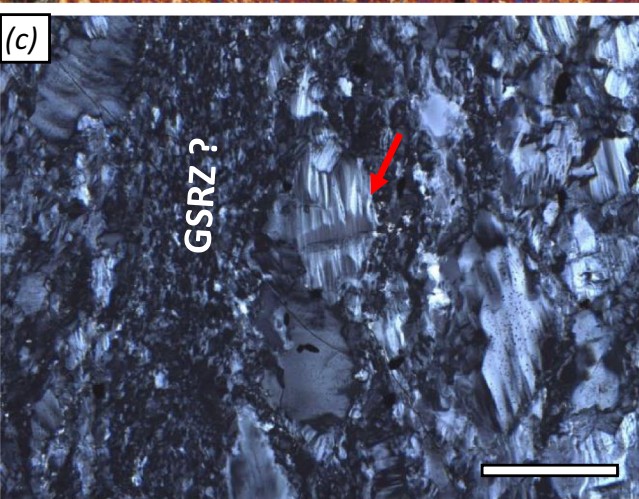

**Fig. 2 Deformation microstructures in serpentinites. a** Foliated serpentinite with crystallographic preferred orientation (CPO) and aligned Fe-oxides tracing former polygonal, now flattened mesh cells (crossed polarized image (xpol) with 1λ-plate; sample OM20-13); **b** Serpentinite mylonite with strong CPO of lizardite and bastite porphyroclast, cut by a younger serpentine vein with anomalous extinction color (xpol with 1λ-plate). **c** Grain size reduction in shear bands (GSRZ), and deformation lamellae and kinking (red arrow in **c**) of larger serpentine grains (xpol). (**b** and **c** are both from sample BT1B_74-1_59-61). Scale bars: 200 μm.

luminescence that is different from the concentric core-rim zoning of magnesite ellipsoids (Fig. 3d). The boudin neck magnesite contains a narrow, Fe-rich zone (dark in the CL image) overgrown by a rim of bright luminescent, Si inclusion-

bearing magnesite (arrows in Fig. 3b, c & d). This rim is similar to the partly dendritic rims that mark the transition between magnesite ellipsoids and interstitial quartz elsewhere in the sample (Fig. 1j). Compositional maps and CL images further reveal that Fe-magnesite cores of matrix grain aggregates occasionally form sigma-clasts (yellow arrows in Fig. 3c, d), with a sense of shear consistent with that of transposed Fe-magnesite veins in the same sample (Supplementary Fig. S9).

**Crystallographic preferred orientations in listvenite.** Listvenites with a SPO of magnesite ellipsoids have a weak but statistically significant CPO of magnesite, with c-axes oriented perpendicular to the magnesite grain elongation direction in thin section (Fig. 4). Poles to a- and m-planes show a weak girdle distribution (Supplementary Figs. S8–S18). This CPO and grain elongation relationship is consistent between samples, and also apparent for fine-grained, dendritic magnesite in the matrix of some foliated listvenites (Fig. 4). In the same samples, quartz locally has a CPO (Fig. 4) with their c-axes parallel to the magnesite SPO, and with weak maxima of poles to a- and m-planes (Supplementary Figs. S8–S18).

**Folding and transposition of early veins.** In many samples with a penetrative foliation, early antitaxial to blocky magnesite veins are folded and/or transposed (Fig. 5). Foliated zones with folded veins in places have sharp transitions to non-foliated listvenite where veins are not folded. In folded listvenite, dendritic magnesite shows a strong SPO approximately parallel to the fold axial planes (Fig. 5a, d) defining an axial planar cleavage. In some fold hinges, quartz shows a CPO with c-axes parallel to the axial planar cleavage (Fig. 5; Supplementary Fig. S16). Fold microstructures show complex cross-cutting relationships, with variable orientations of early veins, and folded and transposed veins overgrown by later, locally euhedral magnesite (Fig. 5a, d). Optical-CL imaging shows that bright pink-luminescent magnesite overgrowths on folded veins are highly irregular in thickness and transition into axial planar dendritic grains (white arrows in Fig. 5b), suggesting that this magnesite formed during or after folding. In contrast, dull-luminescent magnesite in vein centers displays more continuous thickness and sharp contacts with the matrix. These relationships are different around fold hinges, where folded veins coalesce and increase in thickness. Veins commonly have a narrow, bright-luminescent centerline rich in Fe-oxide and/or –hydroxides (Fig. 5b) and compositional zoning that traces the shape of the folds. The folded zonings often have small offsets close to fold hinges (Fig. 5d), although faults in the listvenite matrix are absent. In offsets of transposed veins, dendritic magnesite overgrowths are oriented subparallel to oblique to the matrix foliation (Fig. 5c).

**Low-angle grain boundaries.** Magnesite in folded veins shows abundant low-angle grain boundaries (<10° misorientation) at high angles to the vein margins and subparallel to the axial planar cleavage (Fig. 5e). In the matrix of foliated listvenite, low-angle boundaries are common in ellipsoidal magnesite, and present but less abundant in quartz (Supplementary Fig. S11). Continuous low-angle boundaries that segment grains into subgrains often have traces at high angles to the magnesite SPO, but can also be parallel (Fig. 6a). Radial, discontinuous low-angle boundaries are common in magnesite ellipsoid rims (Supplementary Fig. S14).

Compositional mapping by transmission electron microscopy (TEM) scanning of a low-angle boundary in ellipsoidal magnesite (Fig. 6b) reveals that magnesite is Fe-bearing and contains abundant Si-bearing inclusions (20–150 nm). Along the low-angle boundary, an inclusion-free rim of Fe-poor magnesite is present

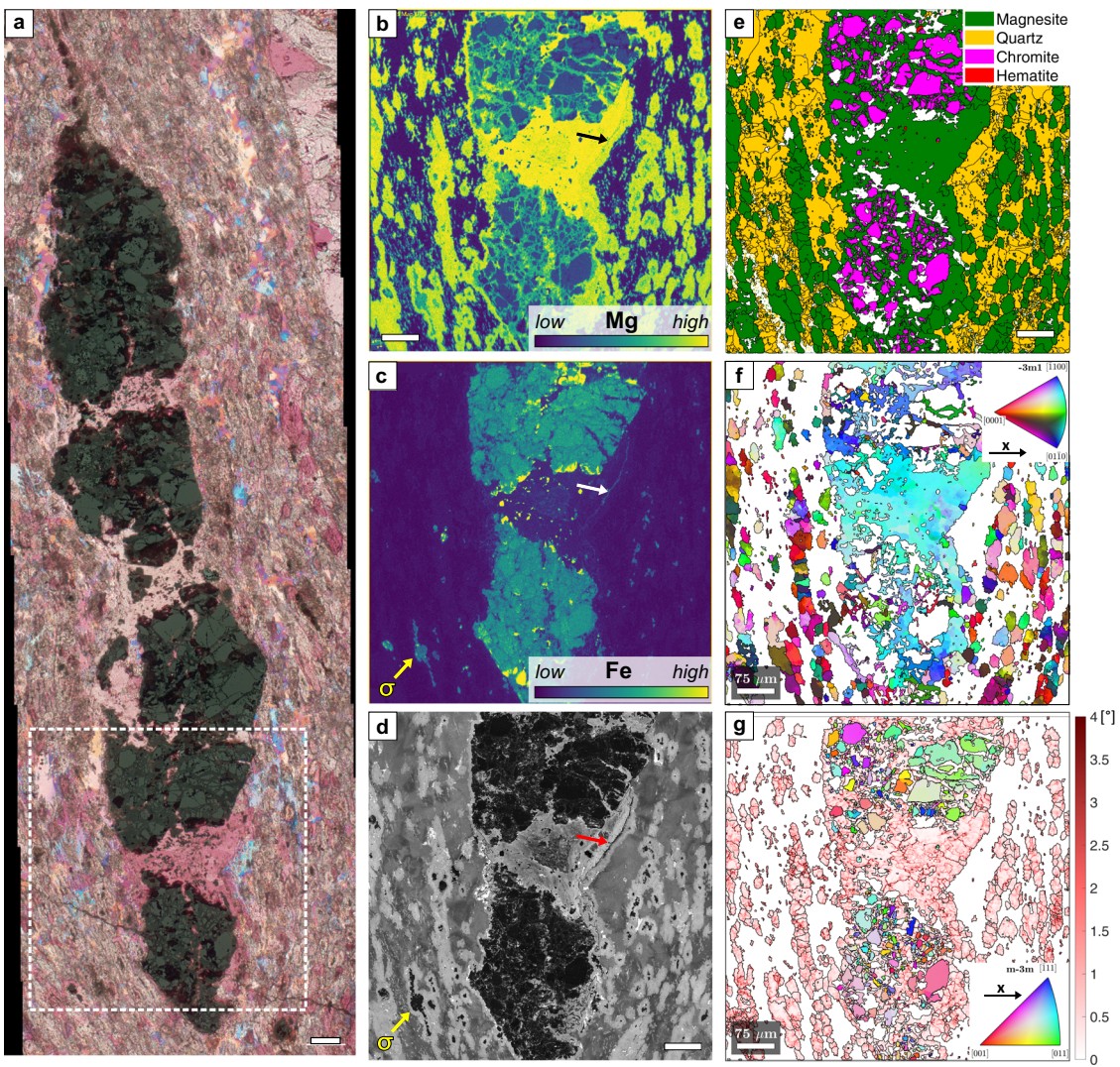

**Fig. 3 Boudinage of Cr-Spinel in foliated listvenite. a** Combined reflected and crossed-polarized light image with 1λ-plate. **b, c** Mg and Fe chemical maps of the area indicated in **a**, showing Fe-rich magnesite between spinel fragments and a Fe-bearing seam at the contact between magnesite in the boudin neck and the quartz-magnesite matrix (white arrow). Fe-magnesite in the core of an ellipsoid forms a sigma-clast in the matrix (yellow arrow, σ). **d** Scanning-electron microscopy cathodoluminescence (SEM-CL) image of the same area showing different magnesite generations and growth zoning in quartz. **e** Phase map from electron backscatter diffraction (EBSD). **f** Crystallographic orientations of magnesite (inverse pole figure (ipf) color scale: see inset). **g** Crystallographic orientation of chromite fragments (ipf color scale see inset) and kernel average misorientation of magnesite, showing low-angle boundaries in red. (Sample BT1B_14-3_77-80). Scale bars: 75 μm.

on both sides of the boundary. A 10–20 nm wide Fe-enriched seam occurs along the comparatively straight and sharp contact between the inclusion-free rims and the host magnesite. The actual crystallographic low-angle boundary (gb* in Fig. 6) is rough on the nm-scale. These observations suggest that the crystallographic misorientation across this low-angle boundary is due to a 500–600 nm wide intra-granular nano-fracture that was sealed by epitaxial precipitation of inclusion-free magnesite onto the walls. In a few places along the boundary, porosity caused by growth misfit is preserved. Despite the significant abundance of low-angle boundaries and misorientations within magnesite, the dislocation density appears to be generally low, with minor dislocations concentrated at low-angle boundaries that lack inclusion-free magnesite precipitates (not shown in the figure).

**Crystal growth microstructures.** Foliated listvenites preserve abundant microstructures related to crystal growth, such as growth zoning in quartz and magnesite, euhedral overgrowths of magnesite

crystals, and dendritic magnesite rims intergrown with quartz. Listvenites also preserve ubiquitous intra-granular nano- and micro-porosity and locally abundant inter-granular microporosity.

SEM-CL imaging reveals concentric zoning in magnesite ellipsoids, locally with euhedral growth zones (Fig. 1j; Fig. 7). The outermost rim of many ellipsoids is dendritic, composed of magnesite-quartz intergrowths. Magnesite dendrites extend into the surrounding quartz (Fig. 7a). In some areas, quartz that surrounds magnesite ellipsoids is massive or featureless under SEM-CL, but locally contains dark-luminescent, rounded domains that do not correspond with grain boundaries. In others, quartz clearly envelops magnesite ellipsoids and shows oscillatory and/or sector growth zoning with euhedral growth facets and remarkable dark-luminescent marker zones that can be correlated across crystals (Fig. 7a). In places, quartz shows botryoidal, concentric growth zoning (Fig. 7b) with spherulitic domains at the transitions into brighter-luminescent zones. CL zoning of quartz indicates hetero-geneous crystallization[47]. Matrix quartz does not cut the CL zoning of magnesite, but is intergrown with the delicate magnesite dendrites

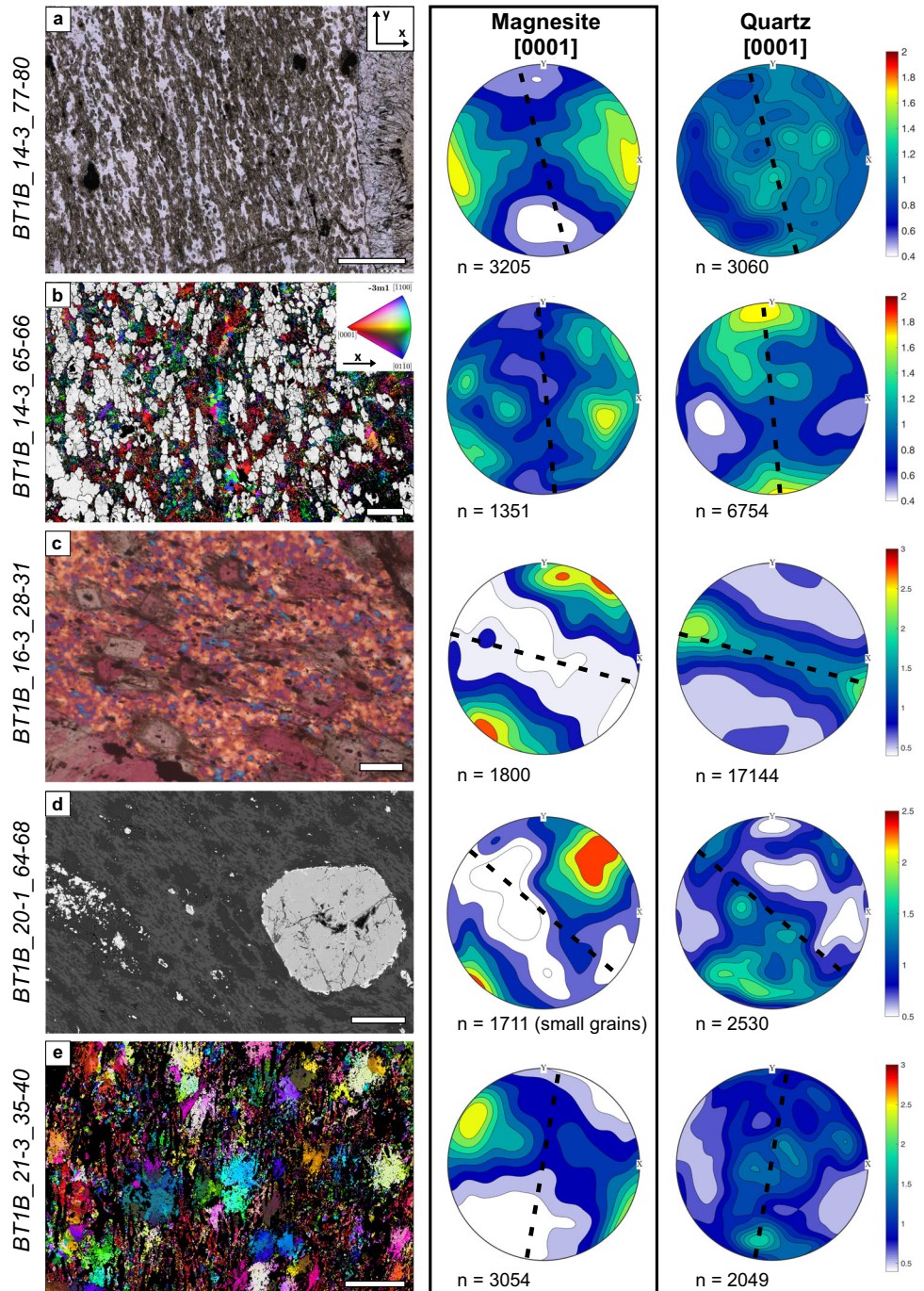

**Fig. 4 Microstructures in listvenites with penetrative foliation and corresponding contoured pole figures of magnesite and quartz c-axes (1 point per grain; lower hemisphere) in thin section reference (x, y coordinates see inset in a).** The black dotted line in pole figures shows the orientation of the foliation trace in thin section based on the elongation direction of magnesite grains; contours are multiples of a random distribution. The orientation and strength of crystallographic preferred orientation of [1014] is not shown, it is similar to [0001]. Pole figures based on all points (not shown) have similar distributions; full pole figures including a- and m- axes see Supplementary figs. S8–S18. **a** plane polarized light (ppol) image; **b** quartz orientations (ipf colorscale see inset), and kernel average misorientation of magnesite grains in grey; **c** crossed polarized light with 1λ-plate; **d** back-scattered electron image; **e** magnesite orientations (colorscale see inset in **b**); black = other phases, not indexed. Scale bar in **a** is 500 μm, in **b**- **e** 100 μm.

at the ellipsoid rims. Similar dendritic magnesite also occurs on straight crystal facets of euhedral magnesite grains (Fig. 7c–f). Crystallographic orientations are commonly the same as the larger grains (Fig. 7d), pointing to epitaxial growth.

Micro-CT indicates that the matrix of foliated listvenites contains ~0.23% preserved porosity, mostly at rims of magnesite ellipsoids (Supplementary Fig. S6; Fig. 1k). Dendritic intergrowths locally contain high inter-granular porosity with sub- to

euhedrally terminated quartz and magnesite (Fig. 7e). Magnesite dendrites and the interstitial quartz locally have abundant intra-granular nano-porosity (Fig. 7e).

## Discussion
Our results provide strong evidence of ductile deformation in serpentinites and during the carbonation reaction from

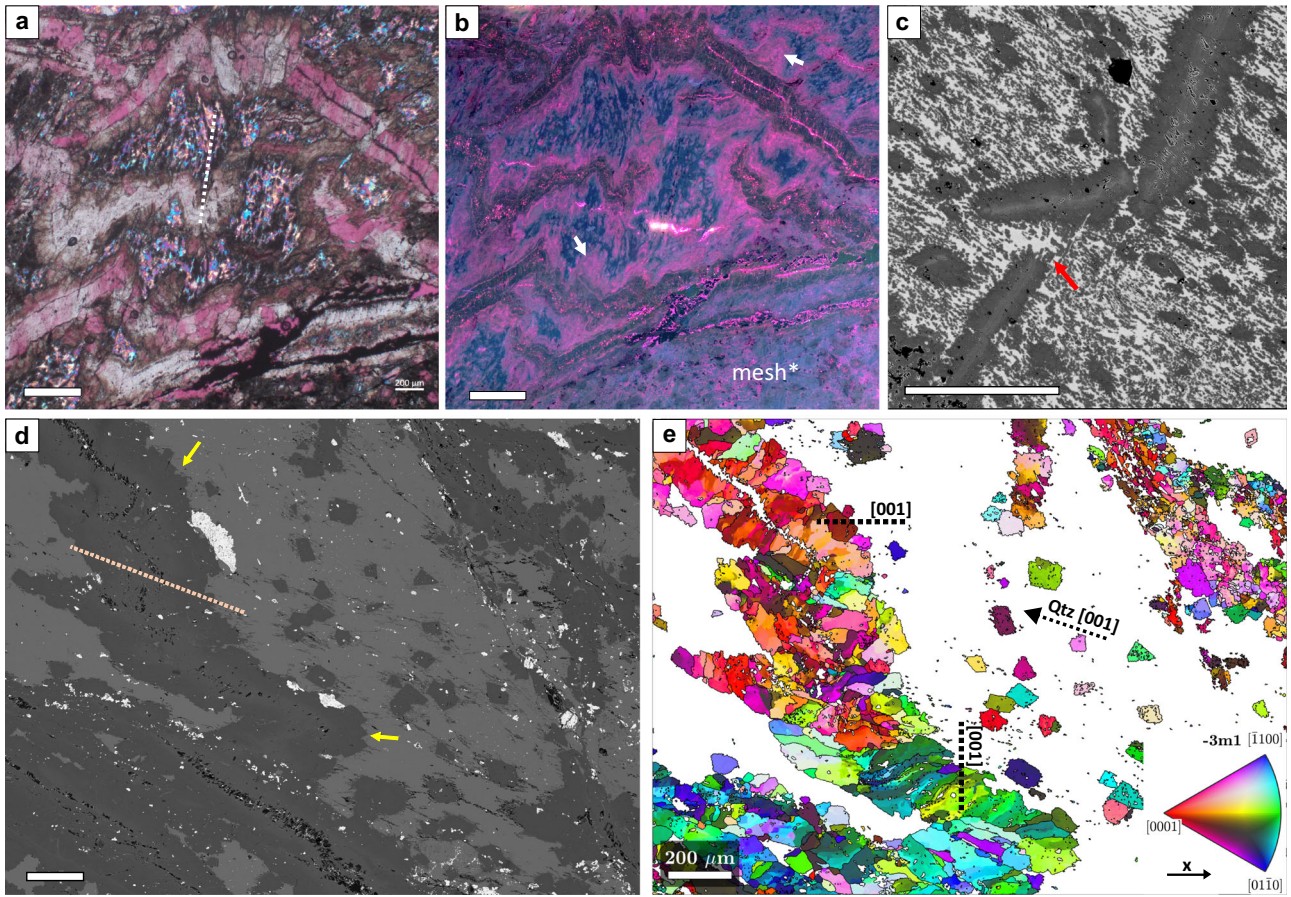

**Fig. 5 Ductile transposition and folding of early magnesite veins. a** folded magnesite veins, the matrix consists of quartz and aligned magnesite dendrites (crossed polarized light with 1λ-plate; BT1B_21-3_35-40). **b** Optical cathodoluminescence image of the same area as in **a**, showing pink luminescent magnesite overgrowth on folded veins (arrows). In the lower right domain listvenite resembles a mesh texture and veins are not folded. **c** Back-scattered electron (BSE) image of transposed magnesite vein with magnesite dendrites growing oblique to the foliation in the opening space (red arrow). In this image, the contrast was enhanced and oxides are rendered black (BT1B_20-1_64-68). **d** BSE image of folded magnesite vein, with euhedral magnesite overgrowths on the vein rims (yellow arrows) (BT1B_16-3_28-31). **e** Electron backscatter diffraction (EBSD) orientation map of magnesite of the same area as in **d** (ipf color scale see inset), with average orientation of magnesite [001] in different vein parts and of quartz (Qtz) [001] in the matrix indicated (c.f. Figure 4). In **a** and **d**, dotted lines mark the trace of the fold axial surfaces and the parallel shape preferred orientation of dendritic magnesite. Scale bar in **a** and **b** is 400 μm, in **c**–**e** 200 μm.

serpentinite to listvenite. We infer a general trend from early ductile deformation to conditions at the brittle-ductile transition (this paper), followed by brittle overprinting of listvenite by cataclasis[42]. Observations in serpentinites of Hole BT1B indicate a similar transition from ductile to brittle deformation over time (Fig. S20).

The earliest preserved deformation microstructures in serpentinite and listvenite are relicts of elongate Cr-spinel and an SPO of pseudomorphs of orthopyroxene (bastites), which suggest that parts of the protolith peridotite had a high-temperature porphyroclastic to mylonitic fabric, a common feature of the "Banded Unit" comprising the lowest ~500 m of the mantle section of the Samail ophiolite[48–50]. This early, pseudomorphic fabric predates serpentinization and carbonation and in places defines a pseudomorphic foliation in listvenite. Serpentinization is inferred to have preceded carbonation, because there are 1–10 m wide, fully hydrated serpentinite zones between listvenite and partially serpentinized peridotite[8], and because Fe-oxides in listvenite commonly trace a former mesh texture that is typical of serpentinization of olivine[32]. However, we do not exclude that direct replacement of olivine or pyroxene by carbonate and quartz may have occurred in places.

Therefore, we infer that the low-temperature reaction sequence in the Oman listvenites was (Fig. 8):

(I) serpentinization of olivine and pyroxene,
(II) incipient carbonate formation in serpentinite ("ophicarbonate"),
(III) continued carbonate growth and local replacement of serpentine by quartz and/or amorphous silica (serpentine-magnesite-quartz disequilibrium assemblages),
(IV) full replacement of remnant serpentine by quartz and dendritic carbonate.

These different reaction stages likely were active in several simultaneous but spatially separated alteration fronts, reflecting increasing water/rock ratios from I to IV and a chemical gradient of decreasing fluid $CO_2$ concentrations from listvenite to carbonate-bearing serpentinite and partially hydrated peridotite. Zoning of Fe and Ca contents in magnesite (Fig. 3b; see also[34]) documents variations in fluid composition and redox conditions between stages II–IV, with reaction of magnetite and Fe-bearing serpentine to Fe-magnesite mostly during carbonation stage II (Fe-enriched magnesite cores; Fig. 1k). An intermediate reaction step forming talc-magnesite assemblages, which is common in many other listvenite occurrences[6,20,51] and predicted by modelling[33,52], is rare in outcrops. In drill core from Hole BT1B, some talc is present in dm–m scale transitions between

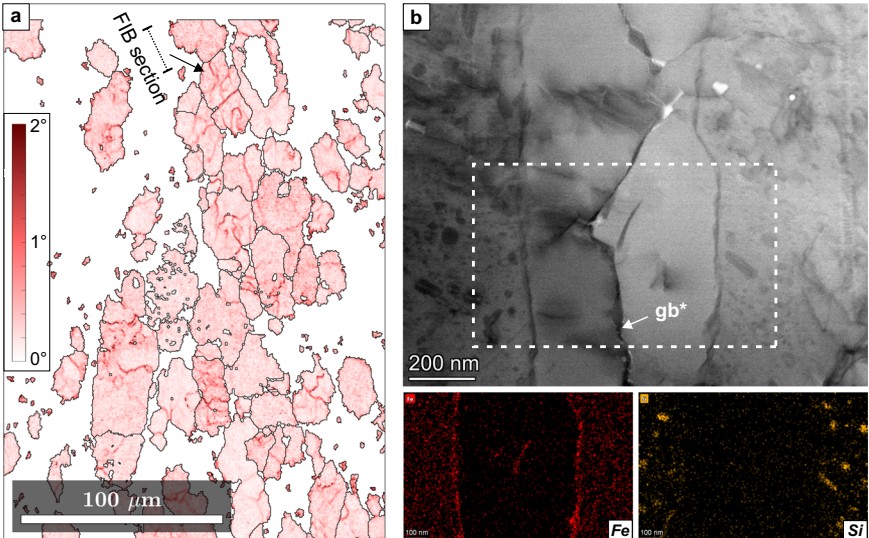

**Fig. 6 Low-angle boundaries in magnesite within foliated listvenite. a** Electron backscatter diffraction (EBSD) kernel average misorientation map of magnesite (3° threshold); the magnesite grain selected for transmission electron microscopy (TEM) analysis and the orientation of the focused ion beam (FIB) section are indicated. **b** Scanning TEM bright-field image of a low-angle boundary (gb*) in the selected magnesite grain, and Fe and Si compositional maps of the framed area. (Sample BT1B_14-3_65-66).

serpentinite and listvenite[32]. This may be attributed to low temperature, a narrow range of water/rock ratios in which talc was stable during carbonation, and/or large disequilibrium of reaction[8,33,34]. Some silica may have initially precipitated as opal[32], similar to low-temperature listvenites elsewhere (e.g.[53,54]), followed by dehydration and recrystallization to quartz or chalcedony.

The SPO and CPO in foliated serpentinite likely formed at P–T conditions similar to those of listvenite formation, as indicated by the presence of flattened mesh textures and the predominance of the low-T serpentine polytype lizardite. Ductile deformation of lizardite by basal glide[45] (Fig. 2) is inferred as coeval to the reaction stages producing the ductile deformation structures in listvenite.

Microstructures in foliated listvenites demonstrate that macroscopically ductile deformation occurred not only in the precursor serpentinite but also during reaction stages II and III (Fig. 8). We infer that the texturally earliest carbonate is Fe-magnesite in the interstices of fragmented Cr-spinel. The spinel fragments are locally rotated (Fig. 3), indicating that the earliest carbonation stage was concomitant with deformation. Formation of early, Fe-bearing magnesite veins followed, showing that fluid flow was initially focused along fractures, with minor precipitation of Fe-magnesite cores of carbonate ellipsoids in the matrix (stage II in Fig. 8).

Deformation of the carbonate-bearing, reacting serpentine matrix led to folding and transposition of the early magnesite veins (Fig. 5) and the development of an axial planar cleavage. Boudinage of Cr-spinel with magnesite precipitation in the necks (Fig. 3) and bending of the foliation around spinel porphyroclasts likely occurred during this phase. Subsequent carbonation under variably supersaturated conditions caused the growth of magnesite ellipsoids with partially euhedral rims in the serpentine matrix. Dendritic magnesite-quartz intergrowths and botryoidal and zoned quartz overgrow all previous (variably deformed) magnesite generations (Fig. 7), suggesting that static crystal growth prevailed in the final reaction step IV.

Thus, we infer that at least part of the carbonation reaction was concomitant with deformation. This poses the question whether external stress and related strain enhances fluid flow and the carbonation reaction progress.

Dislocation creep is commonly inferred to be the most important process forming mineral fabrics. However, crystal plastic deformation is not the only mechanism to form a CPO. Case studies and experiments on mafic rocks indicate that a CPO may also form by preferential crystal growth and dissolution-precipitation creep during metamorphic reactions[17,55–57]. Dissolution-precipitation creep is particularly relevant in the presence of high fluid pressure and in porous, fine-grained polyphase assemblages[17,58,59]. These conditions are prevalent during metamorphic devolatilization reactions and metasomatic fluid-rock interaction. Thus, dissolution-precipitation creep may often be the dominant deformation mechanism, and the main cause for formation of a CPO and substantial transient weakening in reacting assemblages[29,55,60].

Our results point to dissolution-precipitation creep and oriented crystal growth during reaction-assisted, transient weakening of the porous reacting mass as the main cause for the shape and crystallographic preferred orientations in the BT1B listvenites: (i) Crosscutting relationships indicate that the first stages of carbonation were synkinematic (Fig. 3), (ii) distributed deformation was absent during final crystallization of dendritic magnesite rims (Fig. 7), and (iii) dislocation densities are low. Unlike in dislocation creep fabrics, low-angle boundaries in our samples are nano-scale, intragranular fractures filled by precipitated minerals (Fig. 6), or have radial patterns that also occur in non-foliated listvenite, where they have been interpreted as the result of sector zonation or crystal growth competition[34]. This excludes dislocation creep as the main deformation mechanism for magnesite in foliated listvenite. Similarly, the common growth zoning and euhedral crystal terminations (Fig. 7) are evidence against deformation of quartz by dislocation creep.

Observations from flow-through carbonation experiments suggest that the fluid flow rate and permeability structure have a strong influence on the crystallographic orientation of carbonate, with the fast-growing crystallographic directions ([1014] and [0001]) preferentially oriented normal to the fluid flow direction[61]. Based on the SPO and CPO of lizardite in foliated serpentinites adjacent to listvenites at Site BT1, we infer that fluid flow was anisotropic during the initial stages of carbonation, with higher permeability parallel to the foliation. Thus, the CPO of

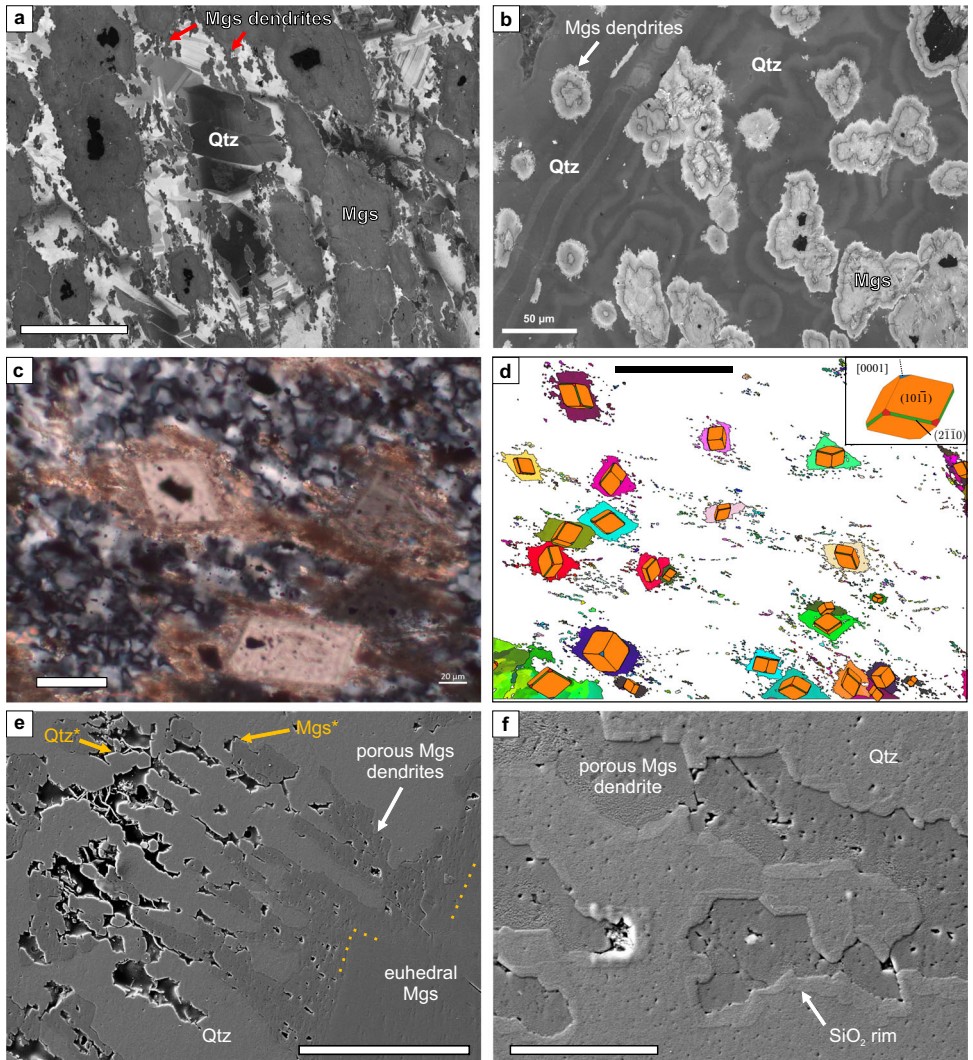

**Fig. 7 Crystal growth microstructures in foliated listvenites. a** Scanning-electron microscopy cathodoluminescence (SEM-CL) image of magnesite (Mgs) ellipsoids with Fe-magnesite (black) cores and aligned magnesite dendrites, in a quartz (Qtz) matrix with euhedral crystal growth zoning (CL filter optimized for quartz). **b** SEM-CL image of magnesite ellipsoids with concentric zoning and dendritic rims, and concentric spherulitic to botryoidal growth zoning of quartz (CL filter optimized for magnesite). **c** Euhedral magnesite with Fe-oxide inclusions and dendritic rims; quartz in the matrix has a crystallographic preferred orientation (c.f. Fig. 4c) (crossed polarized light image). **d** Electron backscatter diffraction (EBSD) orientation map of magnesite (color scale see Fig. 4) with corresponding crystal shapes overlay of rhombohedral magnesite (crystallographic axes see inset). **e** Secondary electron (SE) image of dendritic rim on euhedral magnesite. Sub-micron scale crystal facets of quartz (Qtz*) and magnesite (Mgs*) are visible in the related porosity. **f** High resolution SE image of a nano-porous magnesite dendrite showing a 100−400 nm wide SiO$_2$ rim at the contact to quartz. (a, b: sample BT1B_14-3_77-80; c - f: sample BT1B_16-3_28-31). Scale bars in **a**–**c** are 50 μm, in **d** 200 μm (black bar), in **e** 10 μm and in **f** 2.5 μm.

magnesite in foliated listvenites may be due to preferential growth of matrix magnesite, with [1014] and [0001] normal to fluid flow in the foliation plane. Assuming that the SPO of matrix magnesite in foliated listvenite reflects the orientation of a previous serpentine foliation, the expected preferential growth direction is consistent with the measured CPOs of [0001] and [1014] in magnesite (Fig. 4). The locally observed CPO of quartz may have formed through a similar process of epitaxial, oriented growth. Alternatively, the quartz CPO may be inherited from initially present opal or could have formed during dehydration of opal to quartz.

We infer that the transformation of a serpentinized peridotite precursor to carbonate-bearing serpentinite and listvenite caused changes in rheology due to the changing proportions of olivine/ pyroxene, serpentine, magnesite and quartz (± opal), porosity, and to the different strengths of these minerals. Microstructural analysis of the early carbonate generations suggest that once

formed, magnesite was stronger than the serpentine matrix. This is manifested in preserved euhedral magnesite cores, transposition of magnesite veins in a sheared matrix, and folding of magnesite veins while the (inferred) matrix serpentine formed an axial planar cleavage (Fig. 5). Vein microstructures showing small offsets of the growth zoning and a high abundance of low-angle boundaries oriented subparallel to the fold axial plane suggest that grain boundary sliding in the carbonate veins was the main mechanism accommodating folding, while basal glide of serpentine[44] and dissolution-precipitation accommodated deformation in the reacting matrix.

The complete reaction sequence requires large fluid - rock ratios[33] and, thus, sufficient permeability. Because magnesite has a higher density than serpentine, porosity may be created during the transformation of serpentine to talc + magnesite, in particular at high pressures[62] and if Mg and/or Si is leached by fluids[63]. The reaction of serpentine to magnesite + quartz, however, increases

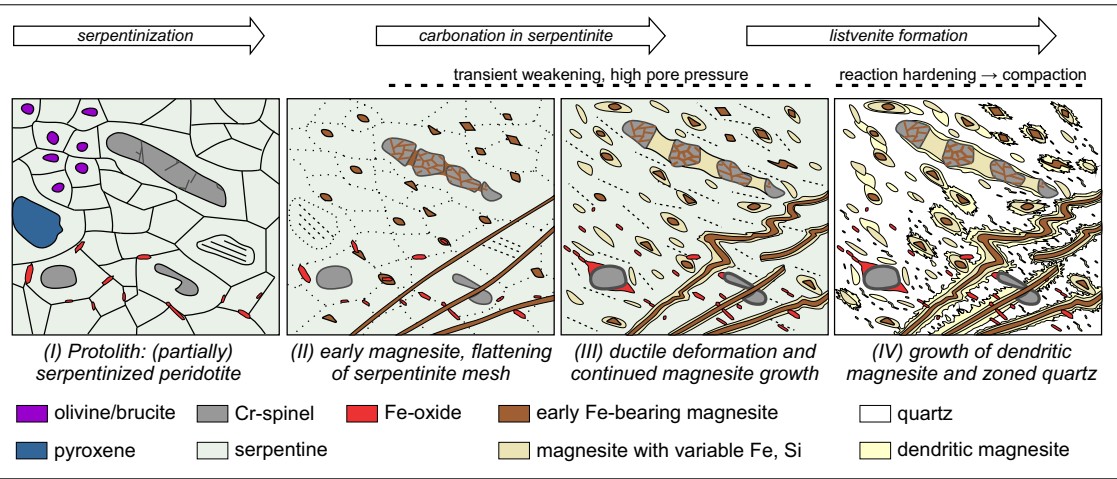

**Fig. 8 Schematic mineral growth and deformation evolution during progressive reaction of serpentinite to listvenite in sheared intervals of Hole BT1B.**
While the sketch illustrates the temporal evolution from reaction stages I–IV, these likely occurred simultaneously across an advancing reaction front, with strain partitioned into the weak, serpentine-bearing and fluid saturated reacting assemblages (Stages II and III). Talc and opal, which may have formed locally during stages II/III and IV, respectively, are omitted in the sketch.

solid volume by 18–20% if non-volatile elements remain immobile[5,33]. Major and trace element geochemistry of the BT1B listvenites indicate no major Mg or Si loss[64], suggesting that another porosity-forming mechanism was necessary. The preserved porosity in the matrix of foliated listvenite (~0.23%) is about one order of magnitude lower than in serpentinite $(2.7 \pm 1.0\%)$[34]. However, locally, inter-granular micro-porosity is abundant in foliated listvenite (Fig. 7e). Together with intra-granular nano-cracks (Fig. 6) and trans-granular fractures now filled by magnesite, these observations point to a dynamically evolving permeability network at lithostatic fluid pressure that allowed pervasive fluid flow and complete carbonation. We infer that lithostatic pore pressures during serpentinite carbonation in turn promoted ductile deformation in the reacting medium, mainly through grain boundary sliding accommodated by dilatant granular flow and dissolution-precipitation.

Locally, listvenite formation could proceed without apparent finite strain; much of core BT1B consists of non-foliated listvenite containing pseudomorphs after mesh and bastite serpentine, enclosing a few bands of non-foliated serpentinite with mesh and bastite textures. In those cases, the pre-existing permeability structure of the serpentinite mesh—deformed in a dilatant fashion under lithostatic pore pressure—may have been the main factor controlling reaction progress. Here we note that although strain in these zones is not apparent, a small, not visible strain (e.g., 10%), can create significant dilatant porosity.

Although volumetrically less abundant, shear zones and early magnesite veins are widespread, and may have acted as conduits for advective fluid flow that also supplied $CO_2$ for the formation of non-foliated listvenite intervals. The strength contrasts between magnesite, quartz/opal, and serpentine minerals, and between serpentinite and listvenite likely played a key role in generating locally high differential stress, and in maintaining a high permeability at the reaction front. The conversion of serpentinite into a polyphase, carbonate-serpentine assemblage at deviatoric stress may enhance pressure solution of serpentine at the interface with stronger magnesite, and produce permeability along the margins of magnesite veins. Carbonate growth at hinges of folded magnesite veins (Fig. 5a, b) may be due to this effect. On a larger scale, we propose that the inferred permeability and strength difference between serpentinite and fully reacted and compacted listvenite caused reactive fluids to accumulate and deformation to focus along the carbonation reaction front. This

may explain why we find 10 s of meters thick, continuous listvenite bands with very few non-reacted serpentinite domains, but also very few quartz-magnesite veins in serpentinite. In contrast, non-reacted serpentinite relicts within the carbonation reaction product are common during the formation of magnesite-talc rocks in other localities[6,20]. This suggests that the strength contrast between talc-magnesite and serpentinite causes a different morphology and permeability profile of the reaction front compared to the direct replacement of serpentinite by listvenite, which is dominated by reaction hardening. Mechanically weaker serpentinite inclusions in a hardening listvenite matrix will preferentially deform and react under tectonic stress. Because the reaction product listvenite is stronger than the serpentine-bearing, fluid-saturated reacting mass, deformation may preferentially partition into the reacting mass, locally enhancing transient fluid flow and, thus, the carbonation reaction progress.

Complete carbonation of partially hydrated peridotite to listvenite at large scale requires (i) a source of large quantities of aqueous, $CO_2$-bearing fluid, (ii) a mechanism that channelizes fluid flow into zones of intense carbonation, and (iii) a process that sustains permeability despite the increase in solid volume during reaction. Our results suggest that semi-brittle to ductile deformation concurrent with chemical reaction enables channelized flow and sustained permeability. A well-suited fluid source are carbonate-bearing sediments rich in hydrous silicates that undergo devolatilization related to heating, for example, due to contact metamorphism or burial during thrusting[6,41,51]. In the Samail ophiolite, burial of the Hawasina nappes and underlying units below the ophiolite caused fluid flux and the formation of overpressure cells[35], which, combined with active deformation along the base of the ophiolite, might have created the circumstances necessary for listvenite formation. Based on (imprecise) geochronological constraints[8,33] and because stable isotope signatures suggest that $CO_2$-bearing fluid has been sourced from deeper, lower-plate sediments[41], listvenite may have formed during ophiolite thrusting – a setting analogous to the "cold nose" of the mantle wedge of subduction zones[4,33]. Carbonation may also have occurred during extensional reactivation of the thrust, in a phase of post-obduction top-to-the-NNE shear[35,40]. The exact timing and tectonic context of listvenite formation at site BT1 remains debated (see supplementary material for further discussion).

More generally, coupling of fluid-rock interaction and deformation as observed in the Samail ophiolite listvenites may be common in a variety of active tectonic settings where

serpentinization and/or carbonation consume fluids and cause volumetric expansion. Among others, these may include transform faults, ophiolite sutures, and the shallow mantle wedge of subduction zones[4]. Under external stress, serpentinization and carbonation reactions are likely to cause cyclic variations of permeability, pore pressure, and differential stress, which may induce repeated fracturing and sealing of veins by serpentine and carbonate. If $CO_2$ fluid concentrations are high, the formation of listvenite may intensify this process due to its higher strength compared to serpentinite, causing dilatancy by granular flow and reaction-assisted ductile deformation along the reacting lithological boundary. Such a process during fluid-rock interaction could explain observed aseismic creep in large-scale shear zones and possibly some subduction zones[65].

We infer that, in a range of tectonic settings, the feedback of external stress, changing rheology, and high pore pressure facilitates continued fluid flow and reaction to listvenite in shear zones despite the increase in solid volume due to $CO_2$ addition. We propose that similar mechanisms were active in most listvenite occurrences worldwide, and that reaction-assisted ductile deformation during fluid-rock interaction may be common in subduction zones and other fluid-rich fault zones.

## Methods

**Samples**. Samples were collected onboard R/V Chikyu in September 2017 during the Oman drilling Phase 1 core logging and during a field campaign in January 2020, covering the broad and diverse range of (micro)structures in serpentinites and listvenites[32,33]. Field and core samples have been collected and exported in a responsible manner and in accordance with relevant permits and local laws of the Sultanate of Oman. Hole BT1B (International Continental Drilling Project Expedition 5057-4B) was drilled with an inclination of 75°. Therefore, and because the drill cores contain discontinuities across which the orientation of single-core sections could not be restored[32], orientations of structures are unfortunately not easily comparable between different parts of the core and the field. Thus, orientations and vergence of folds, apparent s-c fabrics, and shear senses from sigma clasts cannot be unambiguously correlated and attributed to specific state of stress and/or tectonic settings. After detailed inspection of the core of Hole BT1B, and study of 115 thin sections of selected representative samples, we selected a set of 15 listvenite and 6 serpentinite thin sections lacking late overprinting and containing representative ductile deformation structures for detailed investigation (Fig. 1; Supplementary Table S1). Because the penetrative foliation is usually not clearly visible macroscopically, it was not practical to prepare thin sections in the standard structural reference frame. Thus, shear sense indicators like spinel or carbonate sigma-clasts are only well visible in thin sections where the arbitrary core reference frame (CRF) used to cut samples[66] is coincidently oriented with the section perpendicular to foliation and parallel to a lineation or transport direction. Such features are therefore likely more common than observed.

**Optical and scanning electron microscopy (SEM)**. Thin sections were scanned in plane-polarized light, reflected light, and at 10 different crossed polarizer orientations with a 10x objective using a PetroScan Virtual Microscope. The PetroScan system is a high-end polarization microscope equipped with a camera and automated sample stage, developed by RWTH Aachen University and Fraunhofer Institute for Applied Information Technology (FIT). During image post-processing, the extinction behavior of each pixel was extracted and interpolated to visualize the extinction behavior at all polarization angles. The high-resolution digital mosaics were used as a reference layer for images acquired by optical cathodoluminescence (CL) and scanning electron microscopy (SEM), and for image analysis using ImageJ software. A selection of digitized thin sections analyzed in this study are available as supplementary material to the Proceedings of the Oman Drilling Project (http://publications.iodp.org/other/Oman/SUPP_MAT/index.html#SUPP_MAT_Z).

CL can reveal textures that are not visible using any other imaging method. Variations in CL are caused by natural defects in mineral crystal lattices (vacancies, dislocations) as well as changes in the presence and concentration of trace element and rare-earth element activators[47,67]. In the case of magnesite, CL is mainly controlled by Mn and Fe contents; $Mn^{2+}$ activates luminescence, whereas Fe acts as a quencher so that magnesite with high Fe (>7.5 mol% $FeCO_3$) is non-luminescent[68]. Variations in the concentration of Fe, Mn, and trace elements thus can cause variations in the luminescence intensity and color, making CL a useful tool to track the evolution of crystal growth recorded in single crystals and crystal aggregates[47]. In quartz, luminescence depends mainly on structural defects in the crystal lattice and minor substitution of tetrahedral silica by $AlO_4M^+$ [69]. Because of the potential of this method to reveal key microtextures, we used two complementary modes of CL imaging for this study. Optical mosaic panorama

images of large thin section areas were obtained with a Zeiss Axio Scope optical microscope equipped with a "cold" cathode luminoscope CL8200 MK5-2 operating at 15 kV, 320–350 μA. Single images were taken with a 10x objective and exposure times of 10 s. Panchromatic and blue-filtered SEM-CL images were acquired using a Zeiss Sigma High Vacuum field emission (FE) scanning electron microscope (SEM) equipped with a Gatan MonoCL4 system at the University of Texas at Austin. Carbon-coated samples were imaged at accelerating voltages of 5 kV, 120 μm aperture, 125 μs dwell time, and 2048 ×2048 pixel resolutions at magnifications up to 2500x following the guidelines of Ukar and Laubach[70].

For phase identification and imaging of chemical zoning, back-scattered electron (BSE) and energy-dispersive X-ray spectroscopy (EDS) large-area maps were acquired with the Zeiss Sigma as well as a Zeiss Gemini SUPRA 55 field-emission electron microscope at the Institute of Tectonics and Geodynamics of RWTH Aachen University. Whole thin sections and areas of interest were mapped with dwell times of 0.2–1.5 ms/point at 15 kV and 8.5 mm working distance. High-resolution secondary electron (SE) images were acquired at 3 kV, 5 mm working distance and 20.000–30.000x magnification. For conductivity, all samples were coated with a 6–8 nm thick layer of tungsten.

Electron backscatter diffraction (EBSD) maps were acquired on areas of interest in thin sections (up to 5 mm²) using a Zeiss Gemini SEM 300 instrument equipped with an Oxford Symmetry EBSD system at the Central Facility for Electron Microscopy, RWTH Aachen University. Analyses were carried out under variable pressure conditions using $N_2$ at 30 Pa on samples that were tilted 70° at working distances of c. 10 mm, using an accelerating voltage of 20 kV, probe currents of approx. 18 nA, and 0.5–3 μm step sizes. Data were indexed with Aztec analytical software using the ICSD reference database. Post-processing with Oxford Instruments HKL Channel 5 software included the removal of wild spikes, successive filling of non-indexed pixels according to 8, 7, and 6 neighboring pixel orientations, and the correction of non-systematic misindexation between dolomite and magnesite based on simultaneously acquired EDS data. The Matlab-toolbox MTEX (version 5.3.1)[71] was used for grain boundary modelling (10° segmentation angle), small grains removal (10 pixel threshold), calculation of orientation distribution functions, and for plotting orientation maps and pole figures. Kernel average misorientation maps were calculated with a first-order kernel of neighboring pixels in a square. Because thin sections were not prepared in the standard structural reference frame, orientation maps and pole figures are plotted in the arbitrary spatial reference frame of the individual measurement areas within the thin sections. Thin section orientations relative to the core reference frame are given in the supplementary figures.

**X-Ray micro-tomography (micro-CT)**. A micro-tomography scan of a foliated listvenite (sample BT1B_14-3_65-66) was acquired from a volume in a 2 × 2 x 13 mm prism oriented in the core reference frame, using a X-Ray Microscope Zeiss Xradia Versa 520 at the MAPEX Center for Materials and Processes, University of Bremen. The micro-CT scan was obtained at 1.3 μm voxel resolution in propagation phase contrast mode, which allows the distinction of quartz and magnesite despite their similar X-ray attenuation. Measurements without propagation phase-contrast yielded too low attenuation contrasts between magnesite and quartz. Because this method enhances the contrast at phase boundaries, classical segmentation based on the X-ray attenuation alone could only be applied for pore segmentation, but not for distinction of magnesite and quartz. Here we used the trainable Weka segmentation 3D machine learning algorithm of ImageJ[72] for phase segmentation in subvolumes of the micro-CT data. The FastRandomForest classifier was applied using the original image and mean, variance, edges, and derivatives filters (maximum sigma 8) as training features. The classifier training was repeated once after manual adjustment of classes. This approach produced a reasonable segmentation of quartz-magnesite phase boundaries, but interiors of larger grains were not segmented well. Original and segmented volume renderings are provided in Supplementary fig. S6.

**Transmission electron microscopy (TEM, STEM)**. To gain insight into the nature of low-angle boundaries in matrix magnesite and their possible relation to the observed deformation microstructures and CPOs, we prepared several 80–100 nm thin TEM lamellae by FIB milling from selected magnesite grains along different crystallographic orientations and across low-angle boundaries (Supplementary Fig. S19). The electron transparent specimen preparation for TEM studies were carried out using a dual beam scanning electron microscope (Thermo Fisher Helios 400) equipped with a focused Ga ion beam system. A carbon protective layer was used to protect the specimen from ion sputtering at 30 and 5 kV acceleration voltages. The TEM lamellae were attached to a standard Omniprobe support grid made of Cu. Conventional imaging and electron diffraction studies were carried out using a standard transmission electron microscope (Thermo Fisher Tecnai G2) operated at 200 kV. Chemical composition sensitive scanning TEM (STEM) imaging and measurements were obtained using an electron probe aberration-corrected transmission electron microscope (Thermo Fisher Titan 80-200) operated at 200 kV and equipped with an in-column energy dispersive X-ray spectrometry (EDS) detectors. Spectrum imaging using STEM and EDS signals was collected and processed using Velox software (Thermo Fisher). Specimens were aligned and controlled using double tilt TEM holders.

## Data availability

Further data that supports the findings of this study is available in the Supplementary files. Filtered EBSD datasets generated during and analyzed during the current study are available on the data repository pangaea.de (https://doi.pangaea.de/10.1594/PANGAEA.944786).

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

## Acknowledgements
We thank Michael Kettermann and Yumiko Harigane for sampling onboard Chikyu. Werner Kraus and Jonatan Schmidt are thanked for thin section preparation and technical assistance, and we thank Wolf Achim Kahl for conducting the micro-CT measurements. We are grateful to the Oman Public Authority of Mining for support to conduct fieldwork and sample export. MDM and JLU acknowledge funding by the German Research Foundation (DFG grant UR 64/20-1). This research used samples and data provided by the Oman Drilling Project. The Oman Drilling Project (OmanDP) has been possible through co-mingled funds from the International Continental Scientific Drilling Project (ICDP), the Sloan Foundation—Deep Carbon Observatory (Grant 2014-3-01), the US National Science Foundation (NSF-EAR-1516300), NASA —Astrobiology Institute (NNA15BB02A), the German Research Foundation (DFG: KO 1723/21-1), the Japanese Society for the Promotion of Science (JSPS no:16H06347; and KAKENHI 16H02742), the European Research Council (Adv: no.669972), the Swiss National Science Foundation (SNF:20FI21_163073), the Japanese Marine Science and Technology Center (JAMSTEC), the International Ocean Discovery Program (IODP), and contributions from the Sultanate of Oman Ministry of Regional Municipalities and Water Resources, the Oman Public Authority of Mining, Sultan Qaboos University, CRNS-Univ. Montpellier II, Columbia University of New York, and the University of Southampton.

## Author contributions
M.D.M. and J.L.U. designed the study; J.LU. and P.K. were involved in sampling; M.D.M., J.L.U., and P.K. conducted microstructural analysis by optical microscopy; M.D.M. performed SEM imaging, EDS mapping, optical CL analysis, and image, and micro-CT processing; E.U. conducted SEM and SEM-CL analysis; A.S. and M.D.M. conducted EBSD analysis and data treatment; A.K. and L.K. performed FIB preparation, and TEM and STEM analysis. M.D.M., J.L.U., P.K., G.H. and E.U. were involved in extensive discussion and manuscript writing. All authors contributed to the interpretation of data and the manuscript text.

## Funding

## Competing interests
The authors declare no competing interests.
