## [Peer Review File · Nature Communications]

REVIEWER COMMENTS

Reviewer #1 (Remarks to the Author):

The authors provide a technically well-written manuscript. I have no doubts about the quality of their presented microstructural/analytical data from a core of ultramafic rocks from the Oman Mountains (except for the complete lack of orientation information). My main critic concern is that the authors do not present the link between their microstructural work and their conclusion that carbonation of the rocks ensued within a subduction zone. I have commented on this in more detail in the letter to the authors.

In the literature, are currently two different models of listwaenite formation/carbonation of ultramafic rocks in Oman. The authors only focus on one model (subduction zone), while the other model (extension) is not even mentioned in the submitted manuscript. The authors are aware that the two clashing models exist. The 'subduction zone model' was proposed by one of the co-authors of the submitted manuscript in 2015. Since then, he and his co-workers from the Oman Drilling Project are pushing this model forward without presenting any geological/structural evidence (including this submitted manuscript). The submitted manuscript represents another attempt to push their model in a one-sided way. The way they try to establish their model is unbalanced, unprofessional and does not foster progress in science. Thus, I recommend rejection of the manuscript.

Best regards,
Andreas Scharf

Reviewer #2 (Remarks to the Author):

I've thoroughly enjoyed reading the manuscript titled "Ductile deformation during carbonation of serpentinized peridotite" by Menzel, Urai, Ukar, Hirth, Schwedt, Kovacs, Kibkalo and Kelemen. A detailed petrographic description of deformed listwaenites from the Oman ophiolite is presented, and the authors argue that deformation assisted carbonation. I think this paper would make an excellent contribution to Nature Communications after some revisions.

(1) The authors present a temporal evolution (see discussion, summarised in Fig 9) from peridotite hydration to carbonation associated with deformation. The argumentation and reasoning for this model is consistent with their observations. However, alternative explanations should be considered, particularly for their step 1. For instance, the reported sequence of listwaenite, talc-magnesite-zones, serpentinites and partially hydrated peridotites could also present a spatial variation in X_{CO_2} and fluid/rock ratio rather than a temporal evolution. This alternative explanation should be discussed in the manuscript.

(2) The petrographic descriptions of the studied rocks are very nice and detailed. The authors also describe chemical variations (e.g. zoning in Fe content in magnesite and CL-zonation in quartz) but unfortunately do not discuss the causes of this zonation. For instance, early formed magnesite are enriched in Fe and associated with Cr-spinel. This may imply that either Fe is contributed by the partial dissolution of Cr-spinel or the various magnesite compositions may reflect variations in the fluid composition.

(3) I am also missing a good discussion about pore space formation associated with the carbonation reaction. Experimental studies in the context of mineral carbonation of antigorite and serpentinites show a weak carbonation rate, which is taken by the authors to support their argumentation that tectonic deformation is essential in large scale carbonation. Although I have no doubts on their interpretation, the authors should consider that carbonation is reported to be efficient in the replacement of serpentinites at higher pressures and large fluid/rock ratios and that the formation of pore space has been reported in such environments (Scambelluri et al.,

2016; Piccoli et al., 2016, 2016; Sieber et al., 2020). This is particularly relevant since the pressure conditions of listvenite formation are poorly constrained for the Oman ophiolite with estimates ranging up to 1.2 GPa.

Additionally, I have a couple of minor suggestions and remarks, which are outlined below.

Line 37-38: Even for unusually high CO₂ contents in aqueous fluids (on the order of 1 wt%), this amount of carbonation requires time-integrated fluid rock ratios > 30.

Reference missing – please add.

Line 80-81: and a nearly complete lack of talc suggest temperatures of 80 – 150 °C during listvenite formation.

Temperature is not the only explanation for an absence of talc, could also reflect elevated XCO₂-fluid.

Line 81-82: Low temperatures are supported by intergrown hematite and graphite or amorphous carbon, which require < 200 °C to coexist in equilibrium

What observations support equilibrium?

Line 128-131: Please refer to figures showing those two different textures in the listvenite matrix. This will make it easier for the reader to follow your description.

Line 135-136: These foliated listvenites constitute ~10 % (13/115) of the studied listvenite thin sections...

I don't see why this is relevant information. According to your core-logging the penetrative-foliated listvenite comprises around 25% of HoleBT1B.

Line 142: silica inclusions

Just silica, not amorphous or cryptocrystalline SiO₂?

Line 207 and Figure 6: The abbreviation for low-angle boundary is not consistent.

Line 236-237: Instead of referring to a BSE image, please add the image of the micro CT showing the porosity.

Line 257-260: Serpentinization likely preceded carbonation, because there are 1 – 10 m wide, fully hydrated serpentinite zones between listvenite and partially serpentinized peridotite¹⁸, and because Fe-oxides in listvenite commonly trace a former mesh texture that is typical of serpentinization of olivine

I am not convinced that serpentinization must have been completed before carbonation initiated. Please evaluate other potential explanations for the listvenite-serpentinite-partially serpentinized peridotite sequence such as a chemical gradient (decreasing XCO₂ from listenite to partially hydrated peridotite) and decreasing fluid/rock ratio (see Sieber et al., JP, 2020). When a CO₂-bearing aqueous fluid infiltrates into a partially serpentinized peridotite, listvenite might be formed in the centre of the vein, but when the fluid migrates lateral from the injection into the surrounding peridotite, XCO₂ will decrease as carbonates are formed and the fluid/rock-ratio declines which may lead to a pronounced hydration. The described occurrence of talc in m scale transitions between listvenite and serpentinite (line 272-275) may support a variation in the chemical gradient rather than a temporal evolution.

Line 297: Subsequent carbonation under variably supersaturated conditions caused the growth of magnesite ellipsoids with partially euhedral rims in the serpentine matrix.

What do the authors understand by 'variable supersaturation'? In a dissolution-precipitation process (as described by A. Putnis and Co-workers) the stable secondary phase (=magnesite) is supersaturated in the interacting fluid-volume. But what is variable in this context?

Figure 8: What is the difference between the solid lines and the dotted lines?

Figure S13: (a) Are you sure this is a BSE image, not from a light microscope?; (d) Please add labels for Cr-spinel and hematite into the image.

**Best wishes,
Melanie J. Sieber**

Reviewer #3 (Remarks to the Author):

This study aims at constraining the relation between serpentinite carbonation and deformation that affected the Oman ophiolite. Considering the great value of the samples which were collected in cores from the Oman Drilling Project, and the potential high interest of the scientific community that works on carbon sequestration, I believe that the present paper certainly fits the scope of the journal.

Overall comments

First of all, I have to say that, as a non-specialist of the study of microstructures and EBSD facilities, I was impressed by the quality of this paper. In a general point of view, the paper is well-written, geological and mineral descriptions are very detailed and clear, and the figures are highly relevant to answer questions asked by this study (except for one, see below in the specific comments).

My main concern is about the last paragraph of the discussion where the authors assume that the reaction-assisted ductile deformation may be common in subduction zones and could be at the origin of aseismic creep in some regions. Although I understand that the authors need a "punchy" end for their paper (and I have to admit that this may be plausible), I found this assumption highly speculative and additional arguments are required to be written as it.

On the whole, I think that this paper is suitable for publication in Nature Communications after minor revisions. Additional minor comments and suggestions are given below.

Specific comments

Results

- lines 92: The reference 33, i.e., Kelemen PB, et al. Mass transfer into the leading edge of the mantle wedge: Initial Results from Oman Drilling Project Hole BT1B. is currently in review and not published yet. I found nothing against this in the author guidelines, but it's always disturbing having references that are not accessible at the time of publication.
- lines 102: Citation 39 should be replaced by a more appropriate one: mesh and bastite textures well as serpentine veins are obviously typical of hydrated mantle peridotite, but many "older" references had documented this in the literature prior the one you cite.
- lines 109: I do not believe that EBSD is useful to discriminate between serpentine varieties...

Discussion

- lines 414-419: This assumption is highly speculative and should be, if possible, slightly developed and better argued to be considered.
- Figure 8: This figure should be removed (or replaced by one of the supplementary material): it is not clear, too "heavy" likely because there is too much text inside. In addition, I found that the information provided by this figure is redundant with Figure 9, which perfectly illustrates the processes you described and which I consider much clearer

Reply to reviewers comments

(Author replies in blue)

Reviewer #1 (Andreas Scharf)

The authors produced a technically well-written manuscript, which describes in detail different microstructures, microcrystallographic orientations and microanalytical analysis of carbonated ultramafic rocks from the well BT-1 of the Oman Mountains. Deformation is partly ductile. Finally, the authors link the observed data and fluid flow to a feedback mechanism in subduction zones.

The microstructural and analytical results of the submitted manuscript are sound. Although I am not an expert in microstructural analysis, the presented results are comprehensible.

We thank the reviewer, who notes that he is not an expert in microstructural analysis, for his comments and are glad to see that he agrees with the main conclusions that we draw from our results, namely that there was fluid-assisted ductile deformation during the carbonation reaction and that this deformation in turn helped to facilitate fluid flux and reaction progress.

My main critical concern is that the authors link their data to a feedback mechanism in subduction zones as the authors fail to establish that the subduction zone environment is applicable. The link between the author's data and the 'subduction zone fluid flow' is neither supported by their data nor by other data (i.e., structural data from the core or surrounding field area – none of them where presented or discussed).

We thank the reviewer for his clear comments and have rewritten the manuscript to make clear that the deformation-assisted carbonation we discuss here derives directly from the observations and data presented in this paper, that it can operate in a range of different tectonic settings, and that, based on the currently available data from recent and past studies, a subduction-obduction setting is in our opinion the most likely configuration of listwaenite formation at site BT1. For the latter, we now provide an explicit and consistent reasoning in the manuscript; please find further details also in the following replies.

The authors have to keep in mind that two contradicting models for the listwaenite (see Jackson, 2005 for spelling of listwaenite) formation in Oman exist. Model one was proposed by Falk and Kelemen (2015). This model proposes that the listwaenite formed within the 'leading edge of a subduction zone' at ca. 95 Ma, thus, during obduction of the famous Semail Ophiolite. Falk and Kelemen (2015) proposed this model in part based on a single Rb-Sr isochron age (97 ± 29 Ma) of a Cr-bearing fuchsite from the listwaenite (Note the great error margin, which could represent various processes, for example post-obductional extension!). I am referring to this in the detailed comments below. The "subduction model" is not supported by field and structural data of the Fanja area.

An older model for the origin of the listwaenite is proposed from Stanger (1985), Wilde et al. (2002), Nasir et al. (2006), and was confirmed by Mattern and Scharf (2018) and Scharf et al. (2019, 2020). This model suggests that the listwaenite formed during extension after the emplacement of the ophiolite. Thus, has absolutely nothing to do with CO₂-fluid transfer within a subduction zone. Some of the authors must be aware of the contradicting models, because the models were discussed at the Oman Drilling Project and Related Research Conference from the 12-14th January, 2020 at the Sultan Qaboos University, Muscat (Sultanate of Oman). Some of the authors of the submitted manuscript attended this conference.

A major flaw of the submitted manuscript is that the authors only explain the 'subduction model' without even mentioning the obvious alternative model (the elephant in the room). This raises my concern as to how serious this manuscript was prepared by uncritically leaving out a major aspect to be considered.

We understand this comment and have expanded the Geological Background (Line 89) and Discussion (Line 389) to include the hypothesis that listvenite formation was related to extensional tectonics after ophiolite emplacement. We added relevant references.

We would like to point out that a structural / geodynamic reconstruction of the Fanjah area or constraining the absolute timing of listvenite formation is beyond the scope of this paper, and due to limitations in text length we cannot discuss this in great depth. For a lengthier discussion, the reader is referred to a number of the papers cited in our manuscript. Nevertheless, we would like to point out a few things here regarding this comment:

The timing and geodynamic setting of listvenite formation proposed in older papers is, while valid hypotheses, not supported by data at the time of publication of those papers and inconsistent with newer data. For example, meteoric- or seawater-driven carbonation to form listvenites, as proposed by Wilde et al. 2002 and Stanger 1985, is inconsistent with geochemical and isotope data from Hole BT1B (e.g. Falk & Kelemen, 2015; de Obeso et al., preprint). Likewise, the Tertiary high heat flow at 36 – 40 Ma as proposed for listvenite formation by Wilde et. al. (2002), and which also Nasir et al. 2006 discuss as a viable scenario, is neither in agreement with the Rb/Sr age of Falk & Kelemen 2015 (even when taking into account the large error), nor is it consistent with the 55 – 60 Ma U/Pb date of two dolomite or calcite veins cutting listvenite (Scharf et al, 2020), which places listvenite formation itself to before that age. (Nota Bene: we only find conference abstracts reporting these U/Pb dates, so we unfortunately cannot evaluate and cite these). It is therefore puzzling why Reviewer 1 writes that Mattern & Scharf (2018) and Scharf et al. confirmed those previous studies: we cannot find this confirmation in Mattern & Scharf (2018).

With respect to the ages, we would also like to point out that post-obduction extension at 60 Ma, while falling within 2 sigma of the 97 Ma (± 29 Ma 2 sigma) internal Rb-Sr isochron from Falk & Kelemen (2015) and thus not impossible, is much less likely than the timing of subduction and obduction (97 – 74 Ma; e.g., Searle & Cox, 2002), which is well within 1 sigma.

Furthermore, Geologic mapping of the outcrops north and northeast of Site BT1 shows that listvenites are gently folded in a broad antiform. In this structure, they are overlain by banded peridotites and dunites – the Banded Unit – with distinctive major and trace element compositions characteristic of the base of the Samail ophiolite mantle section, and underlain by the metamorphic sole, which in turn overlies highly deformed sediments of the Hawasina Formation (see Kelemen et al., preprint available for reading and commenting on ESSOAR). These structural characteristics indicate that the listvenites formed within or just above a shallowly dipping fault zone along the interface between the ophiolite, the metamorphic sole, and meta-sediments beneath the ophiolite. This geometry is inconsistent with listvenite formation along steep normal / strike-slip faults associated with late extension, as proposed by Scharf et al. Moreover, Menzel et al. (2020, JGR) have shown that high-angle normal and strike-slip faulting is common in listvenite, but occurred later than the carbonation reaction. An extensional setting that we indeed cannot fully exclude is CO₂-fluid flow during potential reactivation of the basal thrust as extensional, low-angle detachments. However, to date there is no evidence for this from the listvenites. We note further that while the basal thrust of the Oman Ophiolite accommodated > 100 km contractional displacement, the displacement related to a (possible but to date not well documented) extensional reactivation of the thrust is much less.

For all these reasons, we are of the opinion that a subduction/obduction setting for listvenite formation is most probable, and most consistent with the currently available data. We modified the discussion (lines 377 – 395) to clarify this (although, due to text length limitations, in a shorter way). For further discussion (since this is not the main topic of this paper), we would like to refer the reviewer to a recent paper by Kelemen et al. (preprint available for reading and commenting on ESSOAR) where this is discussed in more depth.

In our opinion our study also provides an important step: we clearly defined which microstructures are synchronous with listvenite formation as opposed to later overprints, creating a framework for future attempts of structural reconstruction or dating. Our results further underline that careful consideration of the complex (micro)structural evolution is key to obtain meaningful and reliable results during such reconstructions.

The authors interpret their data in a complex and speculative way, just to fit the ‘subduction zone model’, while the alternative model (extension) could easily explain their findings.

We think that the discussed processes and proposed mechanisms, which are based on our microstructural observations, are applicable to listvenite formation regardless of whether that occurred in a contractional or extensional tectonic setting. Where discussing the geodynamic context and implications for subduction zones (Lines 377 - 404), we provide now a clearer, explicit statement of why we consider a thrust setting more likely and better supported.

Here is what they wrote:

“Because of the volume increase of serpentinization, under external stress this is likely to cause cyclic variations in permeability, pore pressure and differential stress, which may induce fracturing and the formation of serpentine and early carbonate veins. The formation of listvenite may intensify this process due to its lower permeability and higher strength compared to serpentinite, causing dilatancy by granular flow and reaction-assisted ductile deformation along the reacting lithological boundary. We speculate that this feedback of external stress, changing rheology and high pore pressure helps to facilitate continued reaction to listvenite despite volume increase, as long as CO₂ supply is sufficiently high.”

Why such a complex speculation about changing rheology, high pore pressure and sufficient CO₂ supply? Massive and ongoing solid-state volume increase reaction of ultramafic rocks to serpentinite and listwaenite can be easily explained during an extensional event where fractures open, allowing for fluid transfer. This kind of extension would not occur within a subduction zone.

Here we would like to respectfully disagree with the reviewer. At the depths considered for Listvenite formation in Oman, all the three principal total stresses are compressive in both contractional and extensional tectonics. Therefore, high fluid pressures are required for dilatancy. It is unfortunate that in tectonics the names "contractional" and "extensional" (these are used to denote displacement directions) are often misunderstood to mean compressive and tensile stress. In our opinion, the processes proposed in our manuscript can occur in both contractional and extensional tectonic settings. Therefore, our analysis of the complex, "ongoing solid-state volume increase reaction of ultramafic rocks to serpentinite and listwaenite" is applicable for both settings, and cannot be easily explained by an "extensional event".

To sum up the validity of the submitted manuscript, the authors provide detailed geochemical/analytical and microstructural results, but the link to the geodynamics setting of listwaenite formation is not at all

conclusive, especially since alternative models for listwaenite formation are left unmentioned and unconsidered.

As mentioned above, the aim and main topic of this paper was to investigate the relationship between deformation, reactive fluid flux, and listvenite formation, irrespective of the tectonic setting where this occurred. All three reviewers agreed with the validity of our analytical and microstructural results and their interpretation. Although outside of the main goal of our study, we have revised the manuscript to include a lengthier discussion around possible tectonic settings. We add to the discussion:

“A well-suited fluid source are carbonate-bearing sediments rich in hydrous silicates that undergo devolatilization related to heating, for example due to contact metamorphism or burial during thrusting^{6,40,50}. In the Samail ophiolite, burial of the Hawasina nappes and underlying units below the ophiolite caused fluid flux and the formation of overpressure cells³⁵, which, combined with active deformation along the basal thrust, might have created the circumstances necessary for listvenite formation. CO₂-bearing fluid may have been sourced from deeper, partially subducted sediments, where temperatures are higher and devolatilization reactions more efficient^{33,40}. Alternatively, it has been proposed that listvenite formation may have coincided with extensional deformation after obduction of the Samail ophiolite^{38,39}. However, the parallel alignment of listvenite bands around Site BT1 with the gently folded, low-angle contacts between the basal mantle section of the ophiolite, the metamorphic sole and the Hawasina sediments³³ indicates that the listvenites formed along the low angle, basal thrust of the ophiolite, and not along steep faults during later extension. This geometry, isotope geochemistry⁴⁰, and the isochron age⁸ indicate that the most likely setting of listvenite formation at Site BT1 is subduction-obduction.”

The work describes detailed microstructural analysis of listwaenite and listwaenitisation. This is interesting for specialists working on this matter (in geochemistry and fluid-rock interaction processes) but its significance to the larger community is limited.

We do not agree with this statement; we believe that this paper presents exciting new results that significantly improve our understanding of the mechanisms of coupled deformation and fluid-rock interaction in mantle rocks, which is of high importance for anyone studying reactive fluid flux and carbonation processes. The implications of our findings reach far beyond the regional context of the Fanjah region of Oman with direct impact on CCSU research and application in mafic and ultramafic rocks.

A discussion of the observed structures with respect to tectonic events would also increase the significance of this manuscript (i.e., the described S-C fabric, boudin and folds; presence of kinematic indicators for events during which these structures formed?). Even though, I am not a specialist in the microanalytical field, I am confident that the described features exist, and that the used methods are of high-quality, but I have to seriously criticize the authors for not indicating in the Methods section what kind of well material was at their disposal. There is no indication whether they worked on an oriented well core or not. In case they did, the fold vergences, shear sense should be provided with cardinal directions – critical in support of the subduction or extension interpretation. The lack of orientation/direction information is another main weakness of the manuscript.

A lack of orientation data of our samples from the inclined drilled Hole BT1B, the difficulties to identify and measure small-scale structures macroscopically in the field, and the ubiquitous occurrence of cataclasites and faults overprinting the listvenites after carbonation (see Menzel et al., 2020 JGR), where rotation of different fault blocks is likely, do not allow a confident correlation of orientations. Therefore, our data do not allow a more detailed structural geologic interpretation: this is not the aim of our paper. We do note

that an extensive and detailed report of the measurements and analyses has been fully published and accessible in detail and with full transparency in the cited Proceedings of the Oman Drilling Project (Kelemen et al. 2020). Per the reviewer's suggestion, we have expanded the explanation of the nature of the samples and orientation data available to us in the Methods section.

As mentioned above, the models for the listwaenite formation are not discussed. Thus, specific references are missing. I have indicated those into the detailed text to the reviewers. Furthermore, some references to the geological map and cross section are missing. I have added them too.

We have added appropriate referencing where necessary.

The authors also missed that ductile-brittle deformation in listwaenite of the greater Fanja area has been described already. I have mentioned the respective reference below.

Here we respectfully disagree: we present here for the first time detailed microstructural analysis and, based on this data, an exciting new model for Listvenite formation, with high relevance for our understanding of fluid-rock interactions.

The conference abstract the reviewer refers to describes field structural relations of brittle and semi-ductile deformation in listvenite. But unfortunately, the actual data have not been published, so we cannot compare our results in detail, and cannot evaluate if the described structures are synchronous to listvenite formation (such as the folding of veins described in this paper) or if they are related to tectonic overprints after carbonation (similar to the cataclasites and faults described by Menzel et al., 2020 JGR).

Finally, the manuscript reveals a limited familiarity of the authors with the regional geology of Oman, and the respective literature. Text, figures and literature references must be improved accordingly.

We respectfully disagree. J.L. Urai and P. Kelemen have investigated different parts of the geology and tectonics of Oman over several decades, with more than 50 publications in high-ranked peer-reviewed journals.

For the reasons stated above, I cannot recommend publication of this manuscript.

Andreas Scharf, Muscat, June, 2nd2021.

Following reviewer #1's comments and recommendations, we have made significant modifications to the manuscript and to Figure 1 (see also replies to detailed comments below), and hope that we have addressed all issues appropriately, providing an improved revised version of the paper.

Replies to detailed comments by Reviewer 1:

“(see Jackson, 2005 for spelling of listwaenite)”

Several forms of spelling of the word listvenite can be found in literature, just as different spellings of Samail / Semail / Sumail exist. We prefer “listvenite” because it is closest to the spelling of the type locality in the Ural mountains, derived from kyrillic as argued by Halls & Zhao 1995.

Abstract

Line 2. "Oman Ophiolite" is misleading. There are two different ophiolites in Oman. I would be specific ("Semail" or "Samail" or "Sumail" Ophiolite – the spelling of "Samail" is different in the literature). Do this for the entire manuscript and art work.

"Oman ophiolite" has been used in papers by pioneers of the ophiolite research such as Nicolas and Boudier; but we recognize that there is a second ophiolite in Oman so Samail Ophiolite is more accurate. We have changed "Oman ophiolite" for "Smail ophiolite" throughout the manuscript and figures.

Line 2. Several. Could you be more specific? Do you have a value?

Line 3. What is "GT"? Do you mean "Gt" (gigatonne)?

2- 3 Gt.

Line 6. Tectonic deformation. This is the key, but what kind of deformation (i.e., compressional, extensional?). Be specific!

See reply to major comment

Line 6. Folded veins. Here you should provide information of the folds. Are they open, isoclinal, harmonic,ptygmatic? Orientation of the fold axes, axial planes? Do you observe several folds with the same orientation?

The word count of the abstract is too limited to include these details here (those are, when possible, described in the results section. See also reply to comment about orientations).

At which temperature does magnesite grow or become folded/ductile? This would provide information about the conditions during the respective deformation event.

There only are deformation experiments for $T > 500$ °C, no data exists for these low temperature conditions.

Line 11. I am not sure how the lithostatic pore pressure can promote a dilatant porosity, especially within a subduction zone (I can imagine that pre-existing pores could be flattened in a subduction zone due to simple shear at an early stage). Generally, I would rather expect the contrary. In a subduction zone, any potential pores will be closed, considering the lithostatic pressure. Why do you not consider CO₂ transport during extension instead within a subduction zone?

High pore pressures can lead to tensile fracturing also in non-extensional conditions. In Oman, this has been shown for carbonate veins in (par)autochthonous units in response to burial below the ophiolite and the Hawasina nappes, which created fluid overpressure cells (Grobe et al., 2019, Solid Earth, 10, 149–175). There are well studied examples of compressional shear zones facilitating fluid flux and related alteration, e.g. the Glarus thrust (e.g. Burkhard et al., 1992, Contributions to Mineralogy and Petrology 112, 293-31) or thrusts in the Pyrenees (e.g. Muñoz-López et al., 2020, Geofluids), as well as numerous subduction melanges (e.g. Angiboust et al., 2014, J Petr). See also Sibson, 2017 (Earth, Planets and Space 69:113). To clarify: the mantle rocks form the upper plate, and they started as essentially zero-porosity peridotites that became more porous and permeable only due to hydration. This is a very different situation than flattening and closure of pores that would occur in a subducted slab. That said, please note that we do not limit our model to compressional settings such as subduction zones, and that also in extensional, (semi)ductile shear zones, creation of fully pervasive porosity is by no means guaranteed.

If your assumption is correct, i.e., dilatant porosity during subduction, then listwaenite must be common in every subduction zone of the world and everywhere distributed at the base of the Semail Ophiolite.

We do not agree with this comment. There are other requirements for a listvenite to form apart from porosity: CO₂ concentration in aqueous fluid must be very high and CO₂ flux must be sustained, also, high / lithostatic pore pressures need to be repeatedly renewed by fluid influx to maintain this process; all these factors need to play together to allow listvenite formation.

However, listwaenite occurs in the Fanja area only with few exceptions near the Hawasina Window and in the UAE. In the Fanja area, listwaenite occurs in extensional fault zones.

There are serious problems with your statement.

That extensional and strike slip faults cut listvenites and surrounding rocks in the Fanjah area (e.g. Scharf et al., 2019 Tectonics) is not proof that listvenite formed during that stage of tectonic deformation. Cross-cutting microstructures of sharp faults actually indicate that they were active after listvenite was formed, and were often related to carbonate dissolution, not carbonation (see Menzel et al., 2020 JGR). Still, as outlined above, we do not exclude the hypothesis that listvenite may have formed during early reactivation of the thrust as extensional shear zones, but we consider it unlikely.

Introduction

Line 16. Do you have a reference for your statement?

Statement removed

Line 21. The topic with the “leading edge of the mantle wedge” is a discussed matter in literature. I do not see any geological or structural convincing evidence that your case stated is correct. It is common practice to mention an established alternative (Listwaenite does not form within a subduction zone but post-obductionally during extension, i.e., Stanger, 1985; Wilde et al., 2002; Nasir et al., 2006; Mattern and Scharf, 2018; Scharf et al., 2019, 2020).

Mattern, F. & Scharf, A. (2018) Postobductional extension along and within the Frontal Range of the Eastern Oman Mountains. *Journal of Asian Earth Sciences* 154, 369-385, doi: 10.1016/j.jseae.2017.12.031.
Scharf, A., Mattern, F., Bolhar, R., Bailey, C.M. & Ring, U. (2019) Early Cenozoic listwaenite formation at a major extensional fault zone of the Oman Mountains (Fanja area) – insights from structural analysis and U-Pb carbonate dating. AAPG Event, Structural styles in the Middle East, 9th-11th December 2019, Muscat, Oman.

Scharf, A., Mattern, F., Bolhar, R., Bailey, C.M. & Ring, U. (2020) U-Pb dating of postobductional carbonate veins in listwaenite of the Oman Mountains. International Conference on Ophiolites and the Oceanic Lithosphere: Results of the Oman Drilling Project and Related Research, 12-14th January, 2020, Sultan Qaboos University, Muscat, Sultanate of Oman.

Stanger, G., 1985, Silicified serpentinite in the Semail nappe of Oman: *Lithos*, v. 18, p. 13-22.

Wilde, A., Simpson, L., and Hanna, S., 2002, Preliminary study of Cenozoic hydrothermal alteration and platinum deposition in the Oman Ophiolite, in Jessell, M. J., ed., *General Contributions: Journal of the Virtual Explorer*, v. 6, p. 7-13.

We have expanded our section on the geological background to mention these alternative models. Line 87: “An internal Rb-Sr isochron with an imprecise age of 97 ± 29 Ma (2σ)⁸ suggests that listvenite formation was concurrent with subduction of the Arabian margin³³, ophiolite obduction (97 – 74 Ma; e.g., 37), or a very early phase of post-obduction extension^{35, 38, 39}.” We believe this is an objective description of the different possible settings. We further provide a short discussion about the geodynamic setting, including the hypothesis of carbonation during extensional tectonics, and explaining why a subduction-obduction

setting is more probable (line 389). However, as that is not the aim of this paper, we do not go into an in-depth discussion about the geodynamic setting of listvenite formation.

Line 29. This sentence is hard to read/understand. Could you please rephrase it and produce two nice sentences?

done

Line 65: I am not sure if it is just "listvenite" (not "listvenites").

Line 69: The last sentence is interpretation and does not belong to the introduction. Thus, skip this sentence.

It is common practice to provide the reader with a key conclusion at the end of the introduction, to increase readability and impact of the paper.

Results

Line 75: mostly at the base. There are examples where the listwaenite occurs in a distance of ca. 2.5 km (in map view) from the base of the ophiolite (i.e., near Buwah). You are showing it in your Fig. 1a.

Changed to: "in serpentinite at or close to the base of the Samail ophiolite"

Line 88: Could you provide the error margin (Sigma 1 or Sigma 2) for the Rb-Sr isochron age? I assume that it is Sigma 1, because no details about the error margin are provided in the mentioned reference (at least I could not find any). However, one of the co-authors of the submitted manuscript is a co-author of the provided reference. Thus, this person must know the missing information. The error margin is just gigantic (± 29 Ma Sigma 1; ± 58 Ma Sigma 2).

We contacted the first author of the paper in question (L. Falk) and she confirmed that the ± 29 Ma error margin is indeed sigma 2. Of course, the error is still large, which is why we added the word "imprecise" in the text.

Line 90: Please provide the obvious alternative model that the listwaenite formed after the obduction was completed (e.g., Stranger, 1985; Wilde et al., 2002; Nasir et al., 2006; Mattern & Scharf, 2018; Scharf et al., 2019, 2020). I wonder why this important aspect is not mentioned in the text.

We modify this line to: "An internal Rb-Sr isochron with an imprecise age of 97 ± 29 Ma (2sigma)⁸ suggests that listvenite formation was concurrent with subduction of the Arabian margin³³, ophiolite obduction (97 – 74 Ma; e.g., 37), or a very early phase of post-obduction extension^{35,38,39}." See reply to previous comment.

Line 90: In the Hawasina rocks underlying the ophiolite in the Fanja area are hardly any carbonate-bearing rocks. Actually, the geological map of Fanja indicates that mostly the Al-Jil Formation (Aj1Mbs) crops out in the Fanja area. This formation consists of radiolarian chert, siltstone and sandstone. There are some minor outcrops of other Hawasina formations exposed (Aj1Vb – basaltic pillow lava; Aj1-2 – flaggy limestone, radiolarian chert and sandstone, and UmC – red radiolarian chert and micritic limestone). Thus, the statement is not supported by evidence. The obvious alternative would be that the source for the carbon is from the Arabian shelf (carbonates), cropping out all over in the area.

First of all, due to abundant tectonic overprint (e.g. Menzel et al. 2020), it is quite clear that the present-day configuration is not the same as it was during listvenite formation. But more importantly, Sr and C isotope signatures of sediments of the Hawasina nappes do provide evidence as they are consistent with those of magnesite in the BT1 listvenite. Furthermore, listvenite formation does not only require a source rock that is rich in carbon, but a rock that will produce aqueous fluids that have a high enough CO₂ concentration. Low temperature hydrothermal fluids that percolate limestones do not meet this requirement. On the other hand, formation of such CO₂-enriched fluids is thermodynamically attainable for sediments that contain carbonate-silicate mixtures if these are heated to (at least) low-grade metamorphic conditions. We address this in the discussion, and further details about this background can be found in a number of papers that concern listvenite formation in general, for example Klein & Garrido 2011 and Menzel et al., 2018, and, with respect to listvenites in Oman, Falk & Kelemen 2015, Kelemen et al (preprint) (all references are cited in the paper).

Putting it another way: if the shelf carbonates were the source, why are there not listvenites all around the Jebel Akhdar and Saih Hatat, where carbonate rocks are very abundant below the ophiolite (as are faults and detachments)? The answer is that there are several requirements that need to be met: i) a source rock that has a suitable composition to produce CO₂-rich fluid, ii) devolatilization or fluid fluxing of that source rock at conditions that produce CO₂-rich fluids in high quantities, which in turn requires some kind of heat source, iii) faults, shear zones, a plate interface or another structure along which these fluids can be channelled into peridotite. We find that under-thrusting of mixed, carbonate bearing marine-siliciclastic sediment rock below the ophiolite (a configuration that is equivalent to a shallow subduction zone) is a very viable scenario where these requirements can easily be met, because the subsidence causes heating of the sediments upon equilibration of the geotherm, allowing for devolatilization to occur. High CO₂ concentrations in fluid are even easier to attain at greater depth, and isotope geochemical data suggests that deeper sourced fluids from the subduction zone are a viable scenario (see de Obeso et al., preprint available for reading and commenting on ESSOAR). Fluid flux during early extensional reactivation of the thrust would also be consistent with these considerations, but, as stated previously, we consider it unlikely. However, a simple “obvious” limestone CO₂ source on the other hand is not well thought through in our opinion.

Line 95: You could mention that two dolomite veins cutting the listvenite from the Fanja area have an age of ca. 60 Ma (Scharf et al., 2020).

The referred work reporting the U/Pb dates is a conference abstract. Because we cannot find a published paper, preprint or full online resource that contains this dataset, unfortunately we cannot fully evaluate and cite it.

Line 114: Could you provide any information about the fold, please (i.e., style, interlimb angle, orientation of the fold axis and axial plane, fold vergence?). How many folds did you observe and are they all consistent in terms of style and orientation? Did you observe folded listwaenite at the surface near the well? Brittle to ductile deformation in listwaenite has been reported already (Scharf et al., 2020b). The location is at the northern margin of the Saih Hatat.

Scharf, A., Mattern, F. & Mattern, P. (2020) Ductile-brittle shear zone in a listwaenite body within the Frontal Range Fault of the Oman Mountains (Sultanate of Oman). EGU 2020-12672, Vienna, Austria.

Line 116: S-C fabric. Interesting. Could you provide an assumed shear sense? Is the shear sense consistent for several S-C fabrics? In which direction did shearing ensue?

I am afraid that you do not have any oriented samples. Thus, you cannot make any statement to shear sense and its relationship to tectonic events. Am I correct?

Line 137: Did you observe any shear sense within the shear zones? How do you know that it is a shear zone?

Line 160: Do you have any shear sense for the sigma-clasts?

Line 160: "with a sense of shear consistent..." What is the shear sense?

Line 174: Also, here what is the shear sense? What is the orientation of the fold axis and axial plane? What is the style/shape of the folds? Do all the folds have a similar orientation and style?

WRT all above comments: as stated in Methods, the samples are oriented only in the arbitrary spatial reference frame of Hole BT1B, which is not easily translatable into cardinal directions due to its inclination and the non-reconstructable orientation of discontinuous core sections (as is also described in detail in ref. 32). Therefore, fold orientation and vergence, orientation of apparent s-c fabrics and shear senses from sigma clasts cannot be unambiguously attributed and related to specific tectonic events and we cannot determine whether they formed due to thrusting or extensional shear. We modified the methods section "Samples" to clarify this better.

Line 138: Do the elongated magnesite ellipsoids have a preferred orientation? The same for the aligned magnesite dendrites?

Yes, see a few lines further down: "In 3D, micro-CT shows that magnesite ellipsoids are oblate in the foliation plane"

Line 147: Could you determine the orientation of the short and long axes of the boudin?

No, we only have the apparent orientation in thin section plane here.

Line 202: Please provide the meaning of all of your abbreviations such as "TEM" and "STEM" at the first appearance in the text.

done

Discussion

First section: According to the detailed microstructural observations, I would suggest that the listvenite formed within brittle to ductile shear zones. Such or a similar statement is missing in the first section.

We prefer to not add such a statement in this paragraph; it is discussed further below.

Line 255: "few kilometers". I have not read the provided literature (45-47) in detail, but I would check if it is really "few kilometers" and not "few hundreds of meters".

Thanks for pointing this out; we corrected the mistake.

Line 303: red dot.

Line 303: Systematically oriented structures including folds may indicate that the deformation is related to "external stress".

This comment is not clear to us. Does the reviewer mean "tectonic stress"? As mentioned above, due to a lack of appropriate orientation and macrostructural data we are not in a position to make inferences about the consistency and orientation of stresses that gave place to the formation of the observed structures.

Line 387: This lithological boundary would be also the locality for a development of a shear zone. This shear zone would also explain why “the listvenite consist of a few major bands of 10s of meter thickness”.

We add that this boundary (which at the time of carbonation was the reaction front) also likely focused deformation.

Line 405: I am not convinced of your statement and I cannot understand your point of this sentence. What do you mean with “cyclic variations in permeability”? The permeability is just close to zero. A dramatic volume increase will just block any further reaction. Further reaction will only occur if external tensional stress is present, which will result in opening of fractures.

We modified this paragraph to make it clearer. Because complete serpentinization obviously did occur in the serpentinites, permeability was not always close to zero. Furthermore, high-resolution images indicate that some level of porosity/permeability is preserved within both serpentinites and listvenites (Figure 7). If the build-up of pore pressure is high enough, tensional fracturing may occur (also if the external stress field is not tensional), allowing for a pulse of fluid flux. Sealing due to volume expansion of serpentinization will disrupt fluid flux again, until pore pressures increase again, thus creating cyclic variations in permeability. We believe that with the modifications it should be clear.

Line 410: I cannot follow your speculation. The speculation is not supported by your data. Why do you not consider that the listwaenite and related reactions formed during an extensional stress regime instead within a subduction zone? This would easily explain your observations without needing to construct such a highly speculative statement.

See reply to previous comment. The proposed mechanism is not exclusive to or implicit of subduction zones, but just as well applicable to ductile/semi-brittle extensional shear zones. The arguments and reasoning for it are elaborated in the preceding paragraphs.

Line 414: For the same reason as above, I cannot follow and agree with this statement.

We modified this paragraph to make it clearer

Methods:

Line 422: What is missing is a clear statement whether the samples are oriented or not. This is an important point and must be included. I wonder why the authors did not mention this in the text.

We realize that referring to the arbitrary core reference frame of Hole BT1 is not self-explanatory for readers who are not familiar with IODP core logging standards. We add: “ Hole BT1B (International Continental Drilling Project Expedition 5057-4B) was drilled with an inclination of 75°. Therefore, and because the drill cores contain discontinuities across which the orientation of single core sections could not be restored ³², orientations of structures are unfortunately not easily comparable between different parts of the core and the field.”

Figure 1.

a)

Muscat is depicted at a wrong location. Muscat is further to the east.

The term “Jebel Akhdar” is not good visible because of the intersection with black lines. Please remove the “Jebel Akhdar” into the light blue rocks and reduce the faults, that there is no intersection anymore.

Figure 1 has been improved to account for the reviewer's comments/corrections.

You have drawn a major extensional fault near the well in your section. This fault is not shown on the map. It is the Wadi Mansah Fault Zone of Bailey et al. (2019) and Scharf et al. (2019b). This fault zone has also a sinistral component, which you also could include in the map and section. Some of your "Hawasina" outcrops to the NNE of BT1 do not exist in the geological map of Fanja. I do not know where you got them from. Please indicate!

Bailey, C.M., Hurtado, C., Scharf, A. & Mattern, F. (2019) Structural controls on listwaenite genesis in the Semail Ophiolite, Northern Oman. EGU 2019-5794, Vienna, Austria.

Scharf, A., Mattern, F., Moraetis, D., Callegari, I. & Weidle, C. (2019b) Postobductional kinematic evolution and geomorphology of a major regional structure – the Semail Gap Fault Zone (Oman Mountains).

Tectonics 38(8), 2756-2778, doi: 10.1029/2019TC005588.

Please provide a reference for your geological map.

The reference of the map source (Béchenec et al, 1992) has been indicated in the figure caption, but we slightly modify where it is placed to make it more clear. Other than simplifying and grouping units and plotting only selected faults for the schematic view, we did not modify the mapped region. As this map is a revision done by Béchenec and others of the earlier Fanja NF40-3F (which dates to 1982- 1985) and neighbouring maps, we used this source.

b)

Why is the contact of the "Paleozoic basement" with the "Mesozoic" folded (in the Saih Hatat Dome)? This is not directly related to the study but what you have drawn is incorrect.

The pre-Permian rocks of the Saih Hatat Dome comprise a major recumbent fold (compare Miller et al., 2002; see also Hansman et al., 2021 for a review of the deformation in the Saih Hatat Dome).

Corrected. The recumbent fold is schematically indicated by a dotted trace, but this detail is not relevant for the present study.

Miller, J.M., Gray, D.R., Gregory, R.T., 2002. Geometry and significance of internal windows and regional isoclinal folds in northeast Saih Hatat, Sultanate of Oman. Journal of Structural Geology 24, 359-386, doi: 10.1016/S0191-8141(01)00061-X.

Hansman, R., Ring, U., Scharf, A., Glodny, J., Wan, B., 2021. Structural architecture and Late Cretaceous exhumation history of the Saih Hatat Dome (Oman), a review based on existing data and semi-restorable cross-sections. Earth-Science Review 217, 103595, doi: 10.1016/j.earscrv.2021.103595

The extensional fault at BT1 is much too steeply dipping. Your trace of the section is oblique to the fault. Thus, in your section, the extensional fault must dip with $<60^\circ$. You could also indicate the sinistral horizontal shear sense as described in Scharf et al. (2019b). The vertical displacement at this fault is ~7 km (see Scharf et al., 2019b). See also for the listric shape of the large extensional fault Scharf et al. (2019b).

We redraw this fault to make it less steep and to schematically indicate that it is likely listric. However, please note that the cross section is, as stated, schematic, and hence the vertical exaggeration is high and not to scale.

c)

The "Autochthonous Mesozoic sed." Are actually "Par-autochthonous Permo-Mesozoic sed."

Corrected

The thickness of the Hawasina nappes in the Fanja area is max. 1 km; see geological map of Fanja. According to the geological map of Fanja, the thickness of the ophiolite is <3 km (not 8-12 km). The thickness of the oceanic crust is <4 km (not 5-7 km).

Please note that the column is a very simplified stratigraphy that serves to give the reader a general context of the Samail ophiolite and the position of BT1 within this idealized stratigraphy (see cited reference). To clarify that thickness of units may be less in the area, we add a statement: “Present-day thickness of units in the area is commonly reduced due to tectonic thinning” to the figure caption. Substantial tectonic thinning was likely also related to abundant cataclasites and faults cutting listvenite (i.e. having formed later than the main carbonation event – see Menzel et al. 2021 JGR), so the accurate thicknesses of units at the moment of listvenite formation is impossible to reconstruct without precise dating of listvenite formation as well as each deformation phase, which is currently not available. A detailed structural and geodynamic reconstruction of the Fanjah region that would additionally be required for that is beyond the scope of this study, that would be a paper in its own right.

Post-obductional sediments (not obduction)

We prefer post-obduction sediments, similar to “post-obduction deformation”

d)

Please provide a scale.

Indications of elevation above sea level for the Wadi bottom and one of the hill summits are now given on the figure.

I would add that the serpentinite is difficult to spot, because it is largely covered by listwaenite scree/talus. done

e)

Why has the metamorphic sole two colors? The word “sole” is hard to read.

These are different sub-units within the metamorphic sole (See cited reference)

J and k)

I would add the scale bar values directly into the images. You would save some text in the caption (do this for all the other figures, too).

This is a formatting requirement / journal policy of Nature Communications.

I cannot comment on the other figures because I am not a specialist in microstructures. However, the images look very nice.

Supplementary figures

Fig. S1. All figures need cardinal directions.

What is the orientation of the shear zone in “b”? Do you observe any shear sense?

The shown field images were not taken with markers of cardinal directions. Unambiguous shear sense indicators were macroscopically not observed in this shear zone. Such structures are typically not very clear macroscopically due to intense oxidation crusts from weathering of listvenite, while polished surfaces lack the 3D information. Note that also in Hole BT1B listvenite, most observed structures that may provide a sense of shear are difficult to see macroscopically and require microscopic investigation.

Fig. S2.

I assume that the samples are not oriented. Cardinal directions?

Samples are oriented in the (semi-arbitrary) core reference frame (CRF) (see Kelemen et al., 2020), which we indicate on the figure.

Fig. S6. What is the orientation of x, y and z?

Directions x, y, z are those of the BT1B core reference frame (CRF), this is now indicated in the caption of the figure.

Reviewer #2 (Melanie Sieber):

I've thoroughly enjoyed reading the manuscript titled "Ductile deformation during carbonation of serpentinized peridotite" by Menzel, Urai, Ukar, Hirth, Schwedt, Kovacs, Kibkalo and Kelemen. A detailed petrographic description of deformed listvenites from the Oman ophiolite is presented, and the authors argue that deformation assisted carbonation. I think this paper would make an excellent contribution to Nature Communications after some revisions.

We are grateful for the Reviewer's high evaluation of the paper and her comments and suggestions, that helped to improve the manuscript.

(1) The authors present a temporal evolution (see discussion, summarised in Fig 9) from peridotite hydration to carbonation associated with deformation. The argumentation and reasoning for this model is consistent with their observations. However, alternative explanations should be considered, particularly for their step 1. For instance, the reported sequence of listenite, talc-magnesite-zones, serpentinites and partially hydrated peridotites could also present a spatial variation in XCO₂ and fluid/rock ratio rather than a temporal evolution. This alternative explanation should be discussed in the manuscript.

We agree with the reviewer that the description of the different reaction stages was not precise enough: we do not want to imply that the different defined reaction steps must have occurred at completely different times, from different fluids. We rather see it as the spatio-temporal evolution of advancing reaction fronts („simultaneous but spatially separated alteration fronts“) which in our view is in agreement with the mentioned chemical CO₂ gradients and differing fluid/rock ratio. To clarify this, we place the following statement in the discussion after the list of reaction stages:

„These different reaction stages likely were active in several simultaneous but spatially separated alteration fronts, reflecting increasing water/rock ratios from I to IV and a chemical gradient of decreasing fluid CO₂ concentrations from listvenite to carbonate-bearing serpentinite and partially hydrated peridotite.“

We further added to the caption of Fig. 8 (previously Fig. 9): “While the sketch illustrates a temporal evolution from reaction stages I – IV, these likely occurred simultaneously across an advancing reaction front, with strain partitioned into the weak, serpentine-bearing and fluid saturated reacting assemblages (stages II and III).”

(2) The petrographic descriptions of the studied rocks are very nice and detailed. The authors also describe chemical variations (e.g. zoning in Fe content in magnesite and CL-zonation in quartz) but unfortunately do not discuss the causes of this zonation. For instance, early formed magnesite are enriched in Fe and associated with Cr-spinel. This may imply that either Fe is contributed by the partial dissolution of Cr-spinel or the various magnesite compositions may reflect variations in the fluid composition.

Yes, a discussion of zoning was lacking. We add after the list of reaction stages in the discussion (line 255): “Zoning of Fe and Ca contents in magnesite (Fig. 3b; see also ref.34) documents variations in fluid

composition and redox conditions between stages II – IV, with reaction of magnetite and Fe-bearing serpentine to Fe-magnesite mostly during carbonation stage II (Fe-enriched magnesite cores; Fig. 1k)."

The association of early Fe-enriched magnesite with Cr-spinel appears also mostly due to magnetite, which typically forms replacement rims around spinel. We infer that a combination of relatively early carbonation of magnetite ($\text{Fe}_3\text{O}_4 + \text{CO}_2 = \text{FeCO}_3 + \text{Fe}_2\text{O}_3$) and variations of influx fluid compositions are the cause of the concentric zoning patterns. We admit that this discussion is still a bit short and does not answer all questions regarding the zoning, but a full understanding of the zoning requires investigation of the interplay between fluid redox conditions and carbonation reaction rates in dependence of Fe contents, and the possible catalytic effect of minor phases like magnetite or spinel, which is beyond the scope of this paper.

(3) I am also missing a good discussion about pore space formation associated with the carbonation reaction. Experimental studies in the context of mineral carbonation of antigorite and serpentinites show a weak carbonation rate, which is taken by the authors to support their argumentation that tectonic deformation is essential in large scale carbonation. Although I have no doubts on their interpretation, the authors should consider that carbonation is reported to be efficient in the replacement of serpentinites at higher pressures and large fluid/rock ratios and that the formation of pore space has been reported in such environments (Scambelluri et al., 2016; Piccoli et al., 2016, 2016; Sieber et al., 2020). This is particularly relevant since the pressure conditions of listvenite formation are poorly constrained for the Oman ophiolite with estimates ranging up to 1.2 GPa.

This is a good point that deserves to be discussed more clear. We add to the discussion (line 333):

"The complete reaction sequence requires large fluid - rock ratios³³ and, thus, sufficient permeability. Because magnesite has a higher density than serpentine, porosity may be created during the transformation of serpentine to talc + magnesite, in particular at high pressures⁶¹ and if Mg and/or Si is leached by fluids⁶². The reaction of serpentine to magnesite + quartz, however, increases solid volume by 18 – 20 % if non-volatile elements remain immobile. Major and trace element geochemistry of the BT1B listvenites indicate no major Mg or Si loss⁶³, suggesting that another porosity-forming mechanism was necessary."

Although the suggested papers by Piccoli et al are very good studies, they refer to a different chemical system (mafic / sedimentary rocks) with different carbonation reactions, thus they are not directly relevant here.

Additionally, I have a couple of minor suggestions and remarks, which are outlined below.

Line 37-38: Even for unusually high CO₂ contents in aqueous fluids (on the order of 1 wt%), this amount of carbonation requires time-integrated fluid rock ratios > 30.

Reference missing – please add.

We understand the comment, but in our opinion this is a simple calculation based on the CO₂ content of > 30 wt% (previous sentence), which will thus require a fluid/rock ratio of >30 (by weight) if the fluid contains 1 wt%. We have reached the maximum number of papers to be cited.

Line 80-81: and a nearly complete lack of talc suggest temperatures of 80 – 150 °C during listvenite formation.

Temperature is not the only explanation for an absence of talc, could also reflect elevated XCO₂-fluid.

It is true that temperature is not the only control of talc stability. But we note that in a reaction progression, CO₂ in fluid becomes depleted with magnesite precipitation, and those lower XCO₂ fluids may

then form talc in domains with lower fluid-rock ratio. This process is less likely to happen at low temperatures, where in a T-XCO₂ diagram the pseudo-univariant lines of the talc forming and quartz forming reactions are very close to each other. We changed the sentence to say the scarcity of talc supports those low temperatures; further details are given in the discussion (including the scenario of elevated XCO₂ fluid, which, in this setting, is essentially the same as „large disequilibrium of reaction“).

Line 81-82: Low temperatures are supported by intergrown hematite and graphite or amorphous carbon, which require < 200 °C to coexist in equilibrium
What observations support equilibrium?

The veins contain hematite and amorphous organic carbon intergrown at the 10 μm scale; but since the temperatures are low we indeed cannot be fully confident that they are in thermodynamic equilibrium. We modify this statement to: “which require < 200 °C if formed in equilibrium”.

Line 128-131: Please refer to figures showing those two different textures in the listvenite matrix. This will make it easier for the reader to follow your description.

Reference to Fig. 1 j & k added; listvenite resembling mesh serpentinite is not shown here, for this we add reference to Kelemen et al 2020 (Proceedings of the Oman Drilling Project)

Line 135-136: These foliated listvenites constitute ~10 % (13/115) of the studied listvenite thin sections... I don't see why this is relevant information. According to your core-logging the penetrative-foliated listvenite comprises around 25% of HoleBT1B.

The information about the percentage of thin section containing these microstructures is relevant, because the penetrative foliation is typically not visible macroscopically. Therefore, we state that penetrative-foliated listvenites are most evident in these intervals (and not that these intervals are entirely composed of penetrative-foliated listvenites). We added a remark (Line 131) about the macroscopic (in)visibility to clarify this.

Line 142: silica inclusions
Just silica, not amorphous or cryptocrystalline SiO₂?

We mean SiO₂ here (corrected). Where they are big enough it is possible to determine them as quartz, but most inclusions are too small (< 100 nm) to be sure.

Line 207 and Figure 6: The abbreviation for low-angle boundary is not consistent.

Corrected in the text

Line 236-237: Instead of referring to a BSE image, please add the image of the micro CT showing the porosity.

We add volume rendered images of segmented porosity to the supplementary material Fig. S6.

Line 257-260: Serpentinization likely preceded carbonation, because there are 1 – 10 m wide, fully hydrated serpentinite zones between listvenite and partially serpentinized peridotite¹⁸, and because Fe-oxides in listvenite commonly trace a former mesh texture that is typical of serpentinization of olivine
I am not convinced that serpentinization must have been completed before carbonation initiated. Please evaluate other potential explanations for the listvenite-serpentinite-partially serpentinized peridotite sequence such as a chemical gradient (decreasing XCO₂ from listenite to partially hydrated peridotite) and decreasing fluid/rock ratio (see Sieber et al., JP, 2020). When a CO₂-bearing aqueous fluid infiltrates into a

partially serpentinized peridotite, listvenite might be formed in the centre of the vein, but when the fluid migrates lateral from the injection into the surrounding peridotite, XCO₂ will decrease as carbonates are formed and the fluid/rock-ratio declines which may lead to a pronounced hydration. The described occurrence of talc in m scale transitions between listvenite and serpentinite (line 272-275) may support a variation in the chemical gradient rather than a temporal evolution.

Thanks for this comment. We do not exclude that some listvenites may have formed by direct replacement of olivine- and pyroxene bearing, partial serpentinized peridotite; we modified this statement (new line 244) to make this clearer. Regarding the CO₂ chemical gradient, please see our previous reply to the major comment.

With respect to: ‘When a CO₂-bearing aqueous fluid infiltrates into a partially serpentinized peridotite, listvenite might be formed in the centre of the vein’:

This is correct, but we would expect this situation to occur only rarely in nature. As stated, the CO₂ concentration will decrease with carbonate precipitation, so downstream along the fluid flow path the fluid is comparatively rich in H₂O, which will not only lead to serpentinization laterally but also along fluid flow, ahead of the listvenite forming reaction front. Thus, if fluid flow follows similar pathways, it is likely that a serpentinization front precedes the listvenite formation. The main reason for our interpretation is however the common observation of Fe-oxides in listvenite that trace former serpentine mesh textures (as stated, and see reference 32); this microstructure would not form during carbonation of fresh olivine/pyroxene. Furthermore, the crystal growth microstructures (Fig. 7) consistently show that quartz formed texturally later than most of the magnesite, which is consistent with an intermediate reaction stage of carbonate-bearing serpentinite.

Regarding talc: it may indeed support this model. But although talc does occur in small amounts at some of the contacts between listvenite and serpentinite in Hole BT1B, due to lack of clear textural relations it cannot unambiguously be attributed to a simple chemical gradient / reaction front. Talc inclusions in magnesite growth rims that one would expect to be common are very rare. Since there is a high degree of post-listvenite overprint in many parts of the cores (e.g. during late stage cataclasis, which was also related to fluid flow and chemical modification; Menzel et al., 2020 JGR), we cannot exclude the possibility that the rare talc formed through alteration of serpentine by late, silica-bearing fluids, unrelated to the carbonation itself. These details are however beyond the scope of this paper.

Line 297: Subsequent carbonation under variably supersaturated conditions caused the growth of magnesite ellipsoids with partially euhedral rims in the serpentine matrix.

What do the authors understand by ‘variable supersaturation’?. In a dissolution-precipitation process (as described by A. Putnis and Co-workers) the stable secondary phase (=magnesite) is supersaturated in the interacting fluid-volume. But what is variable in this context?

Precipitation occurs in theory (if not inhibited by other factors) always if a solution is supersaturated with respect to that phase, but the degree of supersaturation does have a strong control on crystal size, nucleation rate and growth rate (for homogeneous nucleation). Variable supersaturation may therefore explain why some magnesite rims are euhedral (formed closer to equilibrium saturation) while later overgrowth rims are dendritic and contain a lot of intragranular pores (higher supersaturation; i.e. larger disequilibrium).

Figure 8: What is the difference between the solid lines and the dotted lines?

All lines were modified to be dotted (note that this figure, following a recommendation by another reviewer, is now in the supplements).

Figure S13: (a) Are you sure this is a BSE image, not from a light microscope?; (d) Please add labels for Cr-spinel and hematite into the image.

corrected

Best wishes,
Melanie J. Sieber

Thank you for the constructive comments!

Reviewer 3 (Anonymous)

This study aims at constraining the relation between serpentinite carbonation and deformation that affected the Oman ophiolite. Considering the great value of the samples which were collected in cores from the Oman Drilling Project, and the potential high interest of the scientific community that works on carbon sequestration, I believe that the present paper certainly fits the scope of the journal.

We are grateful for the Reviewer's high evaluation of the paper and appreciate the constructive comments and suggestions.

Overall comments

First of all, I have to say that, as a non-specialist of the study of microstructures and EBSD facilities, I was impressed by the quality of this paper. In a general point of view, the paper is well-written, geological and mineral descriptions are very detailed and clear, and the figures are highly relevant to answer questions asked by this study (except for one, see below in the specific comments). My main concern is about the last paragraph of the discussion where the authors assume that the reaction assisted ductile deformation may be common in subduction zones and could be at the origin of aseismic creep in some regions. Although I understand that the authors need a "punchy" end for their paper (and I have to admit that this may be plausible), I found this assumption highly speculative and additional arguments are required to be written as it.

We agree with the reviewer that the last paragraphs were not clear enough. We modified the last part of the discussion to clarify that we think that our conclusions are applicable to coupled deformation and reactive fluid flow in a wide range of tectonic settings. Then we go on and provide an explicit reasoning why we consider a subduction-obduction setting the most likely geodynamic context, and hence that our observations can be seen as analogous to conditions at the shallow mantle wedge of subduction zones, justifying a discussion what that may imply. We hope that with the modifications it is clearer now.

On the whole, I think that this paper is suitable for publication in Nature Communications after minor revisions.

Additional minor comments and suggestions are given below.

Specific comments

Results

- lines 92: The reference 33, i.e., Kelemen PB, et al. Mass transfer into the leading edge of the mantle wedge: Initial Results from Oman Drilling Project Hole BT1B. is currently in review and not published yet. I found nothing against this in the author guidelines, but it's always disturbing having references that are not accessible at the time of publication.

The reference had been made available on a preprint server shortly after submission of this manuscript, therefore the doi of the preprint was not included; we updated the reference details to guide the reader to it and apologize for any inconvenience this caused.

- lines 102: Citation 39 should be replaced by a more appropriate one: mesh and bastite textures well as serpentine veins are obviously typical of hydrated mantle peridotite, but many "older" references had documented this in the literature prior the one you cite.

We replace the reference by *Wicks & Whittaker 1977: Serpentine textures and serpentinization*.

- lines 109: I do not believe that EBSD is useful to discriminate between serpentine varieties...

EBSD can discriminate whether there is any well-crystalline antigorite, but not whether serpentine is lizardite or chrysotile otherwise. We modified the statement to make it clearer.

Discussion

- lines 414-419: This assumption is highly speculative and should be, if possible, slightly developed and better argued to be considered.

We modified the last two paragraphs of the discussion to provide a better discussion of the requirements of listvenite formation and suitable geodynamic settings, and reasoning why our results can be viewed as analogous to processes occurring in the shallow part of the mantle wedge of subduction zones.

- Figure 8: This figure should be removed (or replaced by one of the supplementary material): it is not clear, too "heavy" likely because there is too much text inside. In addition, I found that the information provided by this figure is redundant with Figure 9, which perfectly illustrates the processes you described and which I consider much clearer

We move this figure into the supplementary material. It contains some information not shown in figure 9 but which is less relevant to the paper, so we agree that it does not need to be a main figure.

REVIEWER COMMENTS

Reviewer #2 (Remarks to the Author)

All my major comments and concerns have been addressed, and I am happy with the latest version of the manuscript to be published in nature communications.

Reviewer #3 (Remarks to the Author):

The authors have taken into account most (to not say all) of the comments I made after my first reading of the paper. In particular, I appreciate the new last paragraph, which have been rewritten and which is now clearer and better justified. As a conclusion, I would like to express all my congratulations to the authors for this beautiful work, which I found now ready for publication in Nature Communications

Reviewer #1 (Remarks to the Author):

See attached pdf

Reviewer #1 (Andreas Scharf) (in red is my reply to the authors comments/reply)

The authors produced a technically well-written manuscript, which describes in detail different microstructures, microcrystallographic orientations and microanalytical analysis of carbonated ultramafic rocks from the well BT-1 of the Oman Mountains. Deformation is partly ductile. Finally, the authors link the observed data and fluid flow to a feedback mechanism in subduction zones. The microstructural and analytical results of the submitted manuscript are sound. Although I am not an expert in microstructural analysis, the presented results are comprehensible.

We thank the reviewer, who notes that he is not an expert in microstructural analysis, for his comments and are glad to see that he agrees with the main conclusions that we draw from our results, namely that there was fluid-assisted ductile deformation during the carbonation reaction and that this deformation in turn helped to facilitate fluid flux and reaction progress.

My main critical concern is that the authors link their data to a feedback mechanism in subduction zones as the authors fail to establish that the subduction zone environment is applicable. The link between the author's data and the 'subduction zone fluid flow' is neither supported by their data nor by other data (i.e., structural data from the core or surrounding field area – none of them were presented or discussed).

We thank the reviewer for his clear comments and have rewritten the manuscript to make clear that the deformation-assisted carbonation we discuss here derives directly from the observations and data presented in this paper, that it can operate in a range of different tectonic settings, and that, based on the currently available data from recent and past studies, a subduction-obduction setting is in our opinion the most likely configuration of listwaenite formation at site BT1. For the latter, we now provide an explicit and consistent reasoning in the manuscript; please find further details also in the following replies.

The authors have to keep in mind that two contradicting models for the listwaenite (see Jackson, 2005 for spelling of listwaenite) formation in Oman exist. Model one was proposed by Falk and Kelemen (2015). This model proposes that the listwaenite formed within the 'leading edge of a subduction zone' at ca. 95 Ma, thus, during obduction of the famous Semail Ophiolite. Falk and Kelemen (2015) proposed this model in part based on a single Rb-Sr isochron age (97 ± 29 Ma) of a Cr-bearing fuchsite from the listwaenite (Note the great error margin, which could represent various processes, for example post-obductional extension!). I am referring to this in the detailed comments below. The "subduction model" is not supported by field and structural data of the Fanja area.

An older model for the origin of the listwaenite is proposed from Stanger (1985), Wilde et al. (2002), Nasir et al. (2006), and was confirmed by Mattern and Scharf (2018) and Scharf et al. (2019, 2020). This model suggests that the listwaenite formed during extension after the emplacement of the ophiolite. Thus, has absolutely nothing to do with CO₂-fluid transfer within a subduction zone. Some of the authors must be aware of the contradicting models, because the models were discussed at the Oman Drilling Project and Related Research Conference from the 12-14th January, 2020 at the Sultan Qaboos University, Muscat (Sultanate of Oman). Some of the authors of the submitted manuscript attended this conference.

2

A major flaw of the submitted manuscript is that the authors only explain the 'subduction model' without even mentioning the obvious alternative model (the elephant in the room). This raises my concern as to how serious this manuscript was prepared by uncritically leaving out a major aspect to be considered.

We understand this comment and have expanded the Geological Background (Line 89) and Discussion (Line 389) to include the hypothesis that listvenite formation was related to extensional tectonics after ophiolite emplacement. We added relevant references.

We would like to point out that a structural / geodynamic reconstruction of the Fanjah area or constraining the absolute timing of listvenite formation is beyond the scope of this paper, and due to limitations in text length we cannot discuss this in great depth. For a lengthier discussion, the reader is referred to a number of the papers cited in our manuscript. Nevertheless, we would like to point out a few things here regarding this comment:

The timing and geodynamic setting of listvenite formation proposed in older papers is, while valid hypotheses, not supported by data at the time of publication of those papers and inconsistent with newer data. For example, meteoric- or seawater-driven carbonation to form listvenites, as proposed by Wilde et al. 2002 and Stanger 1985, is inconsistent with geochemical and isotope data from Hole BT1B (e.g. Falk & Kelemen, 2015; de Obeso et al., preprint). Likewise, the Tertiary high heat flow at 36 – 40 Ma as proposed for listvenite formation by Wilde et. al. (2002), and which also Nasir et al. 2006 discuss as a viable scenario, is neither in agreement with the Rb/Sr age of Falk & Kelemen 2015 (even when taking into account the large error), nor is it consistent with the 55 – 60 Ma U/Pb date of two dolomite or calcite veins cutting listvenite (Scharf et al, 2020), which places listvenite formation itself to before that age. (Nota Bene: we only find conference abstracts reporting these U/Pb dates, so we unfortunately cannot evaluate and cite these).

The policy of the authors is not to include abstracts into the reference list but to include at the same time non-peer reviewed articles (e.g., Kelemen et al., etc.). This is a case of double standard.

It is therefore puzzling why Reviewer 1 writes that Mattern & Scharf (2018) and Scharf et al. confirmed those previous studies: we cannot find this confirmation in Mattern & Scharf (2018).

Please have a look in Mattern and Scharf (2018) again. There is written “Because listwaenite is a hydrothermal alteration product of serpentinized peridotite (e.g., Nasir et al., 2007) and because it occurs directly at the ophiolite-Mahil contact it must have formed directly in the extensional fault that provided pathways for hydrothermal fluids to circulate to form listwaenite.”. I do not know what the reviewers mean with “Scharf et al.” - which year and which journal/book chapter?

With respect to the ages, we would also like to point out that post-obduction extension at 60 Ma (Why at 60 Ma? Post-obduction extension started immediately after the emplacement or even slightly overlaps in time with the emplacement, i.e., at ca. 75 Ma. This fact is long known. Nolan et al. (1990) already stated that some of components of the Campanian(?) to Maastrichtian Al-Khod Formation from outside the Saih Hatat Dome are derived from pre-Permian formations of the Saih Hatat Dome (Maastrichtian = 72 – 66 Ma). Thus, the Saih Hatat Dome must have undergone massive exhumation phase already during the Late Cretaceous), while falling within 2 sigma of the 97 Ma (± 29 Ma 2 sigma) internal Rb-Sr isochron from Falk & Kelemen (2015) and thus not impossible, is much less likely than the timing of subduction and obduction (97 – 74 Ma; e.g., Searle & Cox, 2002), which is well within 1 sigma (Why 1 sigma? 1 sigma would be ± 58 Ma, thus includes the Upper Jurassic to Oligocene. The 97 ± 29 Ma (2 sigma) age actually does not support that the listwaenite formed during obduction. The error margin includes the Campanian and large parts of the Maastrichtian, and therefore, post-obductional deformation. Therefore, the Rb-Sr isochron age must be removed from the manuscript or you clearly state that Rb-Sr age has a large error margin which also includes large parts of the post-obductional history of the Oman Mountains.

Furthermore, Geologic mapping of the outcrops north and northeast of Site BT1 shows that listvenites are gently folded in a broad antiform. In this structure, they are overlain by banded peridotites and dunites – the Banded Unit – with distinctive major and trace element compositions characteristic of the base of the Samail ophiolite mantle section, and underlain by the metamorphic sole, which in turn overlies highly deformed sediments of the Hawasina Formation (see Kelemen et al., preprint available for reading and commenting on ESSOAR). These structural characteristics indicate that the listvenites formed within or just above a shallowly dipping fault zone along the interface between the ophiolite, the metamorphic sole, and meta-sediments beneath the ophiolite. This geometry is inconsistent with listvenite formation along steep normal / strike-slip faults associated with late extension (**What do you mean with “late” extension? Latest Cretaceous to Paleocene or Eocene or Neogene?**), as proposed by Scharf et al.

I am not sure to which work of Scharf et al., the authors are referring to, but listwaenite does not have to form exclusively on steeply dipping “late” normal faults or strike-slip faults. When I understand the authors correctly, then they interpret the described shallowly dipping fault zone as a thrust, which formed during obduction. I challenge this interpretation and claim that the shallowly dipping fault zone was the thrust, which was reactivated as a low-angle extensional fault. It is common that extensional shear zones are at an early stage low-angle faults, while they become progressively steeper dipping in a later stage (here an example from the Alps; Scharf et al., 2013).

Scharf, A., Handy, M.R., Favaro, S., Schmid, S.M. & Bertrand, A. (2013) Modes of orogen-parallel stretching and extensional exhumation of thickening orogenic crust in response to microplate indentation and slab roll-back (Tauern Window, Eastern Alps). *International Journal of Earth Sciences*, 102(6), 1627-1654, doi: 10.1007/s00531-013-0894-4. Springer

If the authors wish to underpin their interpretation, they must present structural data of the shallowly dipping fault zone. These data must demonstrate that shearing ensued during shortening (obduction, thrusting), and listwaenite was also sheared during thrusting. Otherwise, the interpretation with the shallowly dipping fault zone = obduction/thrusting is pointless. In both cases (thrust or extensional fault) the shear sense is similar towards the SW. Extensional shearing would be related to post-obductional doming of the Saih Hatat and/or Fanja Saddle, starting at a late stage or immediately after the emplacement of the ophiolite.

Moreover, Menzel et al. (2020, JGR) have shown that high-angle normal and strike-slip faulting is common in listvenite, but occurred later than the carbonation reaction.

As I wrote above, listwaenite does not have to form exclusively in brittle “late” normal faults or strike-slip faults. Listwaenite may formed within low-angle extensional shear zones. The brittle faults/cataclasites studied by Menzel et al. (2020) are not dated. Thus, these structures can be “very young” and related to late Cenozoic deformation. See Mattern et al. (2021) and Scharf et al. (2021) for an explanations of Oligocene/Miocene deformation in the Fanja area.

Mattern, F., Bolhar, R., Scharf, A., Scharf, K., Mattern, P. & Callegari, I. (2021) Novelty discovered post-mid-Eocene sinistral slip in the eastern Oman Mountains: widely distributed shear with wrench-fault assemblage related to Arabia-India convergence. *VEGU 2021-8321*, Vienna, Austria.

Scharf, A., Mattern, F., Bolhar, R., Callegari, I., Mattern, F. & Ring, U. (2021) Oligocene/early Miocene major E/W-shortening and NW-oriented sinistral slip with wrench-fault assemblage in the Oman Mountains related to oblique Arabia-India convergence. Earth and Space Science Open Archive, doi: 10.1002/essoar.10508453.1.

An extensional setting that we indeed cannot fully exclude is CO₂-fluid flow during potential reactivation of the basal thrust as extensional, low-angle detachments.

However, to date there is no evidence for this from the listvenites (but also not for thrust setting, isn't it?). We note further that while the basal thrust of the Oman Ophiolite accommodated > 100 km contractional displacement, the displacement related to a (possible but to date not well documented) extensional reactivation of the thrust is much less. So what? What is the message of this sentence?

For all these reasons, we are of the opinion that a subduction/obduction setting for listvenite formation is most probable, and most consistent with the currently available data (please keep your interpretation open for an extensional setting for listwaenite formation, as stated above). We modified the discussion (lines 377 – 395) to clarify this (although, due to text length limitations, in a shorter way).

Line 390-394: See my comments above. The “argument” with the gently folded low-angle contact does not confirm that the listwaenite formed during convergence. I am actually surprised that the authors consider this low-angle contact as an argument against post-obductional origin of the listwaenite.

For further discussion (since this is not the main topic of this paper), we would like to refer the reviewer to a recent paper by Kelemen et al. (preprint available for reading and commenting on ESSOAR) where this is discussed in more depth.

In our opinion our study also provides an important step: we clearly defined which microstructures are synchronous with listvenite formation as opposed to later overprints, creating a framework for future attempts of structural reconstruction or dating. Our results further underline that careful consideration of the complex (micro)structural evolution is key to obtain meaningful and reliable results during such reconstructions.

The authors interpret their data in a complex and speculative way, just to fit the ‘subduction zone model’, while the alternative model (extension) could easily explain their findings.

We think that the discussed processes and proposed mechanisms, which are based on our microstructural observations, are applicable to listvenite formation regardless of whether that occurred in a contractional or extensional tectonic setting. Where discussing the geodynamic context and implications for subduction zones (Lines 377 - 404), we provide now a clearer, explicit statement of why we consider a thrust setting more likely and better supported.

See my comments above.

Here is what they wrote:

“Because of the volume increase of serpentinization, under external stress this is likely to cause cyclic variations in permeability, pore pressure and differential stress, which may induce fracturing and the formation of serpentine and early carbonate veins. The formation of listvenite may intensify this process due to its lower permeability and higher strength compared to serpentinite, causing dilatancy by granular flow and reaction-assisted ductile deformation along the reacting lithological boundary. We speculate that this feedback of external stress, changing rheology and high pore pressure helps to

facilitate continued reaction to listvenite despite volume increase, as long as CO₂ supply is sufficiently high.”

Why such a complex speculation about changing rheology, high pore pressure and sufficient CO₂ supply? Massive and ongoing solid-state volume increase reaction of ultramafic rocks to serpentinite and listwaenite can be easily explained during an extensional event where fractures open, allowing for fluid transfer. This kind of extension would not occur within a subduction zone.

Here we would like to respectfully disagree with the reviewer. At the depths considered for Listvenite formation in Oman (What is the depth considered for listwaenite formation? When I read again the pressure estimates during listwaenite formation in the resubmitted article, then I must say that the pressure is not constrained at all. Falk and Kelemen (2015) assumed that the listwaenite formed within a subduction zone. They further used respective geothermal gradients to propose the pressure. This approach to “constrain” the pressure is weak. The authors ignored that listwaenite can have formed during post-obductional extension and within the upper crust, as proposed by, i.e., Wilde et al. (2002). Thus, applying the same logic as before, the pressure of listwaenite formation would be considerably less as proposed by Falk and Kelemen (2015).), all the three principal total stresses are compressive in both contractional and extensional tectonics (would your statement be correct if one assumes an upper crustal setting, instead within the mantle, during of listwaenite formation?). Therefore, high fluid pressures are required for dilatancy. It is unfortunate that in tectonics the names "contractional" and "extensional" (these are used to denote displacement directions) are often misunderstood to mean compressive and tensile stress. In our opinion, the processes proposed in our manuscript can occur in both contractional and extensional tectonic settings. Therefore, our analysis of the complex, "ongoing solid-state volume increase reaction of ultramafic rocks to serpentinite and listwaenite" is applicable for both settings, and cannot be easily explained by an "extensional event".

To sum up the validity of the submitted manuscript, the authors provide detailed geochemical/analytical and microstructural results, but the link to the geodynamics setting of listwaenite formation is not at all conclusive, especially since alternative models for listwaenite formation are left unmentioned and unconsidered.

As mentioned above, the aim and main topic of this paper was to investigate the relationship between deformation, reactive fluid flux, and listvenite formation, irrespective of the tectonic setting where this occurred. All three reviewers agreed with the validity of our analytical and microstructural results and their interpretation. (Ok. Then I would recommend to have the manuscript until this point and exclude any speculation about the tectonic setting during listwaenite formation).

Although outside of the main goal of our study, we have revised the manuscript to include a lengthier discussion around possible tectonic settings. We add to the discussion: “A well-suited fluid source are carbonate-bearing sediments rich in hydrous silicates that undergo devolatilization related to heating, for example due to contact metamorphism or burial during thrusting ^{6, 40, 50}. In the Samail ophiolite, burial of the Hawasina nappes and underlying units below the ophiolite caused fluid flux and the formation of overpressure cells ³⁵, which, combined with active deformation along the basal thrust, might have created the circumstances necessary for listvenite formation. CO₂-bearing fluid= may have been sourced from deeper, partially subducted sediments, where temperatures are higher and devolatilization reactions more efficient ^{33, 40}. Alternatively, it has been proposed that listvenite formation may have coincided with extensional deformation after obduction of the Samail ophiolite ^{38, 39}. However, the parallel alignment of listvenite bands around Site BT1 with the gently folded, low-angle contacts between the basal mantle section of the ophiolite, the metamorphic sole and the Hawasina sediments ³³ indicates that the listvenites formed along the low angle, basal thrust of the ophiolite, and

not along steep faults during later extension. This geometry, isotope geochemistry ⁴⁰, and the isochron age ⁸ indicate that the most likely setting of listvenite formation at Site BT1 is subduction-obduction.” See my comments above for the validity of some of your “arguments” about the tectonic setting. I cannot comment on the geochemical statements, because I am not a geochemist.

The work describes detailed microstructural analysis of listwaenite and listwaenitisation. This is interesting for specialists working on this matter (in geochemistry and fluid-rock interaction processes) but its significance to the larger community is limited.

We do not agree with this statement; we believe that this paper presents exciting new results that significantly improve our understanding of the mechanisms of coupled deformation and fluid-rock interaction in mantle rocks, which is of high importance for anyone studying reactive fluid flux and carbonation processes (but this is what I also have written above). The implications of our findings reach far beyond the regional context of the Fanjah region of Oman with direct impact on CCSU research and application in mafic and ultramafic rocks. I do not know what “CCSU” is. I did not state that this research is only interesting for the regional geology of Oman. Thus, I do not understand why the authors wrote this comment.

A discussion of the observed structures with respect to tectonic events would also increase the significance of this manuscript (i.e., the described S-C fabric, boudin and folds; presence of kinematic indicators for events during which these structures formed?). Even though, I am not a specialist in the microanalytical field, I am confident that the described features exist, and that the used methods are of high-quality, but I have to seriously criticize the authors for not indicating in the Methods section what kind of well material was at their disposal. There is no indication whether they worked on an oriented well core or not. In case they did, the fold vergences, shear sense should be provided with cardinal directions – critical in support of the subduction or extension interpretation. The lack of orientation/direction information is another main weakness of the manuscript.

A lack of orientation data of our samples from the inclined drilled Hole BT1B, the difficulties to identify and measure small-scale structures macroscopically in the field, and the ubiquitous occurrence of cataclasites and faults overprinting the listvenites after carbonation (see Menzel et al., 2020 JGR), where rotation of different fault blocks is likely, do not allow a confident correlation of orientations. Therefore, our data do not allow a more detailed structural geologic interpretation: this is not the aim of our paper. (Correct but important for your interpretation: origin of the listwaenite - obduction vs. extension). We do note that an extensive and detailed report of the measurements and analyses has been fully published and accessible in detail and with full transparency in the cited Proceedings of the Oman Drilling Project (Kelemen et al. 2020) (Was this work peer-reviewed?). Per the reviewer’s suggestion, we have expanded the explanation of the nature of the samples and orientation data available to us in the Methods section.

As mentioned above, the models for the listwaenite formation are not discussed. Thus, specific references are missing. I have indicated those into the detailed text to the reviewers. Furthermore, some references to the geological map and cross section are missing. I have added them too.

We have added appropriate referencing where necessary.

See my comments about referencing above.

The authors also missed that ductile-brittle deformation in listwaenite of the greater Fanja area has been described already. I have mentioned the respective reference below.

Here we respectfully disagree: we present here for the first time detailed microstructural analysis and, based on this data, an exciting new model for Listvenite formation, with high relevance for our understanding of fluid-rock interactions. The conference abstract the reviewer refers to describes field structural relations of brittle and semi-ductile deformation in listvenite. But unfortunately, the actual data have not been published, so we cannot compare our results in detail, and cannot evaluate if the described structures are synchronous to listvenite formation (such as the folding of veins described in this paper) or if they are related to tectonic overprints after carbonation (similar to the cataclasites and faults described by Menzel et al., 2020 JGR).

Finally, the manuscript reveals a limited familiarity of the authors with the regional geology of Oman, and the respective literature. Text, figures and literature references must be improved accordingly. We respectfully disagree. J.L. Urai and P. Kelemen have investigated different parts of the geology and tectonics of Oman over several decades, with more than 50 publications in high-ranked peer-reviewed journals.

If the authors do not mention (in the first version of the manuscript) that a long-known alternative model for listwaenite formation is post-obductional, then the authors missed a substantial point (including missing references). Furthermore, the mistakes in the geological map and cross section indicate a limited familiarity with the regional geology of Oman (par-autochthonous vs. autochthonous; folded contact of the base Hajar Supergroup at the SW limb of the Saih Hatat; wrong locality of Muscat (which is still wrong in the revised version of the manuscript – Muscat is 10 km to the E of the present location and within the ophiolite). These (in my view) substantial mistakes should not be made by anyone who is fully familiar with the regional geology of Oman.

For the reasons stated above, I cannot recommend publication of this manuscript.
Andreas Scharf, Muscat, June, 2nd 2021.

Line 84: The authors write that the pressure of listwaenite formation is poorly constrained with a possible range of ca. 0.3 to 1.2 GPa (Falk and Kelemen, 2015). Actually, the pressure is not constrained at all. Falk and Kelemen (2015) assumed that the listwaenite formed in a subduction zone and calculated the respective pressure based on the overburden of the oceanic lithosphere. They also used literature data from the metamorphic sole (which has seen up to 900°C – unlike the listwaenite with <250°C). For a fair evaluation of the pressure during listwaenite formation the authors must acknowledge that the pressure would have been significantly less than 0.3 GPa if one assumes that the listwaenite forms post-obductional and during upper crustal extension as proposed by, e.g., Wilde et al. (2002).

Line 87: Here the referencing is incorrect. Falk and Kelemen (2015) did not provide the 2 sigma error. It cannot be found in this article. The authors mention below that the 2 sigma error is based on personal communication with the first author of Falk and Kelemen (2015). Thus, the authors have to clearly mention this in the text. This is an important detail.

Line 96: “meta-sediments”. Are these rocks really metamorphosed? In the field, the Hawasina sediments are highly folded but appear to be unmetamorphosed. Furthermore, Aldega et al. (2021) reported peak-temperatures for the Hawasina rocks of ca. 200°C or less. Thus, I doubt that the Hawasina rocks are meta-sediments.

Aldega, L., Carminati, E., Scharf, A. & Mattern, F. (2021) Thermal maturity of the Hawasina units and origin of the Batinah Mélange (Oman Mountains): insights from clay minerals. *Marine and Petroleum Geology* 133, 105316, doi: 10.1016/j.marpetgeo.2021.105216

Following reviewer #1's comments and recommendations, we have made significant modifications to the manuscript and to Figure 1 (see also replies to detailed comments below), and hope that we have addressed all issues appropriately, providing an improved revised version of the paper.

Again, the authors produce a technically well-written manuscript. However, the interpretation of a subduction-setting during listwaenite formation is not supported by data and biased (see my detailed comments). This includes that the authors use their own citations/references which are not peer-reviewed to support their interpretation. At the same time, the authors exclude published abstracts with a different model.

The tectonic-setting issue is not the main topic of this manuscript, I agree. However, it is an important section of the interpretation. I would recommend to fully exclude any discussion about the tectonic setting during listwaenite formation because the authors cannot present any structural evidence supporting their hypothesis (see my comments above). The tectonic discussion as it stands now in the manuscript is biased and opinion-driven.

Replies to detailed comments by Reviewer 1:

“(see Jackson, 2005 for spelling of listwaenite)”

Several forms of spelling of the word listwaenite can be found in literature, just as different spellings of Samail / Semail / Sumail exist. We prefer “listwaenite” because it is closest to the spelling of the type locality in the Ural mountains, derived from kyrillic as argued by Halls & Zhao 1995.

Abstract

Line 2. “Oman Ophiolite” is misleading. There are two different ophiolites in Oman. I would be specific (“Semail” or “Samail” or “Sumail” Ophiolite – the spelling of “Samail” is different in the literature). Do this for the entire manuscript and art work.

“Oman ophiolite” has been used in papers by pioneers of the ophiolite research such as Nicolas and Boudier; but we recognize that there is a second ophiolite in Oman so Samail Ophiolite is more accurate. We have changed “Oman ophiolite” for “Smail ophiolite” throughout the manuscript and figures.

Line 2. Several. Could you be more specific? Do you have a value?

Line 3. What is “GT”? Do you mean “Gt” (gigatonne)?

2- 3 Gt.

Line 6. Tectonic deformation. This is the key, but what kind of deformation (i.e., compressional, extensional?). Be specific!

See reply to major comment

Line 6. Folded veins. Here you should provide information of the folds. Are they open, isoclinal, harmonic, ptygmatic? Orientation of the fold axes, axial planes? Do you observe several folds with the same orientation?

The word count of the abstract is too limited to include these details here (those are, when possible, described in the results section. See also reply to comment about orientations).

At which temperature does magnesite grow or become folded/ductile? This would provide information about the conditions during the respective deformation event.

There only are deformation experiments for $T > 500$ °C, no data exists for these low temperature conditions.

Line 11. I am not sure how the lithostatic pore pressure can promote a dilatant porosity, especially within a subduction zone (I can imagine that pre-existing pores could be flattened in a subduction zone due to simple shear at an early stage). Generally, I would rather expect the contrary. In a subduction zone, any potential pores will be closed, considering the lithostatic pressure. Why do you not consider CO₂ transport during extension instead within a subduction zone?

High pore pressures can lead to tensile fracturing also in non-extensional conditions. In Oman, this has been shown for carbonate veins in (par)autochthonous units in response to burial below the ophiolite and the Hawasina nappes, which created fluid overpressure cells (Grobe et al., 2019, *Solid Earth*, 10, 149–175). There are well studied examples of compressional shear zones facilitating fluid flux and related alteration, e.g. the Glarus thrust (e.g. Burkhard et al., 1992, *Contributions to Mineralogy and Petrology* 112, 293-31) or thrusts in the Pyrenees (e.g. Muñoz-López et al., 2020, *Geofluids*), as well as numerous subduction melanges (e.g. Angiboust et al., 2014, *J Petr.*). See also Sibson, 2017 (*Earth, Planets and Space* 69:113). To clarify: the mantle rocks form the upper plate, and they started as essentially zero-porosity peridotites that became more porous and permeable only due to hydration. This is a very different situation than flattening and closure of pores that would occur in a subducted slab. That said, please note that we do not limit our model to compressional settings such as subduction zones, and that also in extensional, (semi)ductile shear zones, creation of fully pervasive porosity is by no means guaranteed.

7

If your assumption is correct, i.e., dilatant porosity during subduction, then listwaenite must be common in every subduction zone of the world and everywhere distributed at the base of the Semail Ophiolite. We do not agree with this comment. There are other requirements for a listvenite to form apart from porosity: CO₂ concentration in aqueous fluid must be very high and CO₂ flux must be sustained, also, high / lithostatic pore pressures need to be repeatedly renewed by fluid influx to maintain this process; all these factors need to play together to allow listvenite formation.

However, listwaenite occurs in the Fanja area only with few exceptions near the Hawasina Window and in the UAE. In the Fanja area, listwaenite occurs in extensional fault zones.

There are serious problems with your statement.

That extensional and strike slip faults cut listvenites and surrounding rocks in the Fanjah area (e.g. Scharf et al., 2019 *Tectonics*) is not proof that listvenite formed during that stage of tectonic deformation.

(Correct. However this is not stated in Scharf et al., 2019. This is what it says in Scharf et al., 2019 “In addition, there is evidence of a ductile shear zone and brittle faults with slickenlines at Wadi Mansah (stop 12 in Figure 8; Table S1) within listwaenite rocks at the base of the Semail Ophiolite. The faults strike parallel to Wadi Mansah with 110–120° and dip of ~75° toward the SW and reveal either extension or oblique sinistral motion based on a large sigma object and mineral steps. Sinistral transtensional faults in Wadi Mansah have been described by Bailey et al. (2019).” Thus, I do not understand the criticism).

Crosscutting microstructures of sharp faults actually indicate that they were

active after listvenite was formed, and were often related to carbonate dissolution, not carbonation (see Menzel et al., 2020 JGR) (Menzel et al. worked at one location, I would be careful to generalize the statement of Menzel et al. for the entire Fanja area. Mattern et al., 2021 and Scharf et al., 2021 provide a reason for the formation of the “late” microstructures”. See above). Still, as outlined above, we do not exclude the hypothesis that listvenite may have formed during early reactivation of the thrust as extensional shear zones, but we consider it unlikely.

Introduction

Line 16. Do you have a reference for your statement?

Statement removed

Line 21. The topic with the “leading edge of the mantle wedge” is a discussed matter in literature. I do not see any geological or structural convincing evidence that your case stated is correct. It is common practice to mention an established alternative (Listwaenite does not form within a subduction zone but postobductionally during extension, i.e., Stanger, 1985; Wilde et al., 2002; Nasir et al., 2006; Mattern and Scharf, 2018; Scharf et al., 2019, 2020).

Mattern, F. & Scharf, A. (2018) Postobductional extension along and within the Frontal Range of the Eastern Oman Mountains. *Journal of Asian Earth Sciences* 154, 369-385, doi: 10.1016/j.jseaes.2017.12.031.

Scharf, A., Mattern, F., Bolhar, R., Bailey, C.M. & Ring, U. (2019) Early Cenozoic listwaenite formation at a major extensional fault zone of the Oman Mountains (Fanja area) – insights from structural analysis and UPb carbonate dating. AAPG Event, Structural styles in the Middle East, 9th-11th December 2019, Muscat, Oman.

Scharf, A., Mattern, F., Bolhar, R., Bailey, C.M. & Ring, U. (2020) U-Pb dating of postobductional carbonate veins in listwaenite of the Oman Mountains. International Conference on Ophiolites and the Oceanic Lithosphere: Results of the Oman Drilling Project and Related Research, 12-14th January, 2020, Sultan Qaboos University, Muscat, Sultanate of Oman.

Stanger, G., 1985, Silicified serpentinite in the Semail nappe of Oman: *Lithos*, v. 18, p. 13-22.

Wilde, A., Simpson, L., and Hanna, S., 2002, Preliminary study of Cenozoic hydrothermal alteration and platinum deposition in the Oman Ophiolite, in Jessell, M. J., ed., *General Contributions: Journal of the Virtual Explorer*, v. 6, p. 7-13.

We have expanded our section on the geological background to mention these alternative models. Line 87: “An internal Rb-Sr isochron with an imprecise age of 97 ± 29 Ma (2sigma) ⁸ (Here the referencing is incorrect. Falk and Kelemen (2015) did not provide the 2 sigma error. It cannot be found in this article. The authors mention that the 2 sigma error is based on personal communication with the first author of Falk and Kelemen (2015). Thus, you also have to clearly mention this in your text. This is an important detail.) suggests that listvenite formation was concurrent with subduction of the Arabian margin ³³, ophiolite obduction (97 – 74 Ma; e.g., 37 – a picky detail: the Arabian margin was not subducted at 97 Ma. At 96 Ma or so, the oceanic lithosphere of the Neo-Tethys Ocean was subducted. Subduction of the Arabian margin, i.e., Hawasina Basin, ensued sometime after 96 Ma. The subduction at ca. 96 Ma took place at an intraoceanic subduction zone at the beginning – subduction of oceanic lithosphere and not continental lithosphere. Obduction started only when the upper plate thrust above continental rocks), or a very early phase of post-obduction extension ^{35, 38, 39.}” (Why the very early phase? Post-

obductional extension started immediately after the emplacement of the ophiolite or even in part overlapping in time. The emplacement was completed at 74 Ma or so). We believe this is an objective description of the different possible settings. We further provide a short discussion about the geodynamic setting, including the hypothesis of carbonation during extensional tectonics, and explaining why a subduction-obduction setting is more probable (line 389). However, as that is not the aim of this paper, we do not go into an indepth discussion about the geodynamic setting of listvenite formation.

I consider the text about the tectonic setting as biased and opinion-derived. See my comments above.

Line 29. This sentence is hard to read/understand. Could you please rephrase it and produce two nice sentences?

Done

Line 65: I am not sure if it is just "listvenite" (not "listvenites").

Line 69: The last sentence is interpretation and does not belong to the introduction. Thus, skip this sentence.

It is common practice to provide the reader with a key conclusion at the end of the introduction, to increase readability and impact of the paper.

Results

Line 75: mostly at the base. There are examples where the listwaenite occurs in a distance of ca. 2.5 km (in map view) from the base of the ophiolite (i.e., near Buwah). You are showing it in your Fig. 1a.

Changed to: "in serpentinite at or close to the base of the Samail ophiolite"

Just a comment about this particular listwaenite outcrop. Can the authors explain how this large listwaenite (within the ophiolite and 2 km (in map view) away from the "basal thrust") formed during thrusting?

Line 88: Could you provide the error margin (Sigma 1 or Sigma 2) for the Rb-Sr isochron age? I assume that it is Sigma 1, because no details about the error margin are provided in the mentioned reference (at least I could not find any). However, one of the co-authors of the submitted manuscript is a co-author of the provided reference. Thus, this person must know the missing information. The error margin is just gigantic (± 29 Ma Sigma 1; ± 58 Ma Sigma 2).

We contacted the first author of the paper in question (L. Falk) and she confirmed that the ± 29 Ma error margin is indeed sigma 2. Of course, the error is still large, which is why we added the word "imprecise" in the text.

Line 90: Please provide the obvious alternative model that the listwaenite formed after the obduction was completed (e.g., Stranger, 1985; Wilde et al., 2002; Nasir et al., 2006; Mattern & Scharf, 2018; Scharf et al., 2019, 2020). I wonder why this important aspect is not mentioned in the text.

We modify this line to: "An internal Rb-Sr isochron with an imprecise age of 97 ± 29 Ma (2sigma) ⁸ suggests that listvenite formation was concurrent with subduction of the Arabian margin ³³, ophiolite obduction (97 – 74 Ma; e.g., 37), or a very early phase of post-obduction extension ^{35, 38, 39}." See reply to previous comment.

Please make a correct referenc at this place. Indicate that the 2 sigma error is not documented in Falk and Kelemen (2015). The 2 sigma error was communicated personally.

Line 90: In the Hawasina rocks underlying the ophiolite in the Fanja area are hardly any carbonate-bearing rocks. Actually, the geological map of Fanja indicates that mostly the Al-Jil Formation (Aj1Mbs) crops out in the Fanja area. This formation consists of radiolarian chert, siltstone and sandstone. There are some minor outcrops of other Hawasina formations exposed (Aj1Vb – basaltic pillow lava; Aj1-2 – flaggy limestone, radiolarian chert and sandstone, and UmC – red radiolarian chert and micritic limestone). Thus, the statement is not supported by evidence. The obvious alternative would be that the source for the carbon is from the Arabian shelf (carbonates), cropping out all over in the area.

First of all, due to abundant tectonic overprint (e.g. Menzel et al. 2020), it is quite clear that the presentday configuration is not the same as it was during listvenite formation. But more importantly, Sr and C isotope signatures of sediments of the Hawasina nappes do provide evidence as they are consistent with those of magnesite in the BT1 listvenite. Furthermore, listvenite formation does not only require a source rock that is rich in carbon, but a rock that will produce aqueous fluids that have a high enough CO₂ concentration (Please note that during the Cenozoic, the Fanja area was below sea level which is evident by the occurrence of shallow marine carbonates. Thus, aqueous fluids were present after obduction. Did you check the Hajar Supergroup carbonates, i.e., Mahil and Saiq Formation for their Sr and C isotope signature? Could they also fit with the magnesite in BT1?). Low temperature hydrothermal fluids that percolate limestones do not meet this requirement (Do you have a reference for this? What do you mean with “low-temperature”? The authors suggest a temperature of listwaenite formation which is <250°C.). On the other hand, formation of such CO₂-enriched fluids is thermodynamically attainable for sediments that contain carbonate-silicate mixtures if these are heated to (at least) low-grade metamorphic conditions (What do you mean with low-grade metamorphic conditions? Do you mean greenschist-facies conditions? If so, please provide P-T values for Hawasina rocks from the Fanja area. In the field, these rocks appear to be unmetamorphosed. Please note that the Saiq Formation is not a pure carbonate formation in the Saih Hatat area. Here, this formation consist of an alternation of siliciclastics, carbonates and volcanic rocks). We address this in the discussion, and further details about this background can be found in a number of papers that concern listvenite formation in general, for example Klein & Garrido 2011 and Menzel et al., 2018, and, with respect to listvenites in Oman, Falk & Kelemen 2015, Kelemen et al (preprint) (all references are cited in the paper).

Putting it another way: if the shelf carbonates were the source, why are there not listvenites all around the Jebel Akhdar and Saih Hatat, where carbonate rocks are very abundant below the ophiolite (as are faults and detachments)? (Because firstly, normal faults which juxtapose mantle rocks and shelf carbonates are largely missing – i.e., at the southern part of the Jabal Akhdar Dome, and secondly large parts of the Oman Mountains were submerged during the post-obductional history, thus, no or only limited fluids were available to form listwaenite). The answer is that there are several requirements that need to be met: i) a source rock that has a suitable composition to produce CO₂-rich fluid, ii) devolatilization or fluid fluxing of that source rock at conditions that produce CO₂-rich fluids in high quantities, which in turn requires some kind of heat source, iii) faults, shear zones, a plate interface or another structure along which these fluids can be channelled into peridotite. We find that underthrusting of mixed, carbonate bearing marine-siliciclastic sediment rock below the ophiolite (a configuration that is equivalent to a shallow subduction zone) is a very viable scenario where these requirements can easily be met, because the subsidence causes heating of the sediments upon equilibration of the geotherm, allowing for devolatilization to occur. (I wonder if the requirements are also met within the upper continental crust. I am not sure what the authors mean with “kind of heat source” above (i.e., what is the temperature range of this heat source). I just want to point out that serpentinization is an exothermal reaction, which is a possible low “heat source”.) High CO₂ concentrations in fluid are even easier to attain at greater depth, and isotope geochemical data suggests that deeper sourced fluids from the subduction zone are a viable scenario (see de Obeso et al., preprint

available for reading and commenting on ESSOAR). Fluid flux during early extensional reactivation of the thrust would also be consistent with these considerations, but, as stated previously, we consider it unlikely. However, a simple “obvious” limestone CO₂ source on the other hand is not well thought through in our opinion.

Line 95: You could mention that two dolomite veins cutting the listvenite from the Fanja area have an age of ca. 60 Ma (Scharf et al., 2020).

The referred work reporting the U/Pb dates is a conference abstract. Because we cannot find a published paper, preprint or full online resource that contains this dataset, unfortunately we cannot fully evaluate and cite it.

See my comments about this attitude above.

Line 114: Could you provide any information about the fold, please (i.e., style, interlimb angle, orientation of the fold axis and axial plane, fold vergence?). How many folds did you observe and are they all consistent in terms of style and orientation? Did you observe folded listwaenite at the surface near the well? Brittle to ductile deformation in listwaenite has been reported already (Scharf et al., 2020b). The location is at the northern margin of the Saih Hatat.

Scharf, A., Mattern, F. & Mattern, P. (2020) Ductile-brittle shear zone in a listwaenite body within the Frontal Range Fault of the Oman Mountains (Sultanate of Oman). EGU 2020-12672, Vienna, Austria.

Line 116: S-C fabric. Interesting. Could you provide an assumed shear sense? Is the shear sense consistent for several S-C fabrics? In which direction did shearing ensue? I am afraid that you do not have any oriented samples. Thus, you cannot make any statement to shear sense and its relationship to tectonic events. Am I correct?

Line 137: Did you observe any shear sense within the shear zones? How do you know that it is a shear zone?

Line 160: Do you have any shear sense for the sigma-clasts?

Line 160: “with a sense of shear consistent...” What is the shear sense?

Line 174: Also, here what is the shear sense? What is the orientation of the fold axis and axial plane? What is the style/shape of the folds? Do all the folds have a similar orientation and style?

WRT (WRT - What does it mean?) all above comments: as stated in Methods, the samples are oriented only in the arbitrary spatial reference frame of Hole BT1B, which is not easily translatable into cardinal directions due to its inclination and the non-reconstructable orientation of discontinuous core sections (as is also described in detail in ref. 32). Therefore, fold orientation and vergence, orientation of apparent s-c fabrics and shear senses from sigma clasts cannot be unambiguously attributed and related to specific tectonic events and we cannot determine whether they formed due to thrusting or extensional shear. We modified the methods section “Samples” to clarify this better.

Line 138: Do the elongated magnesite ellipsoids have a preferred orientation? The same for the aligned magnesite dendrites?

Yes, see a few lines further down: “In 3D, micro-CT shows that magnesite ellipsoids are oblate in the foliation plane”

Line 147: Could you determine the orientation of the short and long axes of the boudin?
No, we only have the apparent orientation in thin section plane here.

Line 202: Please provide the meaning of all of your abbreviations such as “TEM” and “STEM” at the first appearance in the text.
Done

Discussion

First section: According to the detailed microstructural observations, I would suggest that the listvenite formed within brittle to ductile shear zones. Such or a similar statement is missing in the first section.
We prefer to not add such a statement in this paragraph; it is discussed further below.

Line 255: “few kilometers”. I have not read the provided literature (45-47) in detail, but I would check if it is really “few kilometers” and not “few hundreds of meters”.
Thanks for pointing this out; we corrected the mistake.

Line 303: red dot.

Line 303: Systematically oriented structures including folds may indicate that the deformation is related to “external stress”.

This comment is not clear to us. Does the reviewer mean “tectonic stress”? As mentioned above, due to a lack of appropriate orientation and macrostructural data we are not in a position to make inferences about the consistency and orientation of stresses that gave place to the formation of the observed structures.

Line” 387: This lithological boundary would be also the locality for a development of a shear zone. This shear zone would also explain why “the listvenite consist of a few major bands of 10s of meter thickness”.
We add that this boundary (which at the time of carbonation was the reaction front) also likely focused deformation.

Line 405: I am not convinced of your statement and I cannot understand your point of this sentence. What do you mean with “cyclic variations in permeability”? The permeability is just close to zero. A dramatic volume increase will just block any further reaction. Further reaction will only occur if external tensional stress is present, which will result in opening of fractures.
We modified this paragraph to make it clearer. Because complete serpentinization obviously did occur in the serpentinites, permeability was not always close to zero. Furthermore, high-resolution images indicate that some level of porosity/permeability is preserved within both serpentinites and listvenites (Figure 7). If the build-up of pore pressure is high enough, tensional fracturing may occur (also if the external stress field is not tensional), allowing for a pulse of fluid flux. Sealing due to volume expansion of serpentinization will disrupt fluid flux again, until pore pressures increase again, thus creating cyclic variations in permeability. We believe that with the modifications it should be clear.
Ok. I now understand your “cyclic variations in permeability” statement. Thanks for the clarification.

Line 410: I cannot follow your speculation. The speculation is not supported by your data. Why do you not consider that the listwaenite and related reactions formed during an extensional stress regime

instead within a subduction zone? This would easily explain your observations without needing to construct such a highly speculative statement.

See reply to previous comment. The proposed mechanism is not exclusive to or implicit of subduction zones, but just as well applicable to ductile/semi-brittle extensional shear zones. The arguments and reasoning for it are elaborated in the preceding paragraphs.

Line 414: For the same reason as above, I cannot follow and agree with this statement.
We modified this paragraph to make it clearer

Methods:

Line 422: What is missing is a clear statement whether the samples are oriented or not. This is an important point and must be included. I wonder why the authors did not mention this in the text.

We realize that referring to the arbitrary core reference frame of Hole BT1 is not self-explanatory for readers who are not familiar with IODP core logging standards. We add: “ Hole BT1B (International Continental Drilling Project Expedition 5057-4B) was drilled with an inclination of 75°. Therefore, and because the drill cores contain discontinuities across which the orientation of single core sections could not be restored ³², orientations of structures are unfortunately not easily comparable between different parts of the core and the field.”

Figure 1.

a)

Muscat is depicted at a wrong location. Muscat is further to the east. The term “Jebel Akhdar” is not good visible because of the intersection with black lines. Please remove the “Jebel Akhdar” into the light blue rocks and reduce the faults, that there is no intersection anymore.

Figure 1 has been improved to account for the reviewer’s comments/corrections.

Please change the location of Muscat. It is still at the wrong place.

You have drawn a major extensional fault near the well in your section. This fault is not shown on the map. It is the Wadi Mansah Fault Zone of Bailey et al. (2019) and Scharf et al. (2019b). This fault zone has also a sinistral component, which you also could include in the map and section. Some of your “Hawasina” outcrops to the NNE of BT1 do not exist in the geological map of Fanja. I do not know where you got them from. Please indicate!

Still, the Hawasina outcrops (which I mentioned above) are there. It should be Muti Formation.

Bailey, C.M., Hurtado, C., Scharf, A. & Mattern, F. (2019) Structural controls on listwaenite genesis in the Semail Ophiolite, Northern Oman. EGU 2019-5794, Vienna, Austria.

Scharf, A., Mattern, F., Moraetis, D., Callegari, I. & Weidle, C. (2019b) Postobductional kinematic evolution and geomorphology of a major regional structure – the Semail Gap Fault Zone (Oman Mountains). Tectonics 38(8), 2756-2778, doi: 10.1029/2019TC005588.

Please provide a reference for your geological map.

The reference of the map source (Béchenec et al, 1992) has been indicated in the figure caption, but we slightly modify where it is placed to make it more clear. Other than simplifying and grouping units and plotting only selected faults for the schematic view, we did not modify the mapped region. As this map is a revision done by Béchenec and others of the earlier Fanja NF40-3F (which dates to 1982-1985) and neighbouring maps, we used this source.

b)

Why is the contact of the “Paleozoic basement” with the “Mesozoic” folded (in the Saih Hatat Dome)? This is not directly related to the study but what you have drawn is incorrect. The pre-Permian rocks of the Saih Hatat Dome comprise a major recumbent fold (compare Miller et al., 2002; see also Hansman et al., 2021 for a review of the deformation in the Saih Hatat Dome).

Corrected. The recumbent fold is schematically indicated by a dotted trace, but this detail is not relevant for the present study.

Miller, J.M., Gray, D.R., Gregory, R.T., 2002. Geometry and significance of internal windows and regional isoclinal folds in northeast Saih Hatat, Sultanate of Oman. *Journal of Structural Geology* 24, 359-386, doi: 10.1016/S0191-8141(01)00061-X.

Hansman, R., Ring, U., Scharf, A., Glodny, J., Wan, B., 2021. Structural architecture and Late Cretaceous exhumation history of the Saih Hatat Dome (Oman), a review based on existing data and semi-restorable cross-sections. *Earth-Science Review* 217, 103595, doi: 10.1016/j.earscrev.2021.103595

The extensional fault at BT1 is much too steeply dipping. Your trace of the section is oblique to the fault. Thus, in your section, the extensional fault must dip with $<60^\circ$. You could also indicate the sinistral horizontal shear sense as described in Scharf et al. (2019b). The vertical displacement at this fault is ~ 7 km (see Scharf et al., 2019b). See also for the listric shape of the large extensional fault Scharf et al. (2019b).

We redraw this fault to make it less steep and to schematically indicate that it is likely listric. However, please note that the cross section is, as stated, schematic, and hence the vertical exaggeration is high and not to scale.

c)

The “Autochthonous Mesozoic sed.” Are actually “Par-autochthonous Permo-Mesozoic sed.”

Corrected

The thickness of the Hawasina nappes in the Fanja area is max. 1 km; see geological map of Fanja. According to the geological map of Fanja, the thickness of the ophiolite is <3 km (not 8-12 km). The thickness of the oceanic crust is <4 km (not 5-7 km).

Please note that the column is a very simplified stratigraphy that serves to give the reader a general context of the Samail ophiolite and the position of BT1 within this idealized stratigraphy (see cited reference). To clarify that thickness of units may be less in the area, we add a statement: “Present-day thickness of units in the area is commonly reduced due to tectonic thinning” to the figure caption. Substantial tectonic thinning was likely also related to abundant cataclasites and faults cutting listvenite (i.e. having formed later than the main carbonation event – see Menzel et al. 2021 JGR), so the accurate thicknesses of units at the moment of listvenite formation is impossible to reconstruct without precise dating of listvenite formation as well as each deformation phase, which is currently not available. A detailed structural and geodynamic reconstruction of the Fanjah region that would additionally be required for that is beyond the scope of this study, that would be a paper in its own right.

Post-obductional sediments (not obduction)

We prefer post-obduction sediments, similar to “post-obduction deformation”

d)

Please provide a scale.

Indications of elevation above sea level for the Wadi bottom and one of the hill summits are now given on the figure.

I would add that the serpentinite is difficult to spot, because it is largely covered by listwaenite scree/talus.

Done

e)

Why has the metamorphic sole two colors? The word “sole” is hard to read.

These are different sub-units within the metamorphic sole (See cited reference)

J and k)

I would add the scale bar values directly into the images. You would save some text in the caption (do this for all the other figures, too).

This is a formatting requirement / journal policy of Nature Communications.

I cannot comment on the other figures because I am not a specialist in microstructures. However, the images look very nice.

Supplementary figures

Fig. S1. All figures need cardinal directions. What is the orientation of the shear zone in “b”? Do you observe any shear sense?

The shown field images were not taken with markers of cardinal directions. Unambiguous shear sense indicators were macroscopically not observed in this shear zone. Such structures are typically not very clear macroscopically due to intense oxidation crusts from weathering of listvenite, while polished surfaces lack the 3D information. Note that also in Hole BT1B listvenite, most observed structures that may provide a sense of shear are difficult to see macroscopically and require microscopic investigation.

Fig. S2.

I assume that the samples are not oriented. Cardinal directions?

Samples are oriented in the (semi-arbitrary) core reference frame (CRF) (see Kelemen et al., 2020), which we indicate on the figure.

Fig. S6. What is the orientation of x, y and z?

Directions x, y, z are those of the BT1B core reference frame (CRF), this is now indicated in the caption of the figure.

Godard M, *et al.* Geochemical Profiles Across the Listvenite-Metamorphic Transition in the Basal Megathrust of the Semail Ophiolite: Results from Drilling at Oman DP Hole BT1B. *Earth and Space Science Open Archive*, 37 (<https://doi.org/10.1002/essoar.10507497.1>).

Is this a peer-reviewed article? Can this be used in the reference list?

Kelemen PB, Matter JM, Teagle DAH, Coggon JA, and the Oman Drilling Project Science Team. Methods and explanatory notes. In: *Proceedings of the Oman Drilling Project* (eds Kelemen PB, Matter JM, Teagle DAH, Coggon JA, et al.). International Ocean Discovery Program (2020).

Is this a peer-reviewed article? Can this be used in the reference list?

De Obeso JC, *et al.* Deep sourced fluids for peridotite carbonation in the shallow mantle wedge of a fossil subduction zone: Sr and C isotope profiles of OmanDP Hole BT1B. *Earth and Space Science Open Archive*, 27 (<https://doi.org/10.1002/essoar.10507483.1>).

Is this a peer-reviewed article? Can this be used in the reference list?

Kotowski AJ, Cloos M, Stockli DF, Orent EB. Structural and thermal evolution of an infant subduction shear zone: Insights from sub-ophiolite metamorphic rocks recovered from Oman Drilling Project Site BT-1B. *Earth and Space Science Open Archive*, 60 (<https://doi.org/10.1002/essoar.10505943.1>).

Is this a peer-reviewed article? Can this be used in the reference list?

Kelemen PB, *et al.* Mass transfer into the leading edge of the mantle wedge: Initial Results from Oman Drilling Project Hole BT1B. *Earth and Space Science Open Archive*, 59 (<https://doi.org/10.1002/essoar.10507370.1>).

Is this a peer-reviewed article? Can this be used in the reference list?

Kelemen PB, Matter JM, Teagle DAH, Coggon JA, and the Oman Drilling Project Science Team. Site BT1: fluid and mass exchange on a subduction zone plate boundary. In: *Proceedings of the Oman Drilling Project* (eds Kelemen PB, Matter JM, Teagle DAH, Coggon JA, et al.). International Ocean Discovery Program (2020).

Is this a peer-reviewed article? Can this be used in the reference list?

We would like to express our thanks to the three reviewers for their time to re-assess the revised manuscript NCOMMS-21-17452A, and for the positive feedback of reviewers 2 and 3, who did not have further comments. We have now carefully considered the comments of reviewer 1 on the revised manuscript, and have re-revised the respective parts of the manuscript. In the following we explain in detail how we addressed the concerns and implemented improvements where appropriate, and the reasons why we disagree with some of the comments.

Comments by Reviewer #1 (Andreas Scharf) and replies by authors

For full transparency, we keep the complete record of discussion here, with

- Reviewer #1 comments on NCOMMS-21-17452 (initial manuscript) in **black**
- Author replies to first revision comments in **blue**
- Reviewer #1 comments on NCOMMS-21-17452A (revised manuscript) in **red**
- Author replies to comments on NCOMMS-21-17452A in **green**. To aid readers with color-blindness we mark our reply paragraphs additionally with „>>“

Summary comment of Reviewer #1 on NCOMMS-21-17452A

Again, the authors produce a technically well-written manuscript. However, the interpretation of a subduction-setting during listwaenite formation is not supported by data and biased (see my detailed comments). This includes that the authors use their own citations/references which are not peerreviewed to support their interpretation. At the same time, the authors exclude published abstracts with a different model. The tectonic-setting issue is not the main topic of this manuscript, I agree. However, it is an important section of the interpretation. I would recommend to fully exclude any discussion about the tectonic setting during listwaenite formation because the authors cannot present any structural evidence supporting their hypothesis (see my comments above). The tectonic discussion as it stands now in the manuscript is biased and opinion-driven.

Author reply to the above comment:

This aspect of the manuscript is about an interpretation of the tectonic setting of listvenite genesis for which there are two models. None of these two can be definitively rejected or proven in our opinion, due to a lack of data. In our manuscript we outline both models and explain why in our opinion one is more likely.

Following the reviewer comments from the first review phase, in the revised manuscript we stated the possibility that listvenites formed during extensional tectonics (including references where this hypothesis is stated), and that we consider a convergent setting more likely due to the reasons given there, which are based on the results in the cited, recently published literature. To prevent misunderstandings and to clarify this a bit more, we modified again the tectonic discussion (line 395 – 401) to include a more explicit statement that a setting of early extensional reactivation of the thrusts is conceivable, and what type of further research would help to better constrain the geodynamic context of listvenite formation. Research that might allow a conclusive answer will be a considerable undertaking in the future, requiring significant effort. Such a study (involving regional tectonics and a targeted thermochronological campaign) is beyond the scope of the current manuscript.

However, we stand with our interpretation, which includes that the currently available data do not allow one of the models to be rejected. In our opinion, we discuss the matters transparently. We disagree with the recommendation to fully exclude any discussion of the tectonic context because we believe it is useful to define the possible geodynamic framework as well as avenues for future research.

With regard to referencing, we follow the policy of Nature publishing, which endorses references to preprints that are in review. All cited preprints were in the peer-review process and fully accessible on preprint servers around the time of first submission; by now all of these articles have been accepted for publication. The bibliographic details are updated accordingly in the revised version. The cited literature in question is directly related to the drill cores from Site BT1 and thus of high relevance for this study. We note that the tectonic evolution and geodynamic context is not the main topic of this paper and that the journal format only permits a limited number of references, which is why we can only refer to selected publications about this topic here. Because conference abstracts typically only contain preliminary results and lack the data necessary to evaluate their significance and quality, we do not cite these here.

Detailed previous (black, with responses in blue) and new comments (red, with responses in green) of Reviewer #1

The authors produced a technically well-written manuscript, which describes in detail different microstructures, microcrystallographic orientations and microanalytical analysis of carbonated ultramafic rocks from the well BT-1 of the Oman Mountains. Deformation is partly ductile. Finally, the authors link the observed data and fluid flow to a feedback mechanism in subduction zones. The microstructural and analytical results of the submitted manuscript are sound. Although I am not an expert in microstructural analysis, the presented results are comprehensible.

We thank the reviewer, who notes that he is not an expert in microstructural analysis, for his comments and are glad to see that he agrees with the main conclusions that we draw from our results, namely that there was fluid-assisted ductile deformation during the carbonation reaction and that this deformation in turn helped to facilitate fluid flux and reaction progress.

My main critical concern is that the authors link their data to a feedback mechanism in subduction zones as the authors fail to establish that the subduction zone environment is applicable. The link between the author's data and the 'subduction zone fluid flow' is neither supported by their data nor by other data (i.e., structural data from the core or surrounding field area – none of them were presented or discussed).

We thank the reviewer for his clear comments and have rewritten the manuscript to make clear that the deformation-assisted carbonation we discuss here derives directly from the observations and data presented in this paper, that it can operate in a range of different tectonic settings, and that, based on the currently available data from recent and past studies, a subduction-obduction setting is in our opinion the most likely configuration of listvenite formation at site BT1. For the latter, we now provide an explicit and consistent reasoning in the manuscript; please find further details also in the following replies.

The authors have to keep in mind that two contradicting models for the listwaenite (see Jackson, 2005 for spelling of listwaenite) formation in Oman exist. Model one was proposed by Falk and Kelemen (2015). This model proposes that the listwaenite formed within the 'leading edge of a subduction zone' at ca. 95 Ma, thus, during obduction of the famous Semail Ophiolite. Falk and Kelemen (2015) proposed this model in part based on a single Rb-Sr isochron age (97 ± 29 Ma) of a Cr-bearing fuchsite from the listwaenite (Note the great error margin, which could represent various processes, for example postobductional extension!). I am referring to this in the detailed comments below. The "subduction model" is not supported by field and structural data of the Fanja area. An older model for the origin of the listwaenite is proposed from Stanger (1985), Wilde et al. (2002), Nasir et al. (2006), and was confirmed by Mattern and Scharf (2018) and Scharf et al. (2019, 2020). This model suggests that the listwaenite formed during extension after the emplacement of the ophiolite. Thus, has absolutely nothing to do with CO₂-fluid transfer within a subduction zone. Some of the authors must be aware of the contradicting models, because the models were discussed at the

Oman Drilling Project and Related Research Conference from the 12-14th January, 2020 at the Sultan Qaboos University, Muscat (Sultanate of Oman). Some of the authors of the submitted manuscript attended this conference.

2

A major flaw of the submitted manuscript is that the authors only explain the 'subduction model' without even mentioning the obvious alternative model (the elephant in the room). This raises my concern as to how serious this manuscript was prepared by uncritically leaving out a major aspect to be considered.

We understand this comment and have expanded the Geological Background (Line 89) and Discussion (Line 389) to include the hypothesis that listvenite formation was related to extensional tectonics after ophiolite emplacement. We added relevant references.

We would like to point out that a structural / geodynamic reconstruction of the Fanjah area or constraining the absolute timing of listvenite formation is beyond the scope of this paper, and due to limitations in text length we cannot discuss this in great depth. For a lengthier discussion, the reader is referred to a number of the papers cited in our manuscript. Nevertheless, we would like to point out a few things here regarding this comment:

The timing and geodynamic setting of listvenite formation proposed in older papers is, while valid hypotheses, not supported by data at the time of publication of those papers and inconsistent with newer data. For example, meteoric- or seawater-driven carbonation to form listvenites, as proposed by Wilde et al. 2002 and Stanger 1985, is inconsistent with geochemical and isotope data from Hole BT1B (e.g. Falk & Kelemen, 2015; de Obeso et al., preprint). Likewise, the Tertiary high heat flow at 36 – 40 Ma as proposed for listvenite formation by Wilde et al. (2002), and which also Nasir et al. 2006 discuss as a viable scenario, is neither in agreement with the Rb/Sr age of Falk & Kelemen 2015 (even when taking into account the large error), nor is it consistent with the 55 – 60 Ma U/Pb date of two dolomite or calcite veins cutting listvenite (Scharf et al, 2020), which places listvenite formation itself to before that age. (Nota Bene: we only find conference abstracts reporting these U/Pb dates, so we unfortunately cannot evaluate and cite these).

The policy of the authors is not to include abstracts into the reference list but to include at the same time non-peer reviewed articles (e.g., Kelemen et al., etc.). This is a case of double standard.

>> With regard to referencing we follow the policy of Nature publishing. All cited preprints were in the peer-review process and fully accessible on preprint servers at the moment of first submission; by now all of these articles have been accepted for publication (bibliographic details are updated accordingly in the revised version).

It is therefore puzzling why Reviewer 1 writes that Mattern & Scharf (2018) and Scharf et al.

Confirmed those previous studies: we cannot find this confirmation in Mattern & Scharf (2018).

Please have a look in Mattern and Scharf (2018) again. There is written "Because listwaenite is a hydrothermal alteration product of serpentinized peridotite (e.g., Nasir et al., 2007) and because it occurs directly at the ophiolite-Mahil contact it must have formed directly in the extensional fault that provided pathways for hydrothermal fluids to circulate to form listwaenite." I do not know what the reviewers mean with "Scharf et al." - which year and which journal/book chapter?

>> Possibly there has been a misunderstanding; here we referred to the comment „An older model for the origin of the listwaenite is proposed from Stanger (1985), Wilde et al. (2002), Nasir et al. (2006), and was confirmed by Mattern and Scharf (2018) and Scharf et al. (2019, 2020).“ by reviewer 1. We disagree with the second part of this comment because the model by Stanger (1985) and Wilde et al (2002), which is based around Tertiary heat flow and meteoric fluids, is inconsistent with the data on the listvenites around Site BT1, as we have explained above. Similar to those previous studies, Mattern and Scharf (2018) as well as Scharf et al. (2019, 2020) propose a setting of extensional tectonics of listvenite formation, but they do not provide evidence that would confirm the Tertiary heat flow + meteoric fluid model of Stanger (1985) and Wilde et al (2002). We do not doubt that extensional faults are very abundant in the area and also occur at contacts between

peridotite/listvenite and underlying units, as observed by Mattern and Scharf (2018) and Scharf et al. (2019, 2020); but we cannot find proof that listvenite formed during this phase of faulting in these papers. With respect to the above citation from Mattern and Scharf (2018) by reviewer 1, that „*listwaenite [...] must have formed directly in the extensional fault*“: we disagree with the reasoning and the implied certainty – the arguments that listvenite is a hydrothermal alteration product and that they occur along the faulted mantle / sediment contact are not sufficient proof to say that they must have formed during extensional faulting. The listvenites may as well have formed earlier along the thrust or along faults subparallel to the thrust, with later extensional faulting exploiting these earlier structures and the rheological contrast between listvenite, serpentinite, and sedimentary rocks.

It is possible that listvenite formation occurred during extensional faulting, but in our opinion the current data of the detailed investigations around site BT1 all point towards an obduction setting for listvenite formation, which was then overprinted by several phases of extensional tectonics. Providing a definite answer to this question is beyond the scope of this paper; more studies including better dating of listvenite are required to answer this.

With respect to the ages, we would also like to point out that post-obduction extension at 60 Ma (Why at 60 Ma? Post-obduction extension started immediately after the emplacement or even slightly overlaps in time with the emplacement, i.e., at ca. 75 Ma. This fact is long known. Nolan et al. (1990) already stated that some of components of the Campanian(?) to Maastrichtian Al-Khod Formation from outside the Saih Hatat Dome are derived from pre-Permian formations of the Saih Hatat Dome (Maastrichtian = 72 – 66 Ma). Thus, the Saih Hatat Dome must have undergone massive exhumation phase already during the Late Cretaceous), while falling within 2 sigma of the 97 Ma (± 29 Ma 2 sigma) internal Rb-Sr isochron from Falk & Kelemen (2015) and thus not impossible, is much less likely than the timing of subduction and obduction (97 – 74 Ma; e.g., Searle & Cox, 2002), which is well within 1 sigma (Why 1 sigma? 1 sigma would be ± 58 Ma, thus includes the Upper Jurassic to Oligocene. The 97 ± 29 Ma (2 sigma) age actually does not support that the listwaenite formed during obduction. The error margin includes the Campanian and large parts of the Maastrichtian, and therefore, post-obductional deformation. Therefore, the Rb-Sr isochron age must be removed from the manuscript or you clearly state that Rb-Sr age has a large error margin which also includes large parts of the post-obductional history of the Oman Mountains.

>> Apparently our reply was not very clear or somewhat confusing. We would like to clarify: A 2 sigma uncertainty of ± 29 Myr of the isochron mean age of 97 Ma means that with a probability of 95% the true age is between 68 - 126 Ma, and with a probability of 68% (± 1 sigma standard deviation from mean) that it is older than 82 Ma (see e.g. Schoene et al., 2013, 10.2113/gselements.9.1.19, for the probability that a true age falls within different standard deviations of the measured mean). Considering that geochronology shows that the main phase of continental subduction for the As Sifah eclogites in NE Oman occurred around 81 – 77 Ma (Garber et al., 2021 JGR), listvenite formation at BT1 is significantly more likely to have occurred during thrusting than afterwards.

It is clear that the isochron age is imprecise (which we state in line 87) and we are keenly aware of ongoing work trying to precisely date the carbonation process, which is very difficult and has not been successful yet. It is further clear from the error margin that the probability that the listvenites are younger than 77 Ma is, although smaller, not negligible, which is the reason why we do not rule out a post-obductional timing but included this possibility (line 87 - 89: „*An internal Rb-Sr isochron with an imprecise age of 97 \pm 29 Ma (2 σ)⁸ (2 σ uncertainty confirmed by personal communication with L. Falk) suggests that listvenite formation was concurrent with intra-oceanic subduction³³, continental subduction of the Arabian margin and ophiolite obduction (85 – 77 Ma; e.g.,³⁷), or an early phase of post-obduction extension^{35, 38, 39}.*“ In this line, we updated reference 37 to Garber et al. (2021) who present new geochronology data and re-evaluate age estimations for continental subduction from previous studies. In the discussion, we only state that based on these geochron data

(and further indirect geochemical indications as indicated and explained in more detail in our previous reply) we consider subduction-obduction / thrusting as the most likely setting (line 394). Please note that we never „confirm“/“prove“ this proposed setting, as the current data do not allow such certainty.

We disagree with the statement of reviewer 1 that „The 97 ± 29 Ma (2 sigma) age actually does not support that the listwaenite formed during obduction“ – the age is fully consistent with the timing of obduction. This does not exclude a possible younger timing coinciding with early post-obduction extension, but as written in the manuscript we consider that less likely for the reasons given here and in the manuscript.

Furthermore, Geologic mapping of the outcrops north and northeast of Site BT1 shows that listvenites are gently folded in a broad antiform. In this structure, they are overlain by banded peridotites and dunites – the Banded Unit – with distinctive major and trace element compositions characteristic of the base of the Samail ophiolite mantle section, and underlain by the metamorphic sole, which in turn overlies highly deformed sediments of the Hawasina Formation (see Kelemen et al., preprint available for reading and commenting on ESSOAR). These structural characteristics indicate that the listvenites formed within or just above a shallowly dipping fault zone along the interface between the ophiolite, the metamorphic sole, and meta-sediments beneath the ophiolite. This geometry is inconsistent with listvenite formation along steep normal / strike-slip faults associated with late extension (What do you mean with “late” extension? Latest Cretaceous to Paleocene or Eocene or Neogene?), as proposed by Scharf et al.

I am not sure to which work of Scharf et al., the authors are referring to, but listwaenite does not have to form exclusively on steeply dipping “late” normal faults or strike-slip faults. When I understand the authors correctly, then they interpret the described shallowly dipping fault zone as a thrust, which formed during obduction. I challenge this interpretation and claim that the shallowly dipping fault zone was the thrust, which was reactivated as a low-angle extensional fault. It is common that extensional shear zones are at an early stage low-angle faults, while they become progressively steeper dipping in a later stage (here an example from the Alps; Scharf et al., 2013). Scharf, A., Handy, M.R., Favaro, S., Schmid, S.M. & Bertrand, A. (2013) Modes of orogen-parallel stretching and extensional exhumation of thickening orogenic crust in response to microplate indentation and slab roll-back (Tauern Window, Eastern Alps). *International Journal of Earth Sciences*, 102(6), 1627-1654, doi: 10.1007/s00531-013-0894-4. Springer

>> We agree that it is possible that the thrust was reactivated as an extensional shear zone, but to date this is not known for the area surrounding site BT1, and there is currently no evidence that carbonation of peridotite to listvenite was related to this process. Future studies with precise age determination may or may not prove this, but currently we do not see any proof for this scenario. We do not exclude that a younger timing coinciding with early post-obduction extension is possible, but as written in the manuscript we consider that less likely for the reasons given here and in the manuscript.

If the authors wish to underpin their interpretation, they must present structural data of the shallowly dipping fault zone. These data must demonstrate that shearing ensued during shortening (obduction, thrusting), and listwaenite was also sheared during thrusting. Otherwise, the interpretation with the shallowly dipping fault zone = obduction/thrusting is pointless. In both cases (thrust or extensional fault) the shear sense is similar towards the SW. Extensional shearing would be related to post-obductional doming of the Saih Hatat and/or Fanja Saddle, starting at a late stage or immediately after the emplacement of the ophiolite.

>> As we have stated in the methods section, the nature of our data and samples do not permit a conclusive answer whether these structures formed during contraction or extension. However, the aim of this study was not to answer the question whether listvenite formation was during shortening or extension, but to investigate whether and how deformation and carbonation reaction were

coupled, the mechanisms of this coupling and its implication for evolution of fluid flow. We agree that further research including analysis of the structural evolution in a variety of sites and precise dating of carbonation could help to constrain the geodynamic context better; we added a respective statement to line 400. Nonetheless, we disagree that the discussion of the geodynamic context and our interpretation which setting is more likely, as included in the last part of the discussion, would be “pointless”. We believe that it is in the best interest of science to provide the reader with the possible settings that are consistent with the currently available data, and which of these we consider more likely. We do not claim that these data and indirect indications provide certainty, that is not possible at this moment and beyond the scope of this paper.

Moreover, Menzel et al. (2020, JGR) have shown that high-angle normal and strike-slip faulting is common in listvenite, but occurred later than the carbonation reaction.

As I wrote above, listwaenite does not have to form exclusively in brittle “late” normal faults or strikeslip faults. Listwaenite may formed within low-angle extensional shear zones. The brittle faults/cataclasites studied by Menzel et al. (2020) are not dated. Thus, these structures can be “very young” and related to late Cenozoic deformation. See Mattern et al. (2021) and Scharf et al. (2021) for an explanations of Oligocene/Miocene deformation in the Fanja area.

Mattern, F., Bolhar, R., Scharf, A., Scharf, K., Mattern, P. & Callegari, I. (2021) Novelty discovered postmid-Eocene sinistral slip in the eastern Oman Mountains: widely distributed shear with wrench-fault assemblage related to Arabia-India convergence. vEGU 2021-8321, Vienna, Austria.

Scharf, A., Mattern, F., Bolhar, R., Callegari, I., Mattern, F. & Ring, U. (2021) Oligocene/early Miocene major E/W-shortening and NW-oriented sinistral slip with wrench-fault assemblage in the Oman Mountains related to oblique Arabia-India convergence. Earth and Space Science Open Archive, doi: 10.1002/essoar.10508453.1.

>> it is correct that the brittle normal/strike slip faults and cataclasites described in Menzel et al. (2020) are not dated and therefore could by related to Cenozoic deformation, or they could be older – this is not known to date. But their younger timing relative to listvenite formation is clear, and therefore the observation that such faults occur in listvenite and at contacts of listvenite to other units does not imply that the listvenites themselves must have formed during this phase of faulting (as stated for example in Mattern & Scharf, 2018, to which the reviewer refers to in a previous comment).

An extensional setting that we indeed cannot fully exclude is CO₂-fluid flow during potential reactivation of the basal thrust as extensional, low-angle detachments.

However, to date there is no evidence for this from the listvenites (but also not for thrust setting, isn't it?). We note further that while the basal thrust of the Oman Ophiolite accommodated > 100 km contractional displacement, the displacement related to a (possible but to date not well documented) extensional reactivation of the thrust is much less. So what? What is the message of this sentence?

>> as stated before (and as also noticed by Stanger 1985, Nasir et al, 2007, and Wilde et al. 2002), the listvenite occur along or in close proximity to the thrust at the base of the ophiolite. We agree that this alone is no evidence that the listvenites formed during thrusting, and we do not claim this or intend to provide such a proof in this paper. But as written in the manuscript, we consider a convergent setting more likely due to the different indications from geochemistry and geochronology.

For all these reasons, we are of the opinion that a subduction/obduction setting for listvenite formation is most probable, and most consistent with the currently available data (please keep your interpretation open for an extensional setting for listwaenite formation, as stated above). We modified the discussion (lines 377 – 395) to clarify this (although, due to text length limitations, in a shorter way).

Line 390-394: See my comments above. The “argument” with the gently folded low-angle contact does not confirm that the listwaenite formed during convergence. I am actually surprised that the authors consider this low-angle contact as an argument against post-obductional origin of the listwaenite.

>> We believe that there may have been a misunderstanding here: We did not state that this outcrop geometry confirms a convergent setting for carbonation – only that it speaks against steep normal faults and that, in combination with the indications from geochemistry and geochronology, we therefore consider a convergent setting more likely. This interpretation does not exclude the possibility of an extensional reactivation of the thrust as a low-angle detachment. To clarify this better we modified lines 391 – 401 to:

“Alternatively, it has been proposed that listvenite formation may have coincided with extensional deformation after obduction of the Samail ophiolite^{38, 39}. The parallel alignment of listvenite bands around Site BT1 with the gently folded, low-angle contacts between the basal mantle section of the ophiolite, the metamorphic sole and the Hawasina sediments³³ indicates that the listvenites formed along the low angle, basal thrust of the ophiolite, and not along steep normal faults. Listvenite formation during early extensional reactivation of the thrust is conceivable, but to date there is no clear evidence for this. Therefore, based on the outcrop geometry, isotope geochemistry⁴⁰, and the isochron age⁸ we consider input of CO₂-rich fluid flow derived from lower-plate sediments during subduction-obduction to be the most likely setting of listvenite formation at Site BT1. Further research including detailed analysis of the structural evolution of the area and precise dating of carbonation could help to constrain the geodynamic context.”

For further discussion (since this is not the main topic of this paper), we would like to refer the reviewer to a recent paper by Kelemen et al. (preprint available for reading and commenting on ESSOAR) where this is discussed in more depth. In our opinion our study also provides an important step: we clearly defined which microstructures are synchronous with listvenite formation as opposed to later overprints, creating a framework for future attempts of structural reconstruction or dating. Our results further underline that careful consideration of the complex (micro)structural evolution is key to obtain meaningful and reliable results during such reconstructions.

The authors interpret their data in a complex and speculative way, just to fit the ‘subduction zone model’, while the alternative model (extension) could easily explain their findings.

We think that the discussed processes and proposed mechanisms, which are based on our microstructural observations, are applicable to listvenite formation regardless of whether that occurred in a contractional or extensional tectonic setting. Where discussing the geodynamic context and implications for subduction zones (Lines 377 - 404), we provide now a clearer, explicit statement of why we consider a thrust setting more likely and better supported.

See my comments above.

>> Please see our reply to the summary comment.

Here is what they wrote:

“Because of the volume increase of serpentinization, under external stress this is likely to cause cyclic variations in permeability, pore pressure and differential stress, which may induce fracturing and the formation of serpentine and early carbonate veins. The formation of listvenite may intensify this process due to its lower permeability and higher strength compared to serpentinite, causing dilatancy by granular flow and reaction-assisted ductile deformation along the reacting lithological boundary. We speculate that this feedback of external stress, changing rheology and high pore pressure helps to facilitate continued reaction to listvenite despite volume increase, as long as CO₂ supply is sufficiently high.”

Why such a complex speculation about changing rheology, high pore pressure and sufficient CO₂ supply? Massive and ongoing solid-state volume increase reaction of ultramafic rocks to serpentinite and listwaenite can be easily explained during an extensional event where fractures open, allowing for fluid transfer. This kind of extension would not occur within a subduction zone.

Here we would like to respectfully disagree with the reviewer. At the depths considered for Listvenite

formation in Oman (What is the depth considered for listwaenite formation? When I read again the pressure estimates during listwaenite formation in the resubmitted article, then I must say that the pressure is not constrained at all. Falk and Kelemen (2015) assumed that the listwaenite formed within a subduction zone. They further used respective geothermal gradients to propose the pressure. This approach to “constrain” the pressure is weak. The authors ignored that listwaenite can have formed during post-obductional extension and within the upper crust, as proposed by, i.e., Wilde et al. (2002). Thus, applying the same logic as before, the pressure of listwaenite formation would be considerably less as proposed by Falk and Kelemen (2015).), all the three principal total stresses are compressive in both contractional and extensional tectonics (would your statement be correct if one assumes an upper crustal setting, instead within the mantle, during of listwaenite formation?). Therefore, high fluid pressures are required for dilatancy. It is unfortunate that in tectonics the names "contractional" and "extensional" (these are used to denote displacement directions) are often misunderstood to mean compressive and tensile stress. In our opinion, the processes proposed in our manuscript can occur in both contractional and extensional tectonic settings. Therefore, our analysis of the complex, "ongoing solid-state volume increase reaction of ultramafic rocks to serpentinite and listwaenite" is applicable for both settings, and cannot be easily explained by an "extensional event".

>> With regard to pressure (and depth) of listvenite formation: We clarify line 85 to further make sure that all constraints and assumptions pertaining to the proposed pressure range are transparent. It is clear that the constraint is poor (as we had already stated in the manuscript), but that is unfortunately the nature of most ultramafic rocks. We exclude a very low pressure scenario with meteoric fluid circulation as proposed by Wilde et al. (2002) and mentioned here by the reviewer, because this is not consistent with the isotope geochemistry (see Falk & Kelemen 2015 GCA; de Obeso et al., 2022, JGR). It may be useful here to remark that there is ample literature about carbonation processes caused by near-surface fluids such as meteoric fluid or seawater in the Samail ophiolite, related to alkaline springs and carbonate veins (e.g. Noel et al., 2018, but also many other papers) and related to interaction of serpentinites with overlying Maastrichtian to Eocene limestones (de Obeso et al., 2018); these carbonated peridotites have very different geochemical and mineralogical characteristics than the listvenites at site BT1. This speaks against a near-surface setting for listvenite formation, as discussed by Falk & Kelemen (2015) and de Obeso et al. (2022).

de Obeso, J. C., Kelemen, P. B., Leong, J. M., Menzel, M. D., Manning, C. E., Godard, M., Cai, Y., Bolge, L., and Oman Drilling Project Phase 1 Science, P.: Deep Sourced Fluids for Peridotite Carbonation in the Shallow Mantle Wedge of a Fossil Subduction Zone: Sr and C Isotope Profiles of OmanDP Hole BT1B, *Journal of Geophysical Research: Solid Earth*, 127, e2021JB022704, 2022.

Noël, J., Godard, M., Oliot, E., Martinez, I., Williams, M., Boudier, F., Rodriguez, O., Chaduteau, C., Escario, S., and Gouze, P.: Evidence of polygenetic carbon trapping in the Oman Ophiolite: Petro-structural, geochemical, and carbon and oxygen isotope study of the Wadi Dima harzburgite-hosted carbonates (Wadi Tayin massif, Sultanate of Oman), *Lithos*, 323, 218-237, 2018.

de Obeso, J. C. and Kelemen, P. B.: Fluid rock interactions on residual mantle peridotites overlain by shallow oceanic limestones: Insights from Wadi Fins, Sultanate of Oman, *Chemical Geology*, 498, 139-149, 2018.

>> with respect to the question whether the statement “At the depths considered for listvenite formation in Oman all the three principal total stresses are compressive in both contractional and extensional tectonics” is correct: yes, we are of the opinion that this also applies in the scenario that listvenite formed during brittle-ductile reactivation of the thrust as low-angle extensional faults, which is a setting that we do not exclude but consider less likely and less supported by data. For near-surface conditions of < 5 km depth (what we think the reviewer may have had in mind when referring to “upper crustal setting” as opposed to “within the mantle”), that is of course not correct, but as shown above and in our previous reply, such a setting is not consistent with the data from BT1.

To sum up the validity of the submitted manuscript, the authors provide detailed geochemical/analytical and microstructural results, but the link to the geodynamics setting of listwaenite formation is not at all conclusive, especially since alternative models for listwaenite formation are left unmentioned and unconsidered.

As mentioned above, the aim and main topic of this paper was to investigate the relationship between deformation, reactive fluid flux, and listvenite formation, irrespective of the tectonic setting where this occurred. All three reviewers agreed with the validity of our analytical and microstructural results and their interpretation. (Ok. Then I would recommend to have the manuscript until this point and exclude any speculation about the tectonic setting during listwaenite formation).

>> We respectfully disagree with this recommendation. We believe that we have transparently presented our arguments, while not excluding other possibilities (here we have modified the wording in line 394 to clarify this a bit more). We also would like to note that the proposed setting, as well as the way in which we propose it, is not a wild speculation – we consider and weight different observations and data (including geochemistry), which lead us to a preferred setting that is in no way a geo-fantasy: it is undisputed that the base of the ophiolite and the metamorphic sole is a thrust that experienced > 100 km of displacement during obduction, and the listvenites happen to occur along and close to this contact. Future studies with precise age determination may or may not prove whether listvenite formed during thrusting. We admit that the reasoning in the manuscript may be a bit short, which is due to the different main focus of this paper and the text length limitation, but here we refer the reader to the cited studies where this is discussed more extensively.

Although outside of the main goal of our study, we have revised the manuscript to include a lengthier discussion around possible tectonic settings. We add to the discussion: “A well-suited fluid source are carbonate-bearing sediments rich in hydrous silicates that undergo devolatilization related to heating, for example due to contact metamorphism or burial during thrusting ^{6, 40, 50}. In the Samail ophiolite, burial of the Hawasina nappes and underlying units below the ophiolite caused fluid flux and the formation of overpressure cells ³⁵, which, combined with active deformation along the basal thrust, might have created the circumstances necessary for listvenite formation. CO₂-bearing fluid= may have been sourced from deeper, partially subducted sediments, where temperatures are higher and devolatilization reactions more efficient ^{33, 40}. Alternatively, it has been proposed that listvenite formation may have coincided with extensional deformation after obduction of the Samail ophiolite ^{38, 39}. However, the parallel alignment of listvenite bands around Site BT1 with the gently folded, low-angle contacts between the basal mantle section of the ophiolite, the metamorphic sole and the Hawasina sediments ³³ indicates that the listvenites formed along the low angle, basal thrust of the ophiolite, and not along steep faults during later extension. This geometry, isotope geochemistry ⁴⁰, and the isochron age ⁸ indicate that the most likely setting of listvenite formation at Site BT1 is subduction-obduction.”

See my comments above for the validity of some of your “arguments” about the tectonic setting. I cannot comment on the geochemical statements, because I am not a geochemists.

>> Please see our reply to the summary comment and a similar comment above.

The work describes detailed microstructural analysis of listwaenite and listwaenitisation. This is interesting for specialists working on this matter (in geochemistry and fluid-rock interaction processes) but its significance to the larger community is limited.

We do not agree with this statement; we believe that this paper presents exciting new results that significantly improve our understanding of the mechanisms of coupled deformation and fluid-rock interaction in mantle rocks, which is of high importance for anyone studying reactive fluid flux and carbonation processes (but this is what I also have written above). The implications of our findings reach far beyond the regional context of the Fanjah region of Oman with direct impact on CCSU research and application in mafic and ultramafic rocks. I do not know what “CCSU” is. I did not state that this research is only interesting for the regional geology of Oman. Thus, I do not understand why the authors wrote this comment.

>> Reviewer 1 stated that the „significance to the larger community is limited“. We do not agree with this statement, as we consider our results of significance for the wide community of researchers studying coupled thermal-hydraulic-mechanical-chemical processes during fluid-rock interaction, as well as for the broad community of carbon capture, utilization, and storage (in our previous reply two letters were mistakenly mixed up in the abbreviation, the common abbreviations are CCUS or CCS).

A discussion of the observed structures with respect to tectonic events would also increase the significance of this manuscript (i.e., the described S-C fabric, boudin and folds; presence of kinematic indicators for events during which these structures formed?). Even though, I am not a specialist in the microanalytical field, I am confident that the described features exist, and that the used methods are of high-quality, but I have to seriously criticize the authors for not indicating in the Methods section what kind of well material was at their disposal. There is no indication whether they worked on an oriented well core or not. In case they did, the fold vergences, shear sense should be provided with cardinal directions – critical in support of the subduction or extension interpretation. The lack of orientation/direction information is another main weakness of the manuscript.

A lack of orientation data of our samples from the inclined drilled Hole BT1B, the difficulties to identify and measure small-scale structures macroscopically in the field, and the ubiquitous occurrence of cataclasites and faults overprinting the listvenites after carbonation (see Menzel et al., 2020 JGR), where rotation of different fault blocks is likely, do not allow a confident correlation of orientations. Therefore, our data do not allow a more detailed structural geologic interpretation: this is not the aim of our paper.

(Correct but important for your interpretation: origin of the listwaenite - obduction vs. extension).

>> (a note of interest, although this goes far beyond the topic of this manuscript: there is widespread evidence of extensional deformation also in subduction zones (see for example Herman & Govers, 2020, EPSL, doi: 10.1016/j.epsl.2020.116379, and references therein; a similar process has also been proposed for normal faulting in the inner wedge of the Makran subduction zone by Normand et al, 2019, Terra Nova, <https://doi.org/10.1111/ter.12419>). Thus, even if it can be shown that those deformation structures in listvenite that are coeval with carbonation are extensional, this may still not provide a definite answer about the timing and tectonic setting of listvenite formation, unless dating/thermochronology is combined with detailed structural analysis.)

We do note that an extensive and detailed report of the measurements and analyses has been fully published and accessible in detail and with full transparency in the cited Proceedings of the Oman Drilling Project (Kelemen et al. 2020) **(Was this work peer-reviewed?)**. Per the reviewer's suggestion, we have expanded the explanation of the nature of the samples and orientation data available to us in the Methods section.

>> the Proceedings volume of the Oman Drilling Project (Kelemen et al. 2020) follows IODP publication standards, with internal peer-review through the large team of scientists involved in the project, and adheres to the full open access policy of all data (including raw data) to the public and available for further research, which is common to IODP expeditions.

As mentioned above, the models for the listwaenite formation are not discussed. Thus, specific references are missing. I have indicated those into the detailed text to the reviewers. Furthermore, some references to the geological map and cross section are missing. I have added them too.

We have added appropriate referencing where necessary.

See my comments about referencing above.

>> With regard to referencing we follow the policy of Nature publishing. All cited preprints were in the peer-review process and fully accessible on preprint servers around the time of first submission; by now all of these articles have been accepted for publication (bibliographic details are updated accordingly in the revised version).

The authors also missed that ductile-brittle deformation in listwaenite of the greater Fanja area has

been described already. I have mentioned the respective reference below.

Here we respectfully disagree: we present here for the first time detailed microstructural analysis and, based on this data, an exciting new model for Listvenite formation, with high relevance for our understanding of fluid-rock interactions. The conference abstract the reviewer refers to describes field structural relations of brittle and semi-ductile deformation in listvenite. But unfortunately, the actual data have not been published, so we cannot compare our results in detail, and cannot evaluate if the described structures are synchronous to listvenite formation (such as the folding of veins described in this paper) or if they are related to tectonic overprints after carbonation (similar to the cataclasites and faults described by Menzel et al., 2020 JGR).

Finally, the manuscript reveals a limited familiarity of the authors with the regional geology of Oman, and the respective literature. Text, figures and literature references must be improved accordingly. We respectfully disagree. J.L. Urai and P. Kelemen have investigated different parts of the geology and tectonics of Oman over several decades, with more than 50 publications in high-ranked peer-reviewed journals.

If the authors do not mention (in the first version of the manuscript) that a long-known alternative model for listwaenite formation is post-obductional, then the authors missed a substantial point (including missing references). Furthermore, the mistakes in the geological map and cross section indicate a limited familiarity with the regional geology of Oman (par-autochthonous vs. autochthonous; folded contact of the base Hajar Supergroup at the SW limb of the Saih Hatat; wrong locality of Muscat (which is still wrong in the revised version of the manuscript – Muscat is 10 km to the E of the present location and within the ophiolite). These (in my view) substantial mistakes should not be made by anyone who is fully familiar with the regional geology of Oman.

>> Muscat is a quickly expanding metropolitan urban area, therefore it has become common use in recent years to place the city location further west than the historic center (to which reviewer 1 refers to). As stated in our previous reply, we corrected the mistakes in the schematic cross-section and the map legend, and we are thankful that the reviewer raised our attention to these details.

For the reasons stated above, I cannot recommend publication of this manuscript.
Andreas Scharf, Muscat, June, 2nd2021.

Line 84: The authors write that the pressure of listwaenite formation is poorly constrained with a possible range of ca. 0.3 to 1.2 GPa (Falk and Kelemen, 2015). Actually, the pressure is not constrained at all. Falk and Kelemen (2015) assumed that the listwaenite formed in a subduction zone and calculated the respective pressure based on the overburden of the oceanic lithosphere. They also used literature data from the metamorphic sole (which has seen up to 900°C – unlike the listwaenite with <250°C). For a fair evaluation of the pressure during listwaenite formation the authors must acknowledge that the pressure would have been significantly less than 0.3 GPa if one assumes that the listwaenite forms postobductional and during upper crustal extension as proposed by, e.g., Wilde et al. (2002).

>> please see our reply above regarding pressure of listvenite formation

Line 87: Here the referencing is incorrect. Falk and Kelemen (2015) did not provide the 2 sigma error. It cannot be found in this article. The authors mention below that the 2 sigma error is based on personal communication with the first author of Falk and Kelemen (2015). Thus, the authors have to clearly mention this in the text. This is an important detail.

>> we add “(2 σ uncertainty confirmed by personal communication with L. Falk)” in line 88

Line 96: “meta-sediments”. Are these rocks really metamorphosed? In the field, the Hawasina sediments are highly folded but appear to be unmetamorphosed. Furthermore, Aldega et al. (2021) reported peaktemperatures for the Hawasina rocks of ca. 200°C or less. Thus, I doubt that the Hawasina rocks are meta-sediments.

Aldega, L., Carminati, E., Scharf, A. & Mattern, F. (2021) Thermal maturity of the Hawasina units and origin of the Batinah Mélange (Oman Mountains): insights from clay minerals. *Marine and Petroleum Geology* 133, 105316, doi: 10.1016/j.marpetgeo.2021.105216

>> The lower temperature limit of metamorphism is not universally defined, but can be as low as 100 - 200 °C depending on the rock type (Bucher & Grapes, 2011, *Petrogenesis of Metamorphic Rocks*, 8th edition). As the smectite to illite transition is a metamorphic recrystallization and dehydration reaction occurring in this temperature interval, we may consider the Hawasina sediments as low grade metamorphic. This question is somewhat a matter of definition, and not relevant for the topic of this paper. We therefore change the word “meta-sediments” to “(meta-)sediments” in line 96.

Following reviewer #1’s comments and recommendations, we have made significant modifications to the manuscript and to Figure 1 (see also replies to detailed comments below), and hope that we have addressed all issues appropriately, providing an improved revised version of the paper.

Again, the authors produce a technically well-written manuscript. However, the interpretation of a subduction-setting during listwaenite formation is not supported by data and biased (see my detailed comments). This includes that the authors use their own citations/references which are not peerreviewed to support their interpretation. At the same time, the authors exclude published abstracts with a different model. The tectonic-setting issue is not the main topic of this manuscript, I agree. However, it is an important section of the interpretation. I would recommend to fully exclude any discussion about the tectonic setting during listwaenite formation because the authors cannot present any structural evidence supporting their hypothesis (see my comments above). The tectonic discussion as it stands now in the manuscript is biased and opinion-driven.

>> Please find our reply to this summary comment at the beginning of this rebuttal letter, where we have also copied this comment of reviewer 1.

Replies to detailed comments by Reviewer 1:

“(see Jackson, 2005 for spelling of listwaenite)”

Several forms of spelling of the word listwaenite can be found in literature, just as different spellings of Samail / Semail / Sumail exist. We prefer “listwaenite” because it is closest to the spelling of the type locality in the Ural mountains, derived from kyrillic as argued by Halls & Zhao 1995.

Abstract

Line 2. “Oman Ophiolite” is misleading. There are two different ophiolites in Oman. I would be specific (“Semail” or “Samail” or “Sumail” Ophiolite – the spelling of “Samail” is different in the literature). Do this for the entire manuscript and art work.

“Oman ophiolite” has been used in papers by pioneers of the ophiolite research such as Nicolas and Boudier; but we recognize that there is a second ophiolite in Oman so Samail Ophiolite is more accurate.

We have changed “Oman ophiolite” for “Smail ophiolite” throughout the manuscript and figures.

Line 2. Several. Could you be more specific? Do you have a value?

Line 3. What is “GT”? Do you mean “Gt” (gigatonne)?

2- 3 Gt.

Line 6. Tectonic deformation. This is the key, but what kind of deformation (i.e., compressional, extensional?). Be specific!

See reply to major comment

Line 6. Folded veins. Here you should provide information of the folds. Are they open, isoclinal, harmonic, ptygmatic? Orientation of the fold axes, axial planes? Do you observe several folds with the same orientation?

The word count of the abstract is too limited to include these details here (those are, when possible, described in the results section. See also reply to comment about orientations).

At which temperature does magnesite grow or become folded/ductile? This would provide information about the conditions during the respective deformation event.

There only are deformation experiments for $T > 500\text{ }^{\circ}\text{C}$, no data exists for these low temperature conditions.

Line 11. I am not sure how the lithostatic pore pressure can promote a dilatant porosity, especially within a subduction zone (I can imagine that pre-existing pores could be flattened in a subduction zone due to simple shear at an early stage). Generally, I would rather expect the contrary. In a subduction zone, any potential pores will be closed, considering the lithostatic pressure. Why do you not consider CO_2 transport during extension instead within a subduction zone?

High pore pressures can lead to tensile fracturing also in non-extensional conditions. In Oman, this has been shown for carbonate veins in (par)autochthonous units in response to burial below the ophiolite and the Hawasina nappes, which created fluid overpressure cells (Grobe et al., 2019, *Solid Earth*, 10, 149–175). There are well studied examples of compressional shear zones facilitating fluid flux and related alteration, e.g. the Glarus thrust (e.g. Burkhard et al., 1992, *Contributions to Mineralogy and Petrology* 112, 293-31) or thrusts in the Pyrenees (e.g. Muñoz-López et al., 2020, *Geofluids*), as well as numerous subduction melanges (e.g. Angiboust et al., 2014, *J Petr*). See also Sibson, 2017 (*Earth, Planets and Space* 69:113). To clarify: the mantle rocks form the upper plate, and they started as essentially zero-porosity peridotites that became more porous and permeable only due to hydration. This is a very different situation than flattening and closure of pores that would occur in a subducted slab. That said, please note that we do not limit our model to compressional settings such as subduction zones, and that also in extensional, (semi)ductile shear zones, creation of fully pervasive porosity is by no means guaranteed.

7

If your assumption is correct, i.e., dilatant porosity during subduction, then listwaenite must be common in every subduction zone of the world and everywhere distributed at the base of the Semail Ophiolite.

We do not agree with this comment. There are other requirements for a listvenite to form apart from porosity: CO_2 concentration in aqueous fluid must be very high and CO_2 flux must be sustained, also, high / lithostatic pore pressures need to be repeatedly renewed by fluid influx to maintain this process; all these factors need to play together to allow listvenite formation.

However, listwaenite occurs in the Fanja area only with few exceptions near the Hawasina Window and in the UAE. In the Fanja area, listwaenite occurs in extensional fault zones.

There are serious problems with your statement.

That extensional and strike slip faults cut listvenites and surrounding rocks in the Fanjah area (e.g. Scharf et al., 2019 *Tectonics*) is not proof that listvenite formed during that stage of tectonic deformation.

(Correct. However this is not stated in Scharf et al., 2019. This is what it says in Scharf et al., 2019 “In addition, there is evidence of a ductile shear zone and brittle faults with slickenlines at Wadi Mansah (stop 12 in Figure 8; Table S1) within listwaenite rocks at the base of the Semail Ophiolite. The faults strike parallel to Wadi Mansah with $110\text{--}120^{\circ}$ and dip of $\sim 75^{\circ}$ toward the SW and reveal either extension or oblique sinistral motion based on a large sigma object and mineral steps. Sinistral transtensional faults in Wadi Mansah have been described by Bailey et al. (2019).”. Thus, I do not understand the criticism).

>> this seems to be a misunderstanding: it was not our intention to criticise the results presented in Scharf et al. (2019, *Tectonics*), but just to state that the presence of extensional faults in listvenite (as for example observed and documented by Scharf et al, 2019) does not equal “[listwaenite] must have formed directly in the extensional fault” as stated in Mattern & Scharf (2018).

Crosscutting microstructures of sharp faults actually indicate that they were active after listvenite was formed, and were often related to carbonate dissolution, not carbonation (see Menzel et al., 2020 *JGR*) (Menzel et al. worked at one location, I would be careful to generalize the statement of Menzel et al. for the entire Fanja area. Mattern et al., 2021 and Scharf et al., 2021 provide a reason for the formation of the “late” microstructures”. See above). Still, as outlined above, we do not

exclude the hypothesis that listvenite may have formed during early reactivation of the thrust as extensional shear zones, but we consider it unlikely.

>> It is correct that the paper by Menzel et al. (2020) is focussed on site BT1, and that not all observations from BT1 may equally apply to all other listvenite occurrences in Oman. But it is clear that (i) Site BT1 is the best-studied listvenite occurrence in this area (and by now, anywhere in the world), (ii) the implication of Menzel et al (2020) and this manuscript that detailed micro-structural study is required to distinguish syn-carbonation deformation events from later overprints applies to any listvenite occurrence, and (iii) that such detailed investigations have not yet been carried out in other sites of the Fanjah area and are not included in the different published papers by Scharf, Mattern and co-workers (this latter observation is not meant as a criticism of these studies, as they are primarily focussed on the regional geology of the area based on field structural measurements and observations and not so much on listvenites, but to clarify that their relevance to the processes and timing of listvenite formation is therefore somewhat limited in our opinion). We hope that future research will shed more light onto this question.

Introduction

Line 16. Do you have a reference for your statement?

Statement removed

Line 21. The topic with the “leading edge of the mantle wedge” is a discussed matter in literature. I do not see any geological or structural convincing evidence that your case stated is correct. It is common practice to mention an established alternative (Listwaenite does not form within a subduction zone but postobductionally during extension, i.e., Stanger, 1985; Wilde et al., 2002; Nasir et al., 2006; Mattern and Scharf, 2018; Scharf et al., 2019, 2020).

Mattern, F. & Scharf, A. (2018) Postobductional extension along and within the Frontal Range of the Eastern Oman Mountains. *Journal of Asian Earth Sciences* 154, 369-385, doi: 10.1016/j.jseaes.2017.12.031.

Scharf, A., Mattern, F., Bolhar, R., Bailey, C.M. & Ring, U. (2019) Early Cenozoic listwaenite formation at a major extensional fault zone of the Oman Mountains (Fanja area) – insights from structural analysis and UPb carbonate dating. AAPG Event, Structural styles in the Middle East, 9th-11th December 2019, Muscat, Oman.

Scharf, A., Mattern, F., Bolhar, R., Bailey, C.M. & Ring, U. (2020) U-Pb dating of postobductional carbonate veins in listwaenite of the Oman Mountains. International Conference on Ophiolites and the Oceanic Lithosphere: Results of the Oman Drilling Project and Related Research, 12-14th January, 2020, Sultan Qaboos University, Muscat, Sultanate of Oman.

Stanger, G., 1985, Silicified serpentinite in the Semail nappe of Oman: *Lithos*, v. 18, p. 13-22.

Wilde, A., Simpson, L., and Hanna, S., 2002, Preliminary study of Cenozoic hydrothermal alteration and platinum deposition in the Oman Ophiolite, in Jessell, M. J., ed., *General Contributions: Journal of the Virtual Explorer*, v. 6, p. 7-13.

We have expanded our section on the geological background to mention these alternative models.

Line 87: “An internal Rb-Sr isochron with an imprecise age of 97 ± 29 Ma (2sigma) ⁸ (Here the referencing is incorrect. Falk and Kelemen (2015) did not provide the 2 sigma error. It cannot be found in this article. The authors mention that the 2 sigma error is based on personal communication with the first author of Falk and Kelemen (2015). Thus, you also have to clearly mention this in your text. This is an important detail.) suggests that listvenite formation was concurrent with subduction of the Arabian margin ³³, ophiolite obduction (97 – 74 Ma; e.g., ³⁷ – a picky detail: the Arabian margin was not subducted at 97 Ma. At 96 Ma or so, the oceanic lithosphere of the Neo-Tethys Ocean was subducted. Subduction of the Arabian margin, i.e., Hawasina Basin, ensued sometime after 96 Ma. The subduction at ca. 96 Ma took place at an intraoceanic subduction zone at the beginning – subduction of oceanic lithosphere and not continental lithosphere. Obduction started only when the upper plate thrust above continental rocks), or a very early phase of post-obduction extension ^{35, 38, 39.}” (Why the very early phase? Postobductional extension started immediately after

the emplacement of the ophiolite or even in part overlapping in time. The emplacement was completed at 74 Ma or so).

>> It is correct that this sentence could be more precise; we modify it to: *“An internal Rb-Sr isochron with an imprecise age of 97 ± 29 Ma (2σ)⁸ (2σ uncertainty confirmed by personal communication with L. Falk) suggests that listvenite formation was concurrent with intra-oceanic subduction³³, continental subduction of the Arabian margin and ophiolite obduction (85 – 77 Ma; e.g.,³⁷), or an early phase of post-obduction extension^{35, 38, 39}.”* In this line, we updated reference 37 to Garber et al. (2021) who presented new geochronology data and re-evaluated age estimations for continental subduction from previous studies.

We believe this is an objective description of the different possible settings. We further provide a short discussion about the geodynamic setting, including the hypothesis of carbonation during extensional tectonics, and explaining why a subduction-obduction setting is more probable (line 389). However, as that is not the aim of this paper, we do not go into an indepth discussion about the geodynamic setting of listvenite formation.

I consider the text about the tectonic setting as biased and opinion-derived. See my comments above.

>> Please see our reply to the summary comment at the beginning of this rebuttal.

Line 29. This sentence is hard to read/understand. Could you please rephrase it and produce two nice sentences?

Done

Line 65: I am not sure if it is just “listvenite” (not “listvenites”).

Line 69: The last sentence is interpretation and does not belong to the introduction. Thus, skip this sentence.

It is common practice to provide the reader with a key conclusion at the end of the introduction, to increase readability and impact of the paper.

Results

Line 75: mostly at the base. There are examples where the listwaenite occurs in a distance of ca. 2.5 km

(in map view) from the base of the ophiolite (i.e., near Buwah). You are showing it in your Fig. 1a.

Changed to: “in serpentinite at or close to the base of the Samail ophiolite”

Just a comment about this particular listwaenite outcrop. Can the authors explain how this large listwaenite (within the ophiolite and 2 km (in map view) away from the “basal thrust”) formed during thrusting?

>> It could be a minor thrust extending into the peridotite, subparallel to the main thrust – but this is pure speculation: we did not study this particular site. This manuscript focusses on Site BT1 and the surroundings; a detailed structural assessment of every single listvenite body in the area is beyond the scope of this paper.

Line 88: Could you provide the error margin (Sigma 1 or Sigma 2) for the Rb-Sr isochron age? I assume that it is Sigma 1, because no details about the error margin are provided in the mentioned reference (at least I could not find any). However, one of the co-authors of the submitted manuscript is a co-author of the provided reference. Thus, this person must know the missing information. The error margin is just gigantic (± 29 Ma Sigma 1; ± 58 Ma Sigma 2).

We contacted the first author of the paper in question (L. Falk) and she confirmed that the ± 29 Ma error margin is indeed sigma 2. Of course, the error is still large, which is why we added the word “imprecise” in the text.

Line 90: Please provide the obvious alternative model that the listwaenite formed after the obduction was completed (e.g., Stranger, 1985; Wilde et al., 2002; Nasir et al., 2006; Mattern &

Scharf, 2018; Scharf et al., 2019, 2020). I wonder why this important aspect is not mentioned in the text.

We modify this line to: “An internal Rb-Sr isochron with an imprecise age of 97 ± 29 Ma (2σ)⁸ suggests that listvenite formation was concurrent with subduction of the Arabian margin³³, ophiolite obduction (97 – 74 Ma; e.g.,³⁷), or a very early phase of post-obduction extension^{35, 38, 39}.” See reply to previous comment.

Please make a correct referenc at this place. Indicate that the 2 sigma error is not documented in Falk and Kelemen (2015). The 2 sigma error was communicated personally.

>> we modify this sentence to: “An internal Rb-Sr isochron with an imprecise age of 97 ± 29 Ma (2σ)⁸ (2σ uncertainty confirmed by personal communication with L. Falk) suggests that listvenite formation was concurrent with intra-oceanic subduction³³, continental subduction of the Arabian margin and ophiolite obduction (85 – 77 Ma; e.g.,³⁷), or an early phase of post-obduction extension^{35, 38, 39}.”

Line 90: In the Hawasina rocks underlying the ophiolite in the Fanja area are hardly any carbonatebearing rocks. Actually, the geological map of Fanja indicates that mostly the Al-Jil Formation (Aj1Mbs) crops out in the Fanja area. This formation consists of radiolarian chert, siltstone and sandstone. There are some minor outcrops of other Hawasina formations exposed (Aj1Vb – basaltic pillow lava; Aj1-2 – flaggy limestone, radiolarian chert and sandstone, and UmC – red radiolarian chert and micritic limestone). Thus, the statement is not supported by evidence. The obvious alternative would be that the source for the carbon is from the Arabian shelf (carbonates), cropping out all over in the area.

First of all, due to abundant tectonic overprint (e.g. Menzel et al. 2020), it is quite clear that the presentday configuration is not the same as it was during listvenite formation. But more importantly, Sr and C isotope signatures of sediments of the Hawasina nappes do provide evidence as they are consistent with those of magnesite in the BT1 listvenite. Furthermore, listvenite formation does not only require a source rock that is rich in carbon, but a rock that will produce aqueous fluids that have a high enough CO₂ concentration (Please note that during the Cenozoic, the Fanja area was below sea level which is evident by the occurrence of shallow marine carbonates. Thus, aqueous fluids were present after obduction. Did you check the Hajar Supergroup carbonates, i.e., Mahil and Saiq Formation for their Sr and C isotope signature? Could they also fit with the magnesite in BT1?).

>> there is ample literature about carbonation processes caused by surface-near fluids such as meteoric fluid in the Samail ophiolite, related to alkaline springs and carbonate veins (e.g. Noel et al., 2018, but also many other papers) and related to interaction of serpentinites with overlying Maastrichtian to Eocene limestones (de Obeso et al., 2018); these carbonated peridotites have very different geochemical and mineralogical characteristics than the listvenites at site BT1. Carbonation of serpentinites by seawater typically produces ophicalcite and not listvenite (see e.g. Grozeva et al., 2017). Together, this speaks strongly against a surface-near setting (whether on-land or seawater) for listvenite formation, as discussed by Falk & Kelemen (2015) and de Obeso et al. (2022).

The paper by de Obeso et al. (2022) also discusses Sr isotope data of autochthonous sediments from the literature, most of these data points are inconsistent with the Sr signature in listvenites; please check that paper for further details about this matter.

de Obeso, J. C., Kelemen, P. B., Leong, J. M., Menzel, M. D., Manning, C. E., Godard, M., Cai, Y., Bolge, L., and Oman Drilling Project Phase 1 Science, P.: Deep Sourced Fluids for Peridotite Carbonation in the Shallow Mantle Wedge of a Fossil Subduction Zone: Sr and C Isotope Profiles of OmanDP Hole BT1B, Journal of Geophysical Research: Solid Earth, 127, e2021JB022704, 2022.

Grozeva, N. G., Klein, F., Seewald, J. S., and Sylva, S. P.: Experimental study of carbonate formation in oceanic peridotite, Geochimica et Cosmochimica Acta, 199, 264-286, 2017.

Noël, J., Godard, M., Oliot, E., Martinez, I., Williams, M., Boudier, F., Rodriguez, O., Chaduteau, C., Escario, S., and Gouze, P.: Evidence of polygenetic carbon trapping in the Oman Ophiolite: Petro-structural, geochemical, and carbon and oxygen isotope study of the Wadi Dima harzburgite-hosted carbonates (Wadi Tayin massif, Sultanate of Oman), Lithos, 323, 218-237, 2018.

Low temperature hydrothermal fluids that percolate limestones do not meet this requirement (Do you have a reference for this? What do you mean with “low-temperature”? The authors suggest a temperature of listwaenite formation which is <250°C.).

>> interaction of serpentinites with overlying Maastrichtian to Eocene limestones is observed in Wadi Fins (de Obeso et al., 2018). These carbonated peridotites have very different geochemical and mineralogical characteristics (mostly calcite; no quartz) than the listvenites at site BT1, speaking against such a fluid source. Further, thermodynamic modelling indicates that dissolution of limestone does not produce highly CO₂-enriched (and relatively Ca-depleted) fluids that are required for the magnesite-quartz assemblage (see e.g. Klein & Garrido, 2011; Kelemen et al., 2022).

de Obeso, J. C. and Kelemen, P. B.: Fluid rock interactions on residual mantle peridotites overlain by shallow oceanic limestones: Insights from Wadi Fins, Sultanate of Oman, *Chemical Geology*, 498, 139-149, 2018.

Klein, F. and Garrido, C. J.: Thermodynamic constraints on mineral carbonation of serpentinized peridotite, *Lithos*, 126, 147-160, 2011.

Kelemen, P. B., Carlos de Obeso, J., Leong, J. A., Godard, M., Okazaki, K., Kotowski, A. J., Manning, C. E., Ellison, E. T., Menzel, M. D., Urai, J. L., Hirth, G., Rioux, M., Stockli, D. F., Lafay, R., Beinlich, A. M., Coggon, J. A., Warsi, N. H., Matter, J. M., Teagle, D. A. H., Harris, M., Michibayashi, K., Takazawa, E., Al Sulaimani, Z., and the Oman Drilling Project Science, T.: Listvenite formation during mass transfer into the leading edge of the mantle wedge: Initial results from Oman Drilling Project Hole BT1B, *Journal of Geophysical Research: Solid Earth*, n/a, e2021JB022352, 2022.

On the other hand, formation of such CO₂-enriched fluids is thermodynamically attainable for sediments that contain carbonate-silicate mixtures if these are heated to (at least) low-grade metamorphic conditions (What do you mean with low-grade metamorphic conditions? Do you mean greenschist-facies conditions? If so, please provide P-T values for Hawasina rocks from the Fanja area. In the field, these rocks appear to be unmetamorphosed. Please note that the Saiq Formation is not a pure carbonate formation in the Saih Hatat area. Here, this formation consist of an alternation of siliciclastics, carbonates and volcanic rocks).

>> We refer here to temperatures that induce metamorphic dehydration reactions, starting with dehydration of clays to micas and successively different reactions with increasing temperature depending on rock composition. Total quantities and CO₂ activity of such dehydration fluids are predicted by thermodynamic models to be rather low for clay dehydration but increase with increasing temperature (see e.g. Bucher & Grapes, 2011, *Petrogenesis of Metamorphic Rocks*, for a general overview of dehydration reactions and related XCO₂ in fluid). Thus, with more heating (e.g. due to burial or subduction), more fluid with higher CO₂ concentration can form.

>> Please note that de Obeso et al. (2022) conclude that rocks similar (or akin) to those of the Hawasina formation are the likely fluid source, the exact source rocks and their current position (and accordingly, which exact temperatures they reached) are unknown. It is also possible that (e.g. carpholite-bearing) metasediments of the subducted ocean-continent transition produced the CO₂-fluids, which then migrated upwards. We would like to refer the reviewer to the paper by Kelemen et al. (2022) for these questions, as this is discussed there a bit more; these details are beyond the scope of this manuscript.

We address this in the discussion, and further details about this background can be found in a number of papers that concern listvenite formation in general, for example Klein & Garrido 2011 and Menzel et al., 2018, and, with respect to listvenites in Oman, Falk & Kelemen 2015, Kelemen et al (preprint) (all references are cited in the paper). Putting it another way: if the shelf carbonates were the source, why are there not listvenites all around the Jebel Akhdar and Saih Hatat, where carbonate rocks are very abundant below the ophiolite (as are faults and detachments)? (Because firstly, normal faults which juxtapose mantle rocks and shelf carbonates are largely missing – i.e., at the southern part of the Jabal Akhdar Dome, and secondly large parts of the Oman Mountains were submerged during the post-obductional history, thus, no or only limited fluids were available to form listwaenite). The answer is that there are several requirements that need to be met: i) a source rock

that has a suitable composition to produce CO₂-rich fluid, ii) devolatilization or fluid fluxing of that source rock at conditions that produce CO₂-rich fluids in high quantities, which in turn requires some kind of heat source, iii) faults, shear zones, a plate interface or another structure along which these fluids can be channelled into peridotite. We find that underthrusting of mixed, carbonate bearing marine-siliciclastic sediment rock below the ophiolite (a configuration that is equivalent to a shallow subduction zone) is a very viable scenario where these requirements can easily be met, because the subsidence causes heating of the sediments upon equilibration of the geotherm, allowing for devolatilization to occur. (I wonder if the requirements are also met within the upper continental crust. I am not sure what the authors mean with “kind of heat source” above (i.e., what is the temperature range of this heat source). I just want to point out that serpentinization is an exothermal reaction, which is a possible low “heat source”.)

>> the exact CO₂ source and the conditions under which they form are not known, but as commented above, the isotope geochemistry points to deep sourced and not meteoric fluid circulation (de Obeso 2022). Heat sources for dehydration to take place may for example be magmatic intrusions (as proposed by Wilde et al. 2002, but inconsistent with the newer age constraints), or burial/subduction. It is an interesting idea that serpentinization may contribute to this, but we consider it unlikely: serpentinization itself requires aqueous fluid flow, if that fluid flow derives from meteoric waters close to the surface, advective heat loss due to hydrothermal circulation is also likely, so it is difficult to perceive how this can trigger dehydration in the sediments below. The timing and nature of fluids responsible for different phases of serpentinization of the peridotites is a topic of ongoing research. A further point to consider here is that the mass balance of CO₂ content in listvenite and achievable CO₂-concentrations of fluids dictates that the volume of source sediments must be substantially larger than the volume of listvenite (Kelemen et al., 2022). Therefore we consider it likely that serpentinization was rather a result of the deeper sourced fluid flux (as a consequence of the thrusting of the ophiolite on top of the sediments) than the cause. High CO₂ concentrations in fluid are even easier to attain at greater depth, and isotope geochemical data suggests that deeper sourced fluids from the subduction zone are a viable scenario (see de Obeso et al., preprint available for reading and commenting on ESSOAR). Fluid flux during early extensional reactivation of the thrust would also be consistent with these considerations, but, as stated previously, we consider it unlikely. However, a simple “obvious” limestone CO₂ source on the other hand is not well thought through in our opinion.

Line 95: You could mention that two dolomite veins cutting the listvenite from the Fanja area have an age of ca. 60 Ma (Scharf et al., 2020).

The referred work reporting the U/Pb dates is a conference abstract. Because we cannot find a published paper, preprint or full online resource that contains this dataset, unfortunately we cannot fully evaluate and cite it.

See my comments about this attitude above.

>> we hope to see these data published in the future, because from the 2020 conference abstract alone (which we believe is a preliminary result) it is not very clear what kind of vein has been dated (i.e. the timing of this vein generation relative to listvenite), it is impossible for us to evaluate the quality of instrumental analysis, and we find the conclusions drawn there not fully supported by the results (which is likely also due to the preliminary nature of conference abstracts).

Line 114: Could you provide any information about the fold, please (i.e., style, interlimb angle, orientation of the fold axis and axial plane, fold vergence?). How many folds did you observe and are they all consistent in terms of style and orientation? Did you observe folded listwaenite at the surface near the well? Brittle to ductile deformation in listwaenite has been reported already (Scharf et al., 2020b). The location is at the northern margin of the Saih Hatat.

Scharf, A., Mattern, F. & Mattern, P. (2020) Ductile-brittle shear zone in a listwaenite body within the Frontal Range Fault of the Oman Mountains (Sultanate of Oman). EGU 2020-12672, Vienna, Austria.

Line 116: S-C fabric. Interesting. Could you provide an assumed shear sense? Is the shear sense consistent for several S-C fabrics? In which direction did shearing ensue? I am afraid that you do not

have any oriented samples. Thus, you cannot make any statement to shear sense and its relationship to tectonic events. Am I correct?

Line 137: Did you observe any shear sense within the shear zones? How do you know that it is a shear zone?

Line 160: Do you have any shear sense for the sigma-clasts?

Line 160: "with a sense of shear consistent..." What is the shear sense?

Line 174: Also, here what is the shear sense? What is the orientation of the fold axis and axial plane? What is the style/shape of the folds? Do all the folds have a similar orientation and style?

WRT (WRT - What does it mean?) (>> with regard to) all above comments: as stated in Methods, the samples are oriented only in the arbitrary spatial reference frame of Hole BT1B, which is not easily translatable into cardinal directions due to its inclination and the non-reconstructable orientation of discontinuous core sections (as is also described in detail in ref. 32). Therefore, fold orientation and vergence, orientation of apparent s-c fabrics and shear senses from sigma clasts cannot be unambiguously attributed and related to specific tectonic events and we cannot determine whether they formed due to thrusting or extensional shear. We modified the methods section "Samples" to clarify this better.

Line 138: Do the elongated magnesite ellipsoids have a preferred orientation? The same for the aligned magnesite dendrites?

Yes, see a few lines further down: "In 3D, micro-CT shows that magnesite ellipsoids are oblate in the foliation plane"

Line 147: Could you determine the orientation of the short and long axes of the boudin?

No, we only have the apparent orientation in thin section plane here.

Line 202: Please provide the meaning of all of your abbreviations such as "TEM" and "STEM" at the first appearance in the text.

Done

Discussion

First section: According to the detailed microstructural observations, I would suggest that the listvenite formed within brittle to ductile shear zones. Such or a similar statement is missing in the first section.

We prefer to not add such a statement in this paragraph; it is discussed further below.

Line 255: "few kilometers". I have not read the provided literature (45-47) in detail, but I would check if it is really "few kilometers" and not "few hundreds of meters".

Thanks for pointing this out; we corrected the mistake.

Line 303: red dot.

Line 303: Systematically oriented structures including folds may indicate that the deformation is related to "external stress".

This comment is not clear to us. Does the reviewer mean "tectonic stress"? As mentioned above, due to a lack of appropriate orientation and macrostructural data we are not in a position to make inferences about the consistency and orientation of stresses that gave place to the formation of the observed structures.

Line" 387: This lithological boundary would be also the locality for a development of a shear zone. This shear zone would also explain why "the listvenite consist of a few major bands of 10s of meter thickness".

We add that this boundary (which at the time of carbonation was the reaction front) also likely focused deformation.

Line 405: I am not convinced of your statement and I cannot understand your point of this sentence. What do you mean with "cyclic variations in permeability"? The permeability is just close to zero. A dramatic volume increase will just block any further reaction. Further reaction will only occur if external tensional stress is present, which will result in opening of fractures.

We modified this paragraph to make it clearer. Because complete serpentinization obviously did occur in the serpentinites, permeability was not always close to zero. Furthermore, high-resolution

images indicate that some level of porosity/permeability is preserved within both serpentinites and listvenites (Figure 7). If the build-up of pore pressure is high enough, tensional fracturing may occur (also if the external stress field is not tensional), allowing for a pulse of fluid flux. Sealing due to volume expansion of serpentinization will disrupt fluid flux again, until pore pressures increase again, thus creating cyclic variations in permeability. We believe that with the modifications it should be clear.

Ok. I now understand your “cyclic variations in permeability” statement. Thanks for the clarification.

Line 410: I cannot follow your speculation. The speculation is not supported by your data. Why do you not consider that the listwaenite and related reactions formed during an extensional stress regime instead within a subduction zone? This would easily explain your observations without needing to construct such a highly speculative statement.

See reply to previous comment. The proposed mechanism is not exclusive to or implicit of subduction zones, but just as well applicable to ductile/semi-brittle extensional shear zones. The arguments and reasoning for it are elaborated in the preceding paragraphs.

Line 414: For the same reason as above, I cannot follow and agree with this statement.

We modified this paragraph to make it clearer

Methods:

Line 422: What is missing is a clear statement whether the samples are oriented or not. This is an important point and must be included. I wonder why the authors did not mention this in the text.

We realize that referring to the arbitrary core reference frame of Hole BT1 is not self-explanatory for readers who are not familiar with IODP core logging standards. We add: “ Hole BT1B (International Continental Drilling Project Expedition 5057-4B) was drilled with an inclination of 75°. Therefore, and because the drill cores contain discontinuities across which the orientation of single core sections could not be restored ³², orientations of structures are unfortunately not easily comparable between different parts of the core and the field.”

Figure 1.

a)

Muscat is depicted at a wrong location. Muscat is further to the east. The term “Jebel Akhdar” is not good visible because of the intersection with black lines. Please remove the “Jebel Akhdar” into the light blue rocks and reduce the faults, that there is no intersection anymore.

Figure 1 has been improved to account for the reviewer’s comments/corrections.

Please change the location of Muscat. It is still at the wrong place.

>> Muscat is a quickly expanding metropolitan urban area, therefore it has become common use in recent years to place the city location further west than the historic center (to which reviewer 1 refers to).

You have drawn a major extensional fault near the well in you section. This fault is not shown on the map. It is the Wadi Mansah Fault Zone of Bailey et al. (2019) and Scharf et al. (2019b). This fault zone has also a sinistral component, which you also could include in the map and section. Some of your “Hawasina” outcrops to the NNE of BT1 do not exist in the geological map of Fanja. I do not know where you got them from. Please indicate!

Still, the Hawasina outcrops (which I mentioned above) are there. It should be Muti Formation.

>> This schematic overview map is redrafted after the map by Béchenec et al, (1992) (reference indicated in the figure caption). Other than simplifying and grouping units and plotting only selected faults for the schematic view, we did not modify the mapped region. As this map is a revision done by Béchenec and others of the earlier Fanja NF40-3F (which dates to 1982-1985) and neighbouring maps, and since we have not done a complete regional mapping ourselves, we used this source and prefer to leave it as it is. We are aware that the available maps may not be always correct and are sometimes ambiguous in this area (including Fanja NF40-3F, where some units that are hard to distinguish in the field have essentially the same color in the map), but this is not really relevant for this paper. Updating the geological map of this area is not the scope of this study.

Bailey, C.M., Hurtado, C., Scharf, A. & Mattern, F. (2019) Structural controls on listwaenite genesis in the Semail Ophiolite, Northern Oman. EGU 2019-5794, Vienna, Austria.

Scharf, A., Mattern, F., Moraetis, D., Callegari, I. & Weidle, C. (2019b) Postobductional kinematic evolution and geomorphology of a major regional structure – the Semail Gap Fault Zone (Oman Mountains). *Tectonics* 38(8), 2756-2778, doi: 10.1029/2019TC005588.

Please provide a reference for your geological map.

The reference of the map source (Béchenec et al, 1992) has been indicated in the figure caption, but we slightly modify where it is placed to make it more clear. Other than simplifying and grouping units and plotting only selected faults for the schematic view, we did not modify the mapped region. As this map is a revision done by Béchenec and others of the earlier Fanja NF40-3F (which dates to 1982-1985) and neighbouring maps, we used this source.

b)

Why is the contact of the “Paleozoic basement” with the “Mesozoic” folded (in the Saih Hatat Dome)? This is not directly related to the study but what you have drawn is incorrect. The pre-Permian rocks of the Saih Hatat Dome comprise a major recumbent fold (compare Miller et al., 2002; see also Hansman et al., 2021 for a review of the deformation in the Saih Hatat Dome).

Corrected. The recumbent fold is schematically indicated by a dotted trace, but this detail is not relevant for the present study.

Miller, J.M., Gray, D.R., Gregory, R.T., 2002. Geometry and significance of internal windows and regional isoclinal folds in northeast Saih Hatat, Sultanate of Oman. *Journal of Structural Geology* 24, 359-386, doi: 10.1016/S0191-8141(01)00061-X.

Hansman, R., Ring, U., Scharf, A., Glodny, J., Wan, B., 2021. Structural architecture and Late Cretaceous exhumation history of the Saih Hatat Dome (Oman), a review based on existing data and semi-restorable cross-sections. *Earth-Science Review* 217, 103595, doi: 10.1016/j.earscrev.2021.103595

The extensional fault at BT1 is much too steeply dipping. Your trace of the section is oblique to the fault. Thus, in your section, the extensional fault must dip with $<60^\circ$. You could also indicate the sinistral horizontal shear sense as described in Scharf et al. (2019b). The vertical displacement at this fault is ~ 7 km (see Scharf et al., 2019b). See also for the listric shape of the large extensional fault Scharf et al. (2019b).

We redraw this fault to make it less steep and to schematically indicate that it is likely listric. However, please note that the cross section is, as stated, schematic, and hence the vertical exaggeration is high and not to scale.

c)

The “Autochthonous Mesozoic sed.” Are actually “Par-autochthonous Permo-Mesozoic sed.”

Corrected

The thickness of the Hawasina nappes in the Fanja area is max. 1 km; see geological map of Fanja. According to the geological map of Fanja, the thickness of the ophiolite is <3 km (not 8-12 km). The thickness of the oceanic crust is <4 km (not 5-7 km).

Please note that the column is a very simplified stratigraphy that serves to give the reader a general context of the Semail ophiolite and the position of BT1 within this idealized stratigraphy (see cited reference). To clarify that thickness of units may be less in the area, we add a statement: “Present-day thickness of units in the area is commonly reduced due to tectonic thinning” to the figure caption. Substantial tectonic thinning was likely also related to abundant cataclasites and faults cutting listvenite (i.e. having formed later than the main carbonation event – see Menzel et al. 2021 JGR), so the accurate thicknesses of units at the moment of listvenite formation is impossible to reconstruct without precise dating of listvenite formation as well as each deformation phase, which is currently not available. A detailed structural and geodynamic reconstruction of the Fanjah region that would additionally be required for that is beyond the scope of this study, that would be a paper in its own right.

Post-obductional sediments (not obduction)

We prefer post-obduction sediments, similar to “post-obduction deformation”

d)

Please provide a scale.

Indications of elevation above sea level for the Wadi bottom and one of the hill summits are now given on the figure.

I would add that the serpentinite is difficult to spot, because it is largely covered by listwaenite scree/talus.

Done

e)

Why has the metamorphic sole two colors? The word “sole” is hard to read.

These are different sub-units within the metamorphic sole (See cited reference)

J and k)

I would add the scale bar values directly into the images. You would save some text in the caption (do this for all the other figures, too).

This is a formatting requirement / journal policy of Nature Communications.

I cannot comment on the other figures because I am not a specialist in microstructures. However, the images look very nice.

Supplementary figures

Fig. S1. All figures need cardinal directions. What is the orientation of the shear zone in “b”? Do you observe any shear sense?

The shown field images were not taken with markers of cardinal directions. Unambiguous shear sense indicators were macroscopically not observed in this shear zone. Such structures are typically not very clear macroscopically due to intense oxidation crusts from weathering of listvenite, while polished surfaces lack the 3D information. Note that also in Hole BT1B listvenite, most observed structures that may provide a sense of shear are difficult to see macroscopically and require microscopic investigation.

Fig. S2.

I assume that the samples are not oriented. Cardinal directions?

Samples are oriented in the (semi-arbitrary) core reference frame (CRF) (see Kelemen et al., 2020), which we indicate on the figure.

Fig. S6. What is the orientation of x, y and z?

Directions x, y, z are those of the BT1B core reference frame (CRF), this is now indicated in the caption of the figure.

Godard M, *et al.* Geochemical Profiles Across the Listvenite-Metamorphic Transition in the Basal Megathrust of the Semail Ophiolite: Results from Drilling at Oman DP Hole BT1B. *Earth and Space Science Open Archive*, 37 (<https://doi.org/10.1002/essoar.10507497.1>).

Is this a peer-reviewed article? Can this be used in the reference list?

>> Yes. The cited preprint was in the peer-review process and fully accessible on preprint servers around the time of first submission; by now this articles have been accepted for publication. The bibliographic details are now updated accordingly in the revised version.

Kelemen PB, Matter JM, Teagle DAH, Coggon JA, and the Oman Drilling Project Science Team. Methods and explanatory notes. In: *Proceedings of the Oman Drilling Project* (eds Kelemen PB, Matter JM, Teagle DAH, Coggon JA, et al.). International Ocean Discovery Program (2020).

Is this a peer-reviewed article? Can this be used in the reference list?

>> the Proceedings volume of the Oman Drilling Project (Kelemen et al. 2020) follows IODP publication standards, with internal peer-review through the large team of scientists involved in the project, and adheres to the full open access policy of all data (including raw data) to the public and available for further research, which is common to IODP expeditions.

De Obeso JC, *et al.* Deep sourced fluids for peridotite carbonation in the shallow mantle wedge of a

fossil subduction zone: Sr and C isotope profiles of OmanDP Hole BT1B. *Earth and Space Science Open Archive*, 27 (<https://doi.org/10.1002/essoar.10507483.1>).

Is this a peer-reviewed article? Can this be used in the reference list?

>> Yes. The cited preprint was in the peer-review process and fully accessible on preprint servers around the time of first submission; by now this articles have been accepted for publication. The bibliographic details are now updated accordingly in the revised version.

Kotowski AJ, Cloos M, Stockli DF, Orent EB. Structural and thermal evolution of an infant subduction shear zone: Insights from sub-ophiolite metamorphic rocks recovered from Oman Drilling Project Site BT-1B. *Earth and Space Science Open Archive*, 60 (<https://doi.org/10.1002/essoar.10505943.1>).

Is this a peer-reviewed article? Can this be used in the reference list?

>> Yes. The cited preprint was in the peer-review process and fully accessible on preprint servers around the time of first submission; by now this articles have been accepted for publication. The bibliographic details are now updated accordingly in the revised version.

Kelemen PB, *et al.* Mass transfer into the leading edge of the mantle wedge: Initial Results from Oman Drilling Project Hole BT1B. *Earth and Space Science Open Archive*, 59 (<https://doi.org/10.1002/essoar.10507370.1>).

Is this a peer-reviewed article? Can this be used in the reference list?

>> Yes. The cited preprint was in the peer-review process and fully accessible on preprint servers around the time of first submission; by now this articles have been accepted for publication. The bibliographic details are now updated accordingly in the revised version.

Kelemen PB, Matter JM, Teagle DAH, Coggon JA, and the Oman Drilling Project Science Team. Site BT1: fluid and mass exchange on a subduction zone plate boundary. In: *Proceedings of the Oman Drilling Project* (eds Kelemen PB, Matter JM, Teagle DAH, Coggon JA, et al.). International Ocean Discovery Program (2020).

Is this a peer-reviewed article? Can this be used in the reference list?

>> the Proceedings volume of the Oman Drilling Project (Kelemen et al. 2020) follows IODP publication standards, with internal peer-review through the large team of scientists involved in the project, and adheres to the full open access policy of all data (including raw data) to the public and available for further research, which is common to IODP expeditions.

REVIEWERS' COMMENTS

Reviewer #4 (Remarks to the Author):

Review of the manuscript entitled "Ductile deformation during carbonation of serpentinized peridotite", by Menzel et al.

I have been contacted to help clarifying the tectonic setting of listvenite formation in Oman. First of all, I must say that I am not bias by any conflict of interest, and I have found the discussion between the authors and the first reviewer very interesting. This discussion could be worth to be published along with the manuscript as supplementary information.

The manuscript is overall well-written and well-organized. As mentioned by the different reviewers, the authors' model detailing the mechanisms controlling the coupling between deformation and carbonation reaction is convincing.

Yet, I am struggling with the authors' interpretation of the Oman's tectonic setting for the listvenite formation. While I acknowledge that the authors mention the different tectonic settings so far proposed for the listvenite formation (ll. 89-91), they suggest that a subduction setting is more likely, and indicate that the feedback mechanisms characterized in their study may be common in the mantle wedge of subduction zones. I find this interpretation speculative and not the most likely one, especially since it is hardly reconcilable with the estimated thermal gradients in convergent settings.

As written in the manuscript, there is only a single and imprecise date for the listvenite formation, and no pressure of formation has been so far estimated. However, the temperatures of formation seem to be well constrained, between 80 and 150°C (l.79). These temperatures are crucial constraints which are underused in the manuscript. The authors indicate that the pressures of formation must range between 0.3 GPa (assuming a present-day ophiolite thickness of 8-10 km) and 1.2 GPa (the pressure for the metamorphic sole formation).

The temperature estimates are inconsistent with a formation at high-pressure (HP) conditions. Indeed, a formation at 35 km depths implies thermal gradients of 2.3°C/km and 4.2°C/km (for temperatures of 80°C and 150°C, respectively). Those estimates are too cold to be realistic on Earth. Therefore, a formation during subduction infancy and the metamorphic sole formation (from > 97 Ma and 92 Ma) can be excluded. And as a side note, I invite the authors to replace the reference for the P estimates of the sole. I understand that Kotowski et al. are part of the same group working on the ICDP project, but their P estimates of 1 GPa are poorly done and meaningless. Those estimates were calculated from the Si-content in white mica in mafic rocks. However, the barometer of Massonne and Schreyer (1987) should not be applied for this kind of lithologies (without phlogopite/biotite and K-feldspar). Moreover, the HP-LT estimates are inconsistent with the Ca-rich chemistry of the amphibole. So, please, refer to Soret et al. (2017) who provided much more robust P-T conditions, and the first ones derived from pseudosection modelling. I understand that there is only a limited number of references allowed, but this is why the references should be the most appropriate.

Mature (oceanic and continental) subductions are usually associated with much colder thermal gradient. For the Oman case, thermal gradient of 10°C/km is estimated from the HP rocks (e.g. Yamato et al., 2009). This value implies a depth of listvenite formation comprised between 8 and 15 km depth. The rocks studied in this study are localized at the base of the Semail ophiolite. Its current thickness is < 10 km, but during the obduction (final emplacement onto the Arabian plate) the ophiolite underwent a major stage of extension drastically reducing its thickness. It is generally accepted that its original thickness was comprised between 15 and 20 km (Hacker et al., 1996; Searle et al., 2015). As a consequence, the formation of listvenite during the mature subduction history (before the obduction) appears to be also unlikely, or at least very speculative.

The date of 97 +/- 29 Ma should not be used to discriminate the tectonic environment. It comes from a single sample and it is imprecise: the Rb/Sr analyses are highly scattered along the isochron (unfortunately

no MSWD is provided).

I do not know when the listvenite formed, but the reactions probably did not occur during the subduction. Both the reviewers and the authors agree that the Oman's tectonic setting is not the main topic of the manuscript. Since it is mainly speculative, I suggest to remove this aspect and/or to keep the discussion more open than it is presently.

March 16

M. Soret

AUTHOR RESPONSE TO REVIEWER COMMENTS

We thank M. Soret for his time and commitment to provide an additional and well-informed perspective on the discussion of the tectonic setting of listvenite formation. Following his overall suggestion, we removed the disputed part of the discussion of the tectonic setting of listvenite formation at site BT1 (lines 390 – 401 in the previous version), and keep the remaining (short) discussion open to all possible settings.

Reviewer #4 (Remarks to the Author):

Review of the manuscript entitled “Ductile deformation during carbonation of serpentinized peridotite”, by Menzel et al.

I have been contacted to help clarifying the tectonic setting of listvenite formation in Oman. First of all, I must say that I am not bias by any conflict of interest, and I have found the discussion between the authors and the first reviewer very interesting. This discussion could be worth to be published along with the manuscript as supplementary information.

The manuscript is overall well-written and well-organized. As mentioned by the different reviewers, the authors’ model detailing the mechanisms controlling the coupling between deformation and carbonation reaction is convincing.

Yet, I am struggling with the authors’ interpretation of the Oman’s tectonic setting for the listvenite formation. While I acknowledge that the authors mention the different tectonic settings so far proposed for the listvenite formation (ll. 89-91), they suggest that a subduction setting is more likely, and indicate that the feedback mechanisms characterized in their study may be common in the mantle wedge of subduction zones. I find this interpretation speculative and not the most likely one, especially since it is hardly reconcilable with the estimated thermal gradients in convergent settings.

We modified the discussion, stating that the observed mechanism of reaction-assisted ductile deformation is likely relevant in a variety of active tectonic settings where high fluid-flux occurs along serpentinite-bearing faults (including transform faults, ophiolite

sutures, and the shallow mantle wedge of subduction zones), but without a special focus on subduction zones.

As written in the manuscript, there is only a single and imprecise date for the listvenite formation, and no pressure of formation has been so far estimated. However, the temperatures of formation seem to be well constrained, between 80 and 150°C (l.79). These temperatures are crucial constraints which are underused in the manuscript. The authors indicate that the pressures of formation must range between 0.3 GPa (assuming a present-day ophiolite thickness of 8-10 km) and 1.2 GPa (the pressure for the metamorphic sole formation).

We had taken the highest peak pressure found in metamorphic sole as a theoretical maximum upper bound, but we realize that lower- to mid-pressure range recorded by the sole is more adequate as an upper limit due to the higher temperature in the sole. We modified line 89-91 to: “The pressure of listvenite formation is poorly constrained, with a possible range from ~ 0.3 GPa [...] to the pressure recorded by the metamorphic sole as an upper bound (0.7 – 1.0 GPa ref 36, ref 37)”.

The temperature estimates are inconsistent with a formation at high-pressure (HP) conditions. Indeed, a formation at 35 km depths implies thermal gradients of 2.3°C/km and 4.2°C/km (for temperatures of 80°C and 150°C, respectively). Those estimates are too cold to be realistic on Earth. Therefore, a formation during subduction infancy and the metamorphic sole formation (from > 97 Ma and 92 Ma) can be excluded.

It is correct that we have not taken into account thermal considerations sufficiently here, which was mainly because the discussion needed to be very short if it was to be included in the main text. We agree that listvenite formation during subduction infancy appears highly unlikely, thus we have included this argumentation in the supplementary discussion text. We note however that because the contacts between listvenites and the metamorphic sole are faults with unknown amount of displacement (Kelemen et al., 2020; Menzel et al., 2020), it is unclear whether carbonation occurred in-situ, along the contact with the metamorphic sole, or if listvenite-bearing basal peridotites and the sole were tectonically juxtaposed after carbonation took place.

And as a side note, I invite the authors to replace the reference for the P estimates of the sole. I understand that Kotowski et al. are part of the same group working on the ICDP project, but their P estimates of 1 GPa are poorly done and meaningless. Those estimates were calculated from the Si-content in white mica in mafic rocks. However, the barometer of Massonne and Schreyer (1987) should not be applied for this kind of lithologies (without phlogopite/biotite and K-feldspar). Moreover, the HP-LT estimates are inconsistent with the Ca-rich chemistry of the amphibole. So, please, refer to Soret et al. (2017) who provided much more robust P-T conditions, and the first ones derived from pseudosection modelling. I understand that there is only a limited number of references allowed, but this is why the references should be the most appropriate.

We have added reference to Soret et al. (2017) in line 91 since we agree that that contribution contains very robust P-T estimates. But we respectfully disagree with the suggestion to remove the reference to Kotowski et al. (2021). Less so because they formed part of the ICDP project, but because they investigated samples at the exact same location, whereas Soret et al. investigated samples from Wadi Tayin and from the northern Samail ophiolite. It is possible that the sole records some spatial variations of P-T conditions in

different areas. We also note that Kotowski et al did not only apply the mentioned (and here maybe inadequate) Si-in-phengite barometer but also an additional multi-equilibrium approach. This question is however not relevant for this paper.

Mature (oceanic and continental) subductions are usually associated with much colder thermal gradient. For the Oman case, thermal gradient of 10°C/km is estimated from the HP rocks (e.g. Yamato et al., 2009). This value implies a depth of listvenite formation comprised between 8 and 15 km depth. The rocks studied in this study are localized at the base of the Semail ophiolite. Its current thickness is < 10 km, but during the obduction (final emplacement onto the Arabian plate) the ophiolite underwent a major stage of extension drastically reducing its thickness. It is generally accepted that its original thickness was comprised between 15 and 20 km (Hacker et al., 1996; Searle et al., 2015). As a consequence, the formation of listvenite during the mature subduction history (before the obduction) appears to be also unlikely, or at least very speculative.

These are interesting thoughts, but in our view there are too many uncertainties involved in order to reliably pin down (or exclude) tectonic settings for carbonation based on thermal considerations: As site BT1 is located close to the Samail gap fault (Scharf et al., 2019, Tectonics), which has been inferred to have been active already during continental subduction (or derive from a similar, pre-existing large-scale feature that segmented the continental margin) (e.g. Ninkabou et al., 2020), it is not clear at all if the generally accepted thermal gradients and typical thickness of the “normal” Samail ophiolite are applicable here. Because the structural evolution is particularly complex in this region, it is hard to tell how much it differs from the “standard” ophiolite setup. But we note here that with 150 °C and a 10°C/km geotherm the depth of listvenite formation is indeed consistent with the lower estimate of standard ophiolite thickness; a mature to continental subduction setting transitioning into obduction is thus broadly consistent with thermal gradients. We would also like to add that the range of temperature estimates for the listvenites may be due to a prolonged or multi-stage history of fluid-rock interaction, with early carbonation occurring at higher temperatures and a transition to lower temperature carbonation with cooling (see discussion by Beinlich et al., 2020). How long this time span of active carbonation and related variation in P-T were is however unclear, which adds additional uncertainty to inferences about the tectonic setting from thermal conditions.

As outlined in the supplementary tectonic discussion summary, the most likely setting in our opinion is listvenite formation related to a mature or continental subduction / ophiolite obduction setting at around 85 – 74 Ma (e.g., Garber et al., 2021; Warren et al., 2003) in comparatively shallow (possibly 0.3 – 0.7 GPa) and cold upper-plate peridotite due to updip migration of fluids derived from dehydration of lower-plate sediments deeper down along the plate interface where temperatures are higher and devolatilization reactions more efficient. Thus, a subduction-obduction setting and carbonation during thrusting does not imply that the listvenite formed at high pressure, but rather that carbonation may have been a consequence of partial subduction and devolatilization of carbonate-silicate sediments and updip CO₂-fluid migration. We do not exclude a post-obduction extension setting either, although such a scenario is similarly speculative and fails to explain the source of CO₂-rich fluid. The detailed context thus remains an open question until more precise dating and/or structural reconstruction is available.

The date of 97 +/- 29 Ma should not be used to discriminate the tectonic environment. It comes from a single sample and it is imprecise: the Rb/Sr analyses are highly scattered along the isochron (unfortunately no MSWD is provided).

We agree that the low precision of this date does not allow clear and certain discrimination of the tectonic context. But, being the only available date for listvenite formation, it cannot be ignored either. In the manuscript text we therefore list a range of tectonic settings that are consistent with this uncertain age constrain and other data to provide the reader with the alternative models that are debated; and in the supplementary discussion summary we refer the reader to Kelemen et al. (2022) where this is discussed further.

I do not know when the listvenite formed, but the reactions probably did not occur during the subduction. Both the reviewers and the authors agree that the Oman's tectonic setting is not the main topic of the manuscript. Since it is mainly speculative, I suggest to remove this aspect and/or to keep the discussion more open than it is presently.

March 16

M. Soret

As recommended by the reviewer, we removed the disputed part of the discussion of the tectonic setting of listvenite formation at site BT1 (lines 390 – 401 in the previous version), and keep the remaining short discussion open to all possible settings. We further included in the supplementary material a short summary of the discussion of the tectonic setting (again keeping it open, not excluding a post-obduction extensional setting), which we hope solves some possible misunderstandings and serves to spark future research on this question.